# Small molecule SWELL1 complex induction improves glycemic control and nonalcoholic fatty liver disease in murine Type 2 diabetes

Susheel K. Gunasekar[1,13], Litao Xie[1,13], Ashutosh Kumar[1,13], Juan Hong[1], Pratik R. Chheda [2], Chen Kang[1], David M. Kern [3,4], Chau My-Ta[5], Joshua Maurer[1], John Heebink[1], Eva E. Gerber [3,4], Wojciech J. Grzesik[6], Macaulay Elliot-Hudson[7], Yanhui Zhang[8], Phillip Key[1], Chaitanya A. Kulkarni[2], Joseph W. Beals[9], Gordon I. Smith[9], Isaac Samuel[10], Jessica K. Smith[10], Peter Nau[10], Yumi Imai[7], Ryan D. Sheldon[11], Eric B. Taylor [11], Daniel J. Lerner[12], Andrew W. Norris [6], Samuel Klein[9], Stephen G. Brohawn [3,4], Robert Kerns [2] & Rajan Sah [1✉]

Type 2 diabetes is associated with insulin resistance, impaired pancreatic β-cell insulin secretion, and nonalcoholic fatty liver disease. Tissue-specific SWELL1 ablation impairs insulin signaling in adipose, skeletal muscle, and endothelium, and impairs β-cell insulin secretion and glycemic control. Here, we show that $I_{Cl,SWELL}$ and SWELL1 protein are reduced in adipose and β-cells in murine and human diabetes. Combining cryo-electron microscopy, molecular docking, medicinal chemistry, and functional studies, we define a structure activity relationship to rationally-design active derivatives of a SWELL1 channel inhibitor (DCPIB/SN-401), that bind the SWELL1 hexameric complex, restore SWELL1 protein, plasma membrane trafficking, signaling, glycemic control and islet insulin secretion via SWELL1-dependent mechanisms. In vivo, SN-401 restores glycemic control, reduces hepatic steatosis/injury, improves insulin-sensitivity and insulin secretion in murine diabetes. These findings demonstrate that SWELL1 channel modulators improve SWELL1-dependent systemic metabolism in Type 2 diabetes, representing a first-in-class therapeutic approach for diabetes and nonalcoholic fatty liver disease.

[1] Department of Internal Medicine, Cardiovascular Division, Washington University School of Medicine, St. Louis, MO, USA. [2] Department of Pharmaceutical Sciences and Experimental Therapeutics, University of Iowa, College of Pharmacy, Iowa City, IA, USA. [3] Department of Molecular & Cell Biology, University of California Berkeley, Berkeley, CA, USA. [4] Helen Wills Neuroscience Institute, University of California Berkeley, Berkeley, CA, USA. [5] Feinberg School of Medicine, Northwestern University, Chicago, IL, USA. [6] Stead Family Department of Pediatrics, Endocrinology and Diabetes Division, Fraternal Order of Eagles Diabetes Research Center, University of Iowa, Iowa City, IA, USA. [7] Department of Internal Medicine, Cardiovascular Division, University of Iowa, Iowa City, IA, USA. [8] Xiamen Cardiovascular Hospital, Xiamen University, Xiamen, China. [9] Center for Human Nutrition, Washington University School of Medicine, St. Louis, USA. [10] Department of Surgery, University of Iowa, Carver College of Medicine, Iowa City, IA, USA. [11] Department of Biochemistry, University of Iowa, Iowa City, IA, USA. [12] Senseion Therapeutics Inc, BioGenerator Labs, St Louis, MO, USA. [13]These authors contributed equally: Susheel K. Gunasekar, Litao Xie, Ashutosh Kumar. ✉email: rajan.sah@wustl.edu

Type 2 diabetes mellitus (T2D) is a globally ubiquitous metabolic disease characterized by hyperglycemia that is caused by reduced insulin sensitivity in target tissues and impaired insulin secretion from pancreatic β-cells[1–3]. T2D accounts for 90–95% of all diabetes mellitus in the United States, or about 24 M people[4]. It is associated with increased risk of cardiovascular disease, renal disease, liver disease, cancer, and infection and a hazard ratio for all-cause mortality of 1.80 compared to patients without T2D[5,6]. The cost of medical care for patients with diabetes is 2.3-fold the cost in non-diabetics. In 2017, the direct medical cost of diabetes in the United States was $237B[7].

There are at least ten distinct classes of medications approved to treat T2D: sulfonylureas, meglitinides, amylin mimetics, biguanides, alpha-glucosidase inhibitors, thiazolidinediones, glucagon-like peptide-1 analogs (GLP-1a), dipeptidyl peptidase-4 inhibitors (DPPi), sodium-glucose co-transporter (SGLT)-2 inhibitors (SGLT2i), and insulin. Despite this diverse array of T2D medications, there are several reasons why new medications for T2D are needed. First, cardiovascular disease (CVD) is the leading cause of death in diabetics[8,9], and although newer T2D medications like SGLT2i and GLP-1a affect a reduction in CVD mortality, significant residual CVD mortality remains[10], which presents a therapeutic opportunity for T2D medications with novel mechanisms of action. Second, 25–33% of patients with T2D have inadequate glycemic control, with HbA1c levels above guideline recommendations[6,11–14]. This poor glucose control is associated with an increased risk of death from vascular causes, non-vascular causes, and cancer[8]. Third, T2D medication-induced hypoglycemia remains a significant problem for patients with T2D, especially with patients on multiple T2D medications[4,15,16]. For all these reasons, there remains sustained interest in developing new T2D and metabolic syndrome therapeutics, especially with novel mechanisms of action[17].

*SWELL1 or LRRC8a* (leucine-rich repeat-containing protein 8a) encodes a transmembrane protein first described in 2003 as the site of a balanced translocation in an immunodeficient child with agammaglobulinemia and absent B-cells[18,19]. Subsequent work revealed that this condition was caused by impaired SWELL1-dependent GRB2-PI3K-AKT signaling in lymphocytes, resulting in a developmental block in lymphocyte differentiation[20]. So, for about a decade, SWELL1 was considered a membrane protein that regulates PI3K-AKT mediated lymphocyte function[18,19], and it was not until 2014 that SWELL1/LRRC8a was discovered to also form an essential component of the volume-regulated anion channel (VRAC)[21,22], forming hetero-hexamers with LRRC8b-e[22,23]. Therefore, historically, the SWELL1-LRRC8 complex was first described as a membrane protein that participated in non-ion channel-mediated protein–protein signaling (non-conductive signaling) and then later found to form an ion channel complex with ion conductive signaling properties. Indeed, prior work highlights each of these modes of SWELL1-LRRC8 channel complex signaling. We previously showed SWELL1 to mediate insulin-PI3K-AKT signaling in adipocytes, skeletal muscle, and endothelium via non-conductive signaling mechanisms[24–27]. Also, we and others showed SWELL1-LRRC8 channel activity (conductive signaling) in the pancreatic β-cell is required for normal insulin secretion[28,29]. Thus, SWELL1-LRRC8 loss-of-function both downregulates insulin signaling in target tissues[24,30] and insulin secretion from the pancreatic β-cell[28,29] inducing a state of glucose intolerance[24,28,30]. Since T2D is characterized by both a loss of insulin sensitivity of target tissues (fat, skeletal muscle, liver) and ultimately, impaired insulin secretion from the pancreatic β-cell[1–3], these data raised the question: could impaired SWELL1-mediated signaling contribute to T2D pathogenesis, and if so,

could this be corrected pharmacologically to improve systemic glycemia?

In this study, we provide evidence that SWELL1-mediated currents and SWELL1 protein are reduced in murine and human adipocytes and pancreatic β-cells in the setting of T2D and hyperglycemia suggesting that dysfunctional SWELL1-mediated signaling could contribute to T2D pathogenesis by impairing insulin sensitivity and insulin secretion. Next, we identify a small molecule modulator, DCPIB (renamed SN-401), as a tool compound that binds the SWELL1-LRRC8 complex[31], and potentially functions as a chemical chaperone to augment SWELL1 expression and plasma membrane trafficking at concentrations 5-fold lower than its IC$_{50}$ of ~5 μM for I$_{Cl,SWELL}$[32]. In vivo, SN-401 normalizes glucose tolerance by increasing insulin sensitivity and secretion in insulin-resistant T2D mouse models, while augmenting tissue glucose uptake, suppressing hepatic glucose production, inducing serum FGF21 levels, and reducing hepatic steatosis and hepatocyte damage (ballooning) in obese T2D mice. Importantly, while SN-401 normalizes glycemia in diabetic mice, it has very mild glucose-lowering effects on non-obese euglycemic mice— indicating a low risk of hypoglycemic events associated with other commonly used anti-diabetic therapies, including sulfonylureas and insulin. Combining cryo-EM structure data of SN-401 bound to its target SWELL1/LRRC8a[31] with molecular docking simulations, and cryo-EM structure data of an active SN-40X congener bound to SWELL1 hexameric channels in lipid nanodiscs, we validate a structure-activity relationship (SAR) based approach to generate SN-401 congeners with subtle molecular changes to "tune" on-target activity, both in vitro and in vivo. This approach allows us to attribute the cellular and systemic SN-40X effects to drug-target binding while controlling for off-target effects. We propose small molecule SWELL1 modulators may represent a first-in-class therapeutic approach to treat metabolic syndrome and associated diseases by restoring SWELL1 signaling across multiple organ systems that are dysfunctional in T2D.

## Results

**I$_{Cl,SWELL}$ and SWELL1 protein are reduced in T2D β-cells and adipocytes.** SWELL1/LRRC8a ablation impairs insulin signaling in target tissues[24,30] and insulin secretion from the pancreatic β-cell[28,29], inducing a pre-diabetic state of glucose intolerance[24,28,30]. These recent findings suggest that reductions in SWELL1 may contribute to Type 2 diabetes (T2D). To determine if SWELL1-mediated currents are altered in T2D we measured I$_{Cl,SWELL}$ in pancreatic β-cells freshly isolated from T2D mice raised on HFD for 5–7 months (Fig. 1a, c) and from individuals with T2D (Fig. 1b, d and Supplementary Table S1) compared to non-T2D controls. In both mouse and human T2D β-cells, the maximum I$_{Cl,SWELL}$ current density (measured at +100 mV) upon stimulation with hypotonic swelling is significantly reduced (90% in murine; 63% in human, Fig. 1c, d) compared to non-T2D controls, similar to reductions observed in SWELL1 knockout (KO) and knockdown (KD) murine and human β-cells[28], respectively. As SWELL1/LRRC8a is a critical component of I$_{Cl,SWELL}$/VRAC[21,22] in both adipose tissue[24,30] and β-cells[28,29], we asked whether these reductions in I$_{Cl,SWELL}$ in the setting of T2D[33] are associated with reductions in SWELL1 protein expression. Total SWELL1 protein in murine islets (largely representing β-cells) isolated from T2D mice raised on HFD for 33 weeks (40 weeks old) is significantly reduced 66% compared to islets from lean, non-T2D control mice (Fig. 1e). Similarly, diabetic human cadaveric islets (representing numerous islet cell types) also shows a trend toward being reduced 50% compared to islets from non-diabetics (Fig. 1f and

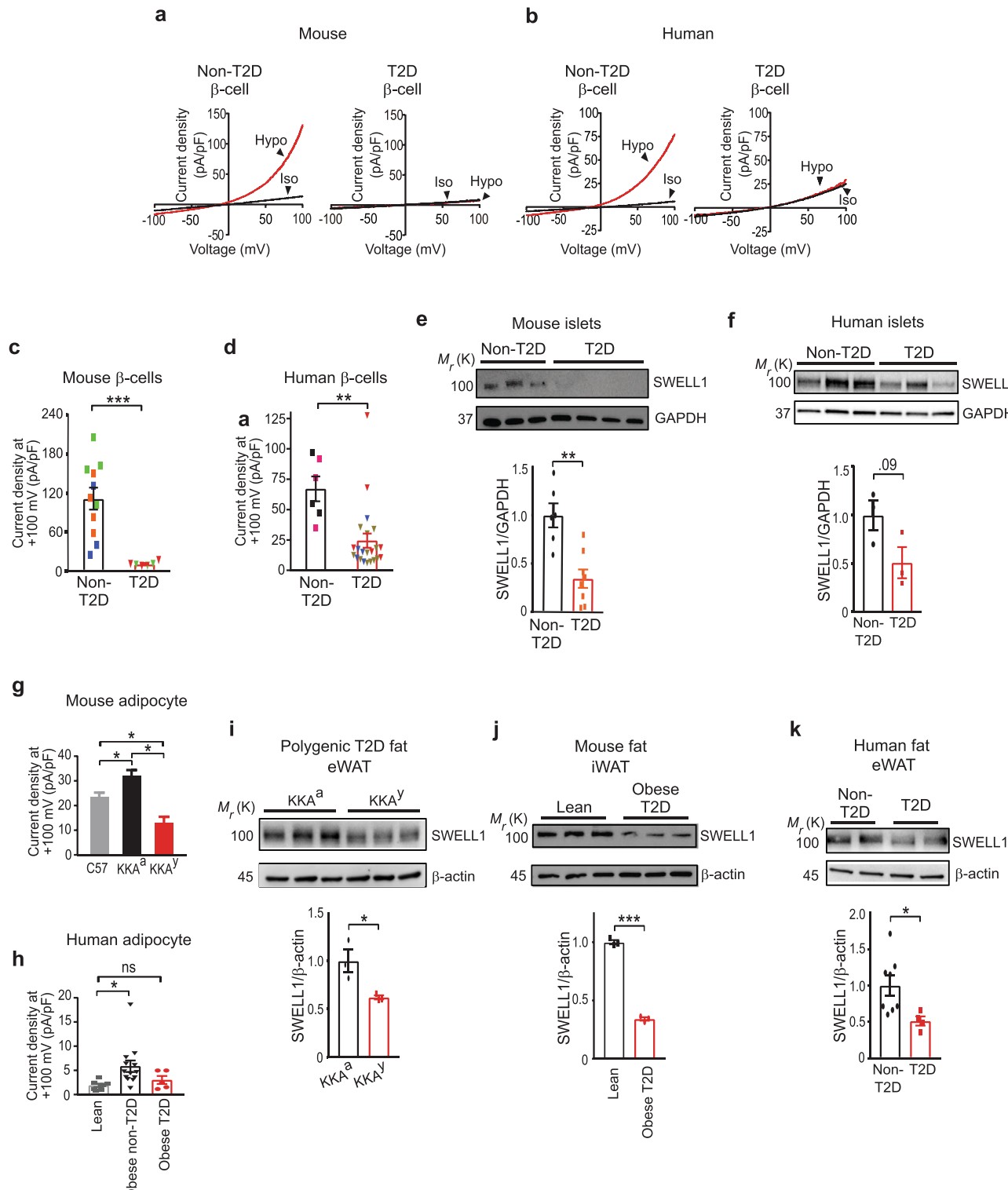

Supplementary Table S2), suggesting that reduced SWELL1 protein may underlie these reductions in $I_{Cl,SWELL}$ currents.

Curiously, reductions in β-cell $I_{Cl,SWELL}$ observed in the setting of T2D (Fig. 1a–d) are consistent with previous measurements of VRAC/$I_{Cl,SWELL}$ in the adipocytes of the murine KKA[y] T2D model[33], which are reduced by 60% in T2D KKA[y] mice compared to KKA[a] controls[33] (Fig. 1g, data plotted from Inoue et al., 2010[33]). Similarly, SWELL1-mediated $I_{Cl,SWELL}$ measured in isolated human adipocytes from a diabetic individual with obesity (BMI = 52.3, HgbA1c = 6.9%; Fasting glucose = 148–151 mg/dl)

show a trend toward being reduced 48% compared to non-diabetic individuals with obesity that we reported previously[24], and not different from $I_{Cl,SWELL}$ in adipocytes from lean individuals (Fig. 1h and Supplementary Table S3). Consistent with reductions observed in murine and human adipocyte $I_{Cl,SWELL}$, SWELL1 protein is also reduced (38%) in adipose tissue of T2D KKA[y] mice as compared to parental control KKA[a] mice (Fig. 1i), and 66% in obese T2D mice raised on HFD for 33 weeks (40 weeks old) compared to lean mice (Fig. 1j). Similarly, SWELL1 protein is 50% lower in adipose tissue from

**Fig. 1 $I_{Cl,SWELL}$ and SWELL1 protein are reduced in T2D β-cells and adipocytes. a, b** Current-voltage plots of $I_{Cl,SWELL}$ measured in non-T2D and T2D mouse (**a**) and human (**b**) β-cells at baseline (iso, black) and with hypotonic (210 mOsm) stimulation (hypo, red). **c, d** Mean outward $I_{Cl,SWELL}$ current densities at +100 mV from non-T2D ($n = 11$ cells) and T2D ($n = 6$ cells) mouse (**c**) and non-T2D ($n = 6$ cells) and T2D ($n = 22$ cells) human (**d**) β-cells. Symbol color represents cells recorded from different individual mice or humans. **e** Representative western blot for SWELL1 in islets isolated from C57 mice fed HFD for 33 weeks (T2D, $n = 8$) and age-matched controls (non-T2D, $n = 6$). **f** Representative western blot for SWELL1 in cadaveric islets of non-T2D and T2D donors ($n = 3$ each). **g** Mean outward $I_{Cl,SWELL}$ current densities at +100 mV from adipocytes isolated from epididymal fat of C57[$], control strain KKA[a$], and polygenic-T2D KKA[y$] mice ($n = 14–34$ cells). **h** Mean outward $I_{Cl,SWELL}$ current densities recorded at +100 mV from adipocytes isolated from lean group[#] ($n = 7$ cells), obese non-T2D[#] ($n = 13$ cells), and T2D ($n = 5$ cells) groups. **i** Representative western blot for SWELL1 in epididymal fat isolated from polygenic-T2D KKA[y] mice compared to parental control strain KKA[a] ($n = 3$ each). **j** Representative western blot for SWELL1 protein from inguinal fat in C57 mice fed HFD (Obese T2D) for 33 weeks compared to lean mice ($n = 3$ each). **k** Representative western blot comparing SWELL1 in visceral fat from individuals with non-T2D ($n = 8$) and T2D ($n = 4$). [$]Data for C57, KKA[a], and KKA[y] mouse adipocytes are plotted from previously reported tabular data in Inoue et al. (2010). [#]Lean and obese non-T2D adipocytes are plotted from previously reported data in Zhang et al. (2017) for comparison in **g** and **h**. Data were represented as mean ± SEM. A two-tailed unpaired $t$-test was used in **c–f** and **i–k**. One-way ANOVA for mouse and two-tailed permutation $t$-test group comparison for human adipocytes were used in **g** and **h** respectively. *, **, and *** represent $p$ values of <0.05, <0.01, and <0.001.

individuals with hyperglycemic T2D and obesity (HgbA1c > 6.0%) compared to adipose tissue from normoglycemic individuals (HgbA1c < 6.0% Fig. 1k and Supplementary Table S4). Taken together, these findings suggest reduced SWELL1 activity in adipocytes and β-cells (and possibly other tissues) may underlie insulin resistance and impaired insulin secretion associated with T2D. Moreover, SWELL1 protein expression increases in both adipose tissue and liver in the setting of early euglycemic obesity[30], and shRNA-mediated suppression of this SWELL1 induction exacerbates insulin resistance and glucose intolerance[30]. Similar to these prior findings, hepatic SWELL1 protein is induced ninefold in mice raised on HFD for 33 weeks compared to lean, non-T2D mice (Supplementary Fig. 1a), and increased ~3.2-fold in T2D KKA[y] mice compared to non-T2D control KKA[a] mice (Supplementary Fig. 1b). This is in contrast to the reductions of SWELL1 protein, and $I_{Cl,SWELL}$ observed in T2D adipose and islets/β-cells. Therefore, we speculate that maintenance or induction of SWELL1 expression/signaling in peripheral tissues may support insulin sensitivity and secretion to preserve systemic glycemia in the setting of T2D.

**SWELL1 protein expression regulates insulin-stimulated PI3K-AKT2-AS160 signaling.** To test whether SWELL1 regulates insulin signaling, we re-expressed Flag-tagged SWELL1 (SWELL1 O/E) in SWELL1 KO 3T3-F442A adipocytes and measured insulin-stimulated phosphorylated AKT2 (pAKT2; Ser474) and phosphorylated AS160 (pAS160; Ser1177) as a readout of insulin signaling (Fig. 2a, b). SWELL1 KO 3T3-F442A adipocytes exhibit significantly blunted insulin-mediated pAKT2 and pAS160 signaling compared to WT adipocytes, similar to described previously[24,26], and this is fully rescued by reexpression of SWELL1 in SWELL1 KO adipocytes (KO + SWELL1 O/E, Fig. 2a, b)[26]. SWELL1 reexpression also recapitulates SWELL1-mediated $I_{Cl,SWELL}$ in SWELL1 KO cells in response to hypotonic stimulation (Fig. 2c and Supplementary Fig. S2a–c), which is consistent with restoration of SWELL1-LRRC8a signaling complexes at the plasma membrane. Notably, the reductions in total AKT2 protein expression observed in SWELL1 KO adipocytes is not rescued by SWELL1 reexpression, indicating that transient changes in SWELL1 protein expression in adipocytes regulate pAKT2 signaling, as opposed to total AKT2 protein expression. We confirmed FLAG-tagged SWELL1 traffics normally to the plasma membrane when expressed in both WT and SWELL1 KO adipocytes visualized by immunofluorescence (IF) using anti-FLAG and SWELL1 KO-validated anti-SWELL1 antibodies, respectively (Supplementary Fig. S2d, e). FLAG-tagged SWELL1 overexpressed in WT and SWELL1 KO adipocytes assume a punctate pattern at the cell periphery, similar to endogenous SWELL1 in WT adipocytes. Overall, these data indicate that SWELL1 expression levels regulate insulin-PI3K-AKT2-AS160 signaling in adipocytes—potentially by

modulating GRB2 signaling[20,24–26]. Furthermore, these data imply that pharmacological SWELL1 induction in peripheral tissues in the setting of T2D may enhance insulin signaling and improve systemic insulin sensitivity and glycemic control.

**A small molecule binds SWELL1-LRRC8 channel complexes, increases adipocyte SWELL1 protein expression, and SWELL1-dependent insulin signaling.** The small molecule 4-[(2-Butyl-6,7-dichloro-2-cyclopentyl-2,3-dihydro-1-oxo-1*H*-inden-5-yl)oxy]butanoic acid (*DCPIB*, Fig. 2d) is among a series of structurally diverse (acylaryloxy)acetic acid derivatives, that were synthesized and studied for diuretic properties in the late 1970s[34,35] and evaluated in the 1980s as potential treatments for brain edema[36,37]. While DCPIB was derived from the FDA-approved diuretic ethacrynic acid, it has minimal diuretic activity[38] and has instead been used as a selective VRAC/$I_{Cl,SWELL}$ inhibitor[24,28,32] (Fig. 2e), binding at a constriction point within the SWELL1-LRRC8 hexamer[31,39–41]. Having demonstrated that SWELL1 regulates insulin-AKT2 signaling in multiple cell types, including adipocytes[24,25,30], skeletal muscle[26], and endothelium[27], we anticipated pharmacological inhibition of VRAC/$I_{Cl,SWELL}$ with DCPIB, which we here re-name SN-401, would decrease insulin signaling. Unexpectedly, SN-401 increased SWELL1 protein expression in 3T3-F442A preadipocytes (threefold control expression; Fig. 2f, g) and adipocytes (1.5-fold control expression; Fig. 2h) when applied for 96 h, and was associated with enhanced insulin-stimulated levels of pAKT2 (Fig. 2f, g, i, j), and insulin-stimulated levels of pAS160 (Fig. 2i, j). These SN-401-mediated effects on insulin-AKT2-AS160 signaling are absent in SWELL1 KO 3T3-F442A adipocytes, consistent with an on-target SWELL1-mediated mechanism of action for SN-401 (Fig. 2i, j). The SN-401-mediated increases in SWELL1 protein expression are not associated with increases in SWELL1 mRNA, nor in the mRNA for other LRRC8 subunits: LRRC8b, LRRC8c, LRRC8d, or LRRC8e that form the SWELL1 channel complex (Supplementary Fig. S3a–c), implicating post-transcriptional mechanisms for increased SWELL1 expression and SWELL1-LRRC8 associated signaling.

**SN-401 increases SWELL1 and improves systemic glucose homeostasis in murine T2D models by enhancing insulin sensitivity and secretion.** To determine if SN-401 improves insulin signaling and glucose homeostasis in vivo we treated two T2D mouse models: obese, HFD-fed mice and the polygenic-T2D KKA[y] mouse model with SN-401 (5 mg/kg i.p. for 4–10 days). In vivo, SN-401 augmented SWELL1 expression twofold in adipose tissue of HFD-fed T2D (Fig. 3a), threefold in adipose tissue of

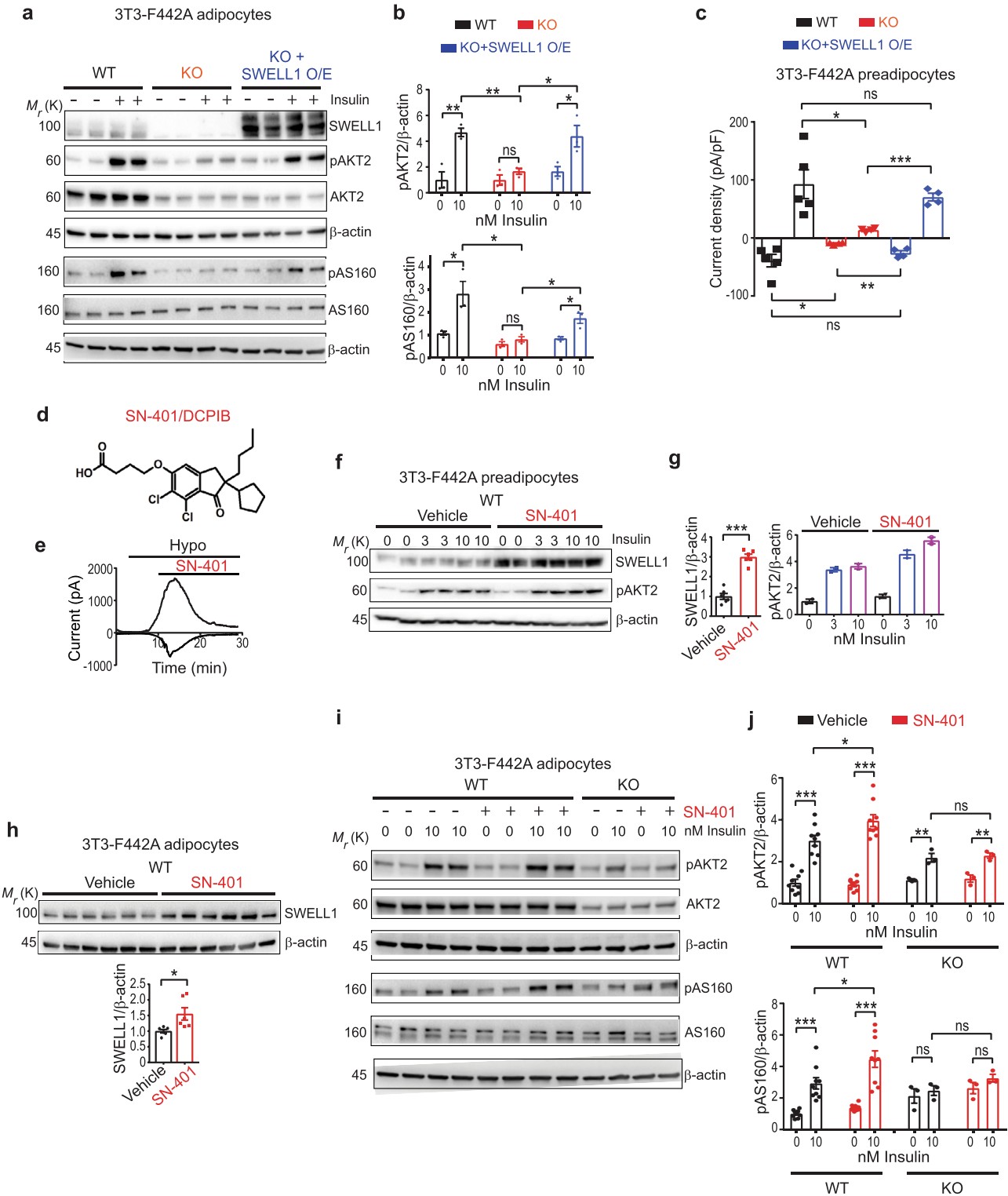

KKA$^y$ T2D (Fig. 3b) mice, and was associated with improved fasting blood glucose (FG), glucose tolerance (GTT), and insulin tolerance (ITT) in both T2D models (Fig. 3c–f). These metabolic improvements were not associated with significant differences in body weight (Supplementary Table S5). Adipocyte size was mildly reduced 9% in KKAy mice (Fig. 3g) and unchanged in the HFD-fed mice (Fig. 3h) after SN-401 treatment for 5 days as compared to vehicle. Remarkably, treating the control KKA$^a$ parental strain with SN-401 at the same treatment dose (5 mg/kg × 4–10 days) does not cause hypoglycemia, nor does it alter glucose and insulin

tolerance (Fig. 3d–f). Similarly, lean, non-T2D, glucose-tolerant mice treated with SN-401 have similar FG, GTT and ITT compared to vehicle-treated mice (Fig. 3i, j and Supplementary Fig. S4a–c). However, when made insulin-resistant and diabetic after 16 weeks of HFD-feeding, these same mice (from Fig. 3i, j) treated with SN-401 show marked improvements in FG (Fig. 3k), GTT, and ITT (Fig. 3l) as compared to vehicle. These data show that SN-401 restores glucose homeostasis in the setting of T2D, but has little effect on glucose homeostasis in non-T2D mice. Importantly, this portends a low risk for inducing hypoglycemia.

**Fig. 2 SWELL1 protein expression regulates insulin-stimulated PI3K-AKT2-AS160 signaling. a, b** Western blots detecting SWELL1, pAKT2[Ser474], AKT2, pAS160[Thr642], AS160, and β-actin in wildtype (WT, black), SWELL1 knockout (KO, red), and adenoviral reexpression of SWELL1 in KO (KO + SWELL1 O/E, blue) 3T3-F442A adipocytes stimulated with 0 and 10 nM insulin for 15 min (**a**) and densitometric quantification for pAKT2/β-actin and pAS160/β-actin (**b**) (n = 3 independent experiments for each condition). **c** Mean inward and outward current densities recorded at −100 and +100 mV from WT (black, n = 5 cells), KO (red, n = 4 cells), and KO + SWELL1 O/E (blue, n = 4 cells) 3T3-F442A preadipocytes. **d** SN-401/DCPIB chemical structure. **e** $I_{Cl,SWELL}$ inward and outward current over time upon hypotonic (210 mOsm) stimulation and subsequent inhibition by 10 μM SN-401 in a HEK-293 cell. **f, g** Western blots detecting SWELL1, pAKT2[Ser474], and β-actin in WT 3T3-F442A preadipocytes treated with either vehicle or SN-401 (10 μM) for 96 h (**f**) (n = 2 each) and the corresponding densitometric ratio (**g**). **h** Western blots detecting SWELL1 and β-actin in WT 3T3-F442A adipocytes treated with either vehicle or SN-401 (10 μM) for 96 h (n = 6 each) and the corresponding densitometric ratio below. **i, j** Western blots detecting pAKT2[Ser474], AKT2, pAS160[Thr642], AS160, and β-actin in WT (n = 9 each) and SWELL1 KO (n = 3 each) 3T3-F442A adipocytes treated with either vehicle or SN-401 (500 nM) for 96 h and stimulated with 0 and 10 nM insulin for 15 min (**i**) and the corresponding densitometric ratio for pAKT2/β-actin and pAS160/β-actin respectively (**j**). Data were represented as mean ± SEM. Two-tailed unpaired t-test was used in **b**, **c**, **g**, **h**, and **j** where *, **, and *** represent p values of <0.05, <0.01, and <0.001, respectively. ns difference did not exceed the threshold for significance.

SN-401 was well-tolerated during chronic i.p. injection protocols, with no overt signs of toxicity with daily i.p. injections for up to 8 weeks, despite striking effects on glucose tolerance (Supplementary Fig. S4d).

To examine SN-401-mediated enhancements in insulin secretion from pancreatic β-cells, we next measured glucose-stimulated insulin secretion (GSIS) in SN-401 treated mice subjected to 21 weeks of HFD. We found that the impairments in GSIS commonly observed with long-term HFD (21 weeks HFD) are significantly improved in SN-401-treated HFD mice based on serum insulin measurements (Fig. 3m) and perifusion GSIS from isolated islets (Fig. 3n), consistent with the predicted effect of SWELL1 induction in pancreatic β-cells[28,29]. Similar results were obtained from islets isolated from SN-401 treated T2D KKA[y] mice compared to vehicle-treated mice (Fig. 3o). Collectively, these data suggest that SN-401-mediated improvements in systemic glycemia in T2D occur via augmentation of both peripheral insulin sensitivity and β-cell insulin secretion—the inverse phenotype observed in in vivo loss-of-function studies[24,28–30].

**SN-401 improves systemic insulin sensitivity, tissue glucose uptake, and nonalcoholic fatty liver disease in murine T2D models.** To more rigorously evaluate SN-401 effects on insulin sensitization and glucose metabolism in T2D mice we performed euglycemic hyperinsulinemic clamps traced with [3]H-glucose and [14]C-deoxyglucose in T2D KKA[y] mice treated with SN-401 or vehicle. SN-401 treated T2D KKA[y] mice require a higher glucose-infusion rate (GIR) to maintain euglycemia compared to vehicle, consistent with enhanced systemic insulin sensitivity (Fig. 4a). The rate of glucose appearance (R$_a$), which reflects hepatic glucose production from gluconeogenesis and/or glycogenolysis, was reduced 40% in SN-401-treated T2D KKA[y] mice at baseline (Basal, Fig. 4b), and further suppressed 75% during glucose/insulin infusion (Clamp, Fig. 4b), revealing SN-401 increases hepatic insulin sensitivity—similar to thiazolidinediones (TZD)[42].

As the SN-401-mediated increase in SWELL1 signaling is expected to enhance insulin-pAKT2-pAS160 signaling, GLUT4 plasma membrane translocation, and tissue glucose uptake[24], we next measured the effect of SN-401 on tissue glucose uptake in KKA[y] mice using 2-deoxyglucose (2-DG). SN-401 enhanced insulin-stimulated 2-DG uptake into inguinal white adipose tissue (iWAT), gonadal white adipose tissue (gWAT), and myocardium (Fig. 4c). Similarly, we found previously that SWELL1 ablation markedly reduces insulin-pAKT2-pGSK3β signaling[24,26,27] and cellular glycogen content[24], and accordingly asked whether the SN-401-mediated increase in SWELL1 signaling would increase glucose incorporation into tissue glycogen in the setting of T2D. Indeed, SN-401 markedly increased glucose incorporation into

glycogen in liver, adipose, and skeletal muscle (Fig. 4d). Consistent with improved glucose metabolism in peripheral tissues, we observe a trend toward SN-401 mediated SWELL1 induction in skeletal muscle (Supplementary Fig. 5a, b) in HFD-T2D mice (18 weeks HFD), associated with increases in downstream pAKT2, pAS160, and pGSK3β (Ser9) signaling (Supplementary Fig. 5a, b). In contrast to adipose and skeletal muscle, SN-401 does not induce hepatic SWELL1 protein in T2D KKA[y] mice (Supplementary Fig. 5c), nor when applied for 24–48 h to cultured primary hepatocytes (Supplementary Fig. 5d–f). However, SN-401 application in vitro to cultured primary hepatocytes robustly induces pAS160 in a SWELL1-dependent manner, but without significant pAKT2 activation (Supplementary Fig. 5d, e, g). Collectively, these data suggest that SN-401 mediated effects on systemic metabolism in vivo occur via SWELL1 modulation in multiple tissues, including adipose, skeletal muscle, liver, and pancreatic islets.

Nonalcoholic fatty liver disease (NAFLD), like T2D, is associated with insulin resistance[43]. NASH is an advanced form of nonalcoholic liver disease defined by three histological features: hepatic steatosis, hepatic lobular inflammation, hepatocyte damage (ballooning) and can be present with or without fibrosis. NAFLD and T2D likely share at least some pathophysiologic mechanisms because more than one-third of patients (37%) with T2D have NASH[44] and almost one-half of patients with NASH (44%) have T2D[45]. Mice raised on HFD for 16 weeks followed by intermittent dosing with SN-401 over the course of 5 weeks (Fig. 4e) were noted to have grossly smaller livers (Fig. 4f), lower hepatic triglyceride concentrations (Fig. 4g), and experienced a mild 14% reduction in body weight compared to vehicle-treated mice (Fig. 4f). Histologic evaluation revealed significant reductions in hepatic steatosis and hepatocyte damage compared to vehicle-treated mice (Fig. 4h, i and Supplemental Fig. S6). The NAFLD-activity score (NAS), which integrates histologic scoring of hepatic steatosis, lobular inflammation, and hepatocyte ballooning[46] (Fig. 4i), also improved >2 points in SN-401-treated mice compared to vehicle-treated mice. These SN-401 mediated reductions in hepatic steatosis and hepatocyte damage are consistent with the observed increases in hepatic insulin sensitivity and consequent reductions in hepatic glucose production via gluconeogenesis available for hepatic de novo lipogenesis (Fig. 4b), as observed with other insulin sensitizers, such as metformin and TZDs[47] that have both primary effects on liver and secondary effects on other peripheral tissues.

To evaluate other possible mechanisms for the observed improvements in systemic metabolism we measured serum adiponectin and Fibroblast Growth Factor 21 (FGF21) in KKA[y] mice treated with SN-401 for 5 days (Fig. 4j, k). Adiponectin is an adipokine that is increased by other anti-diabetic agents, such as TZDs, and is thought to improve glycemic control and lipid

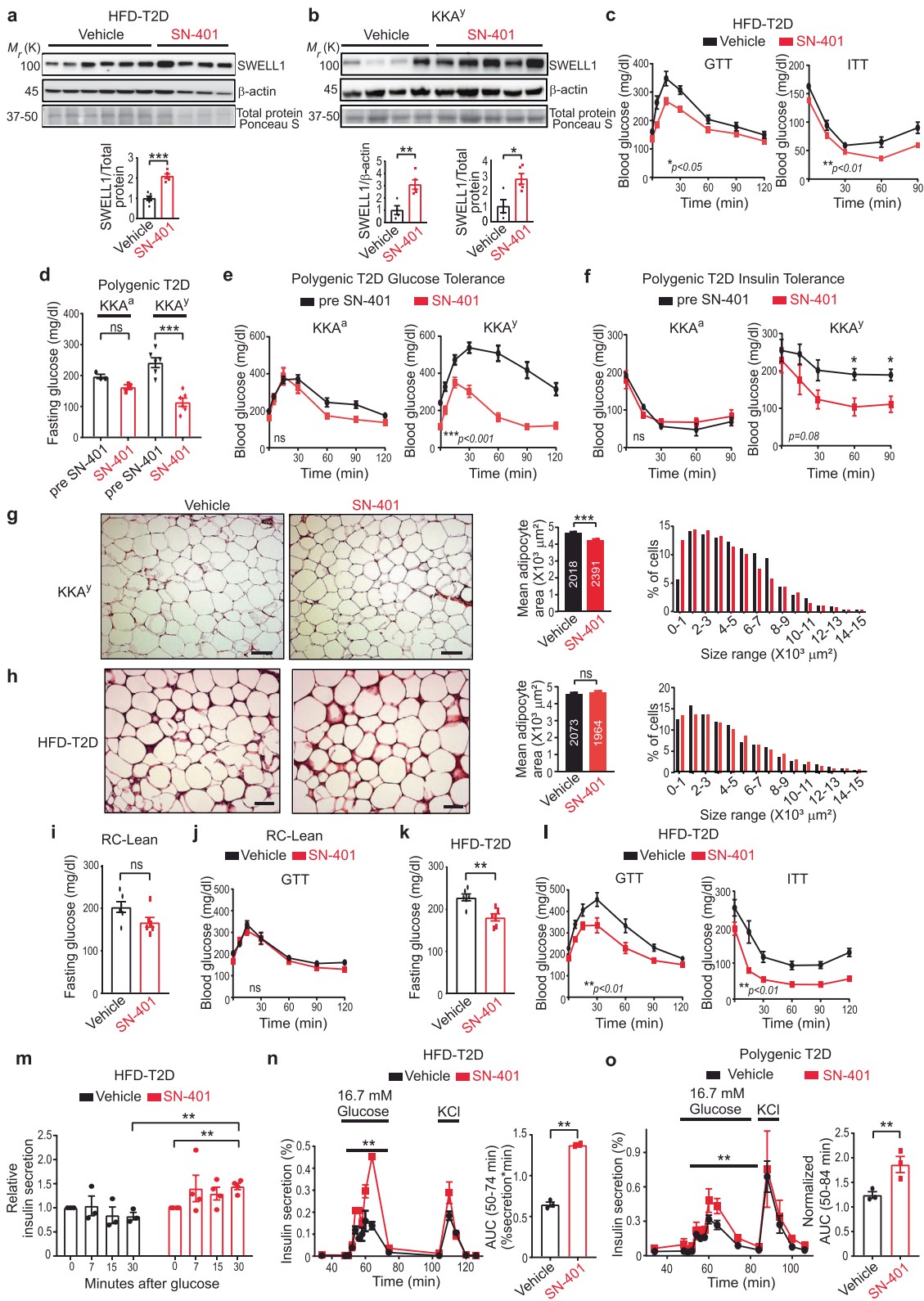

metabolism via effects in the liver, pancreatic β-cells, and adipocytes[48]. FGF21 is an endocrine hormone that is expressed in the liver, pancreas, and adipose tissue, but under physiological conditions is largely a liver-derived hepatokine[49], and is a potent regulator of glucose and lipid metabolism[50]. Curiously, we found that serum adiponectin was unchanged in SN-401 treated KKA[y] mice as compared to vehicle (Fig. 4j), while serum FGF21 was

increased 2.8-fold by SN-401 (Fig. 4k). Taken together, these data reveal that SN-401 augments SWELL1 protein and SWELL1-mediated signaling in some tissues, while also increasing serum FGF21 to concomitantly enhance both systemic insulin sensitivity and pancreatic β-cell insulin secretion, thereby normalizing glycemic control in T2D mouse models. This improved metabolic state can reduce ectopic lipid deposition,

**Fig. 3 SN-401 increases SWELL1 and improves glycemic control in murine T2D models by enhancing insulin sensitivity and secretion. a, b** Western blots for SWELL1 in eWAT of C57BL/6 mice on HFD for 21 weeks treated with vehicle ($n = 6$) or SN-401 ($n = 5$) as in scheme Fig. 4e. **b** Western blots for SWELL1 in inguinal fat of KKA$^y$ mice treated with vehicle ($n = 4$) or SN-401 (5 mg/kg i.p. × 5 days, $n = 5$). **c** Glucose tolerance test (GTT) and insulin tolerance test (ITT) of C57BL/6 mice on HFD for 8 weeks treated with vehicle or SN-401 for 10 days ($n = 7$ mice). **d–f** Fasting glucose (**d**), GTT (**e**), and ITT (**f**) of polygenic-T2D KKA$^y$ mice ($n = 6$) and its control strain KKA$^a$ ($n = 3$) compared pre- and post-SN-401 treatment for 4 days. **g, h** Haematoxylin and eosin (H&E) staining of eWAT, mean adipocyte cross-sectional area and size distribution of KKA$^y$ (**g**, $n = 4$ mice each group**)** and C57 HFD (**h**, $n = 5$ vehicle and $n = 4$ SN-401**)** mice treated with vehicle or SN-401 for 5 days. **i, j** Fasting glucose (**i**) of regular chow-diet fed (RC), lean mice treated with vehicle or SN-401 for 6 days and the corresponding GTT (**j**) ($n = 6$ in each). **k, l** Fasting glucose (**k**), GTT (16 weeks HFD), and ITT (18 weeks HFD) (**l**) of HFD-T2D mice treated with vehicle or SN-401 for 4 days as in Fig. 4e. **m** Relative insulin secretion in plasma of HFD-T2D mice (18 weeks HFD, 4 days treatment) after i.p. glucose (0.75 g/kg BW) treated with vehicle ($n = 3$) or SN-401 ($n = 4$). **n, o** Glucose-stimulated insulin secretion (GSIS) of islets isolated from HFD-T2D mice (21-week HFD) treated with vehicle ($n = 3$ mice, and 3 experimental replicates) or SN-401 ($n = 3$ mice, and 2 experimental replicates) (**n**) and from polygenic-T2D KKA$^y$ mouse treated with vehicle or SN-401 for 6 days ($n = 3$ mice in each group, 3 experimental replicates), (**o**); and area under the curve (AUC) respectively. Mean presented ±SEM. A two-tailed unpaired $t$-test was used in **a, b, g, h, i, k, n**, and **o**. Paired $t$-test used in **d**. Paired (in-group) and unpaired (between-group) $t$-tests were performed in **m**. Two-way ANOVA was used for **c, e, f, j**, and **l**. *, ** and, *** represent $p$ values of $<0.05$, $<0.01$, and $<0.001$, respectively. ns not significant. Scale bar - 100 μm in **g** and **h**.

hepatocyte damage, and NAFLD that is associated with obesity and T2D.

**Chemical synthesis, molecular docking, and cryo-EM reveal specific SN-401-SWELL1 interactions important for on-target binding.** To confirm that SN-401-induced increases in SWELL1 protein and signaling are mediated by direct binding to the SWELL1-LRRC8 channel complex, as opposed to off-target effects, we designed and synthesized SN-401 congeners (Fig. 5a) with subtle structural changes predicted to either enhance (SN-403, SN-406, SN-407), or reduce (Inactive 1, Inactive 2, Inactive 3) SN-401-SWELL1 complex binding (Fig. 5a). The cryo-EM structure of SN-401/DCPIB bound within the SWELL1 homo-hexamer, published by Kern and colleagues[31] revealed that SN-401 binds at a constriction point in the pore wherein the electronegative SN-401 carboxylate group interacts electrostatically with the R103 residue in one or more of the SWELL1 monomers[31]. Moreover, SN-401 appeared to stabilize the pore region of the SWELL1 hexamer in lipid nanodiscs[31]. To characterize the structural features of SN-401 responsible for binding to SWELL1-LRRC8, we performed molecular docking simulations of SN-401 and its analogs into the SWELL1 homo-hexamer (PDB: 6NZZ), and identified two molecular determinants predicted to be critical for SN-401-SWELL1-LRRC8 binding (Fig. 5b i, ii): (1) the length of the carbon chain leading to the anionic carboxylate group predicted to electrostatically interact with one or more R103 guanidine groups (found in SWELL1/LRRC8a and LRRC8b; Fig. 5b-solid circles); and (2) proper orientation of the hydrophobic cyclopentyl group that slides into a hydrophobic cleft at the interface of LRRC8 monomers (conserved among all LRRC8 subunit interfaces; Fig. 5b-broken circles). Docking simulations predicted shortening the carbon chain leading to the carboxylate by 2 carbons would yield a molecule, Inactive 1, that could either interact with R103 through the carboxylate group (Fig. 5c(i)) or have the cyclopentyl ring occupy the hydrophobic cleft (Fig. 5c(ii)), but unable to simultaneously participate in both interactions (Fig. 5c). Similarly, docking simulations predicted removing the butyl group of SN-401 would yield a molecule, Inactive 2, unable to orient the cyclopentyl group into a position favorable for interaction with the hydrophobic cleft without introducing structural strain in the molecule (Supplementary Fig. S7a). Finally, replacing the carboxylic acid with boronic acid, yielded Inactive 3, a molecule predicted to be uncharged at physiological pH, thereby chemically removing the predicted electrostatic interaction of SN-401 with R103. Indeed, docking simulations of Inactive 3 revealed no poses that included an electrostatic interaction with R103, with the molecule largely sitting at the mouth of the pore (Fig. 5d i, ii). Importantly, all of

these structural modifications, predicted to abrogate either carboxylate-R103 electrostatic binding (Inactive 1, Inactive 3) or cyclopentyl-hydrophobic pocket binding (Inactive 2) were sufficient to impair SWELL1 channel binding as demonstrated by a rightward shift in the IC$_{50}$ for I$_{Cl,SWELL}$ inhibition in vitro for Inactive 1 (Fig. 6b, e; IC$_{50}$ = 35.5 μM), Inactive 2 (Fig. 6c, e; IC$_{50}$ = 10.6 μM), and Inactive 3 (Fig. 6d, e: IC$_{50}$ = 18.3 μM) as compared to SN-401 (Fig. 6a, e; IC$_{50}$ = 3.9 μM)[32].

Conversely, lengthening the carbon chain attached to the carboxylate group of SN-401 by 1–3 carbons (Fig. 5a: SN-403, SN-406, and SN-407) was predicted to enhance the R103 electrostatic interactions (Fig. 5e–g; black solid circle), and better orient the cyclopentyl group to bind within the hydrophobic cleft (Fig. 5e–g, broken circle). Additional binding interactions for SN-403, SN-406 and SN-407 are also predicted along the channel, due to the longer carbon chains affording additional hydrophobic interactions with side-chain carbons of the R103 residues (Fig. 5e–g; purple dashes). As anticipated, SN-403, SN-406, and SN-407 had increased I$_{Cl,SWELL}$ inhibitory activity compared to SN-401 (Fig. 6f–i and Supplementary Fig. S7b, c), with SN-406 and SN-407 having IC$_{50}$ of 3.1 and 1.6 μM, respectively, and SN-401 IC$_{50}$ = 3.9 μM (Fig. 6f–i). Overexpressing a SWELL1 mutant in WT HEK cells wherein the endogenous SWELL1 R103 is replaced with an electronegative E103 (R103E-SWELL1) functioned as a partial dominant negative for I$_{Cl,SWELL}$ current, reducing I$_{Cl,SWELL}$ 84% (Fig. 6j). This residual current in R103E-SWELL1 expressing cells is less sensitive to both SN-401 and SN-407 inhibition, as demonstrated by a rightward shift in the IC$_{50}$ of SN-401 from 3.9 to 6.0 μM (Fig. 6k–m), and SN-407 from 1.6 to 4.2 μM (Fig. 6n–p). Molecular docking simulations also reveal that R103E mutant SWELL1 homomers are unable to dock SN-401 as predicted in WT homomers (Supplementary Fig. S7d i, ii), and as demonstrated in the published cryo-EM structure[31], unless at least 1 LRRC8 subunit contain an arginine at amino acid 103 (Supplementary Fig. S7d(iii)), further supporting the requirement of R103 for SN-401 binding determined experimentally (Fig. 6k–m). A similar docking result is obtained with SN-407 binding to R103E mutant SWELL1 homomers (Supplementary Fig. S7e), also consistent with experimental data (Fig. 6n–p).

To further test the predictions of our molecular docking simulations, we used cryo-EM to determine the structure of the SWELL1 homo-hexamer in lipid nanodiscs in the presence of SN-407 (Fig. 7, Supplementary Figs. S8–10, and Supplementary Table. S6). In initial maps using sixfold symmetry, SN-407 density was less apparent than for DCPIB/SN-401[31], potentially due to a reduction in SN-407 occupancy that is a consequence of lower compound solubility or the presence of multiple drug poses in different particles (Supplementary Fig. S9). Therefore, we

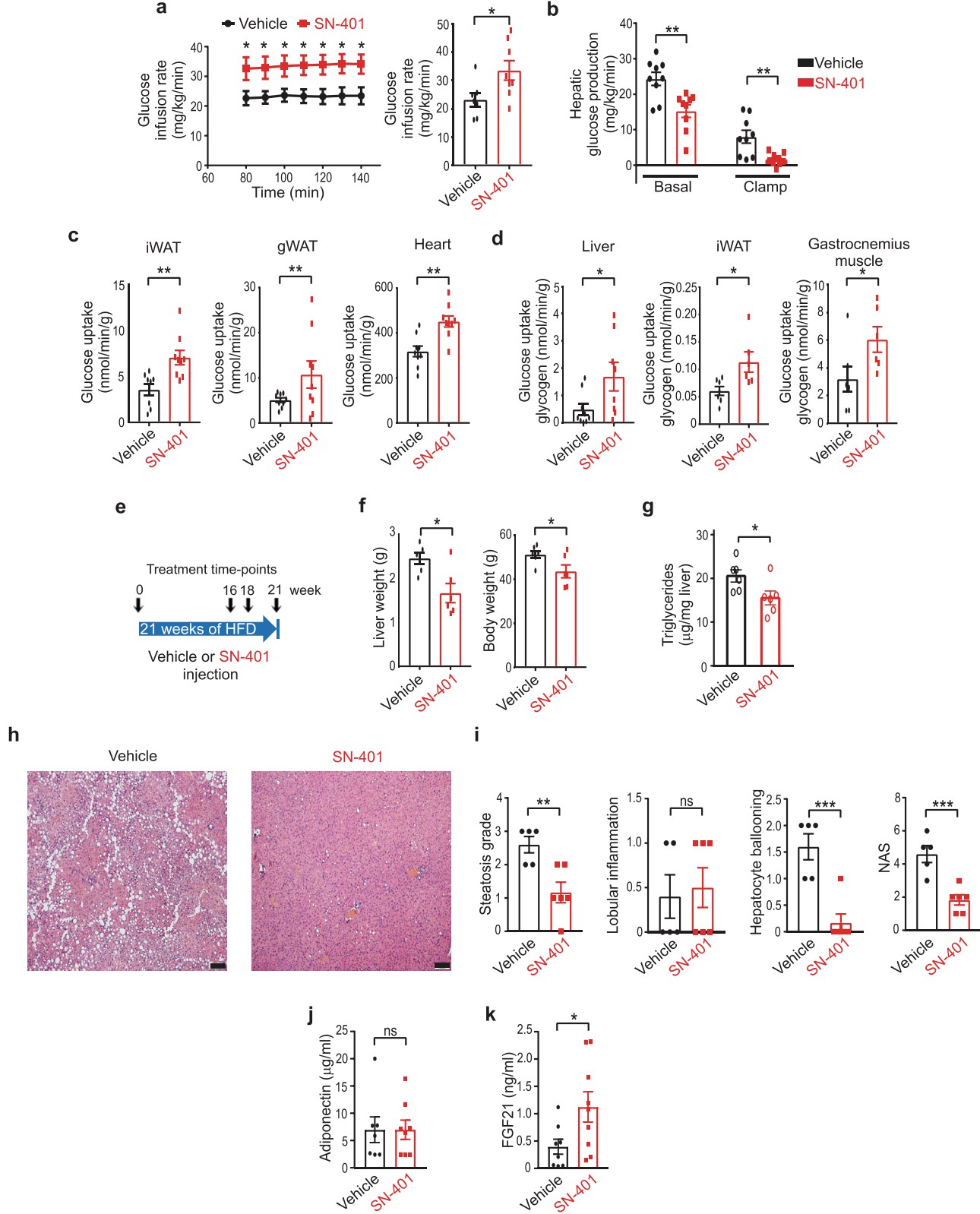

utilized sixfold symmetry expansion and symmetry relaxation[51] and were able to resolve two distinct poses for SN-407 with similar occupancy. Pose 1 shows the drug oriented vertically in the channel's selectivity filter in a manner that is similar to that observed for DCPIB/SN-401, but with the lengthened carboxylate chain coiling to maintain its interaction with R103 (Fig. 7a). In Pose-2, SN-407 is tilted off the SWELL1 central axis, positioning

its cyclopentyl group closer to the hydrophobic cleft between SWELL1 subunits (Fig. 7b). These data confirm lengthening the carboxylate chain in SN-407 preserves electrostatic interactions with R103 and enables additional contacts between the carboxylate chain and upper hydrophobic moieties in distinct binding poses, consistent with the results of our docking simulations. We note that while these models represent the best

**Fig. 4 SN-401 increases systemic insulin sensitivity, tissue glucose uptake, serum FGF21, and improves nonalcoholic fatty liver disease in murine T2D models. a** Mean glucose-infusion rate during euglycemic hyperinsulinemic clamps of T2D KKA$^y$ mice treated with vehicle ($n = 7$) or SN-401 ($n = 8$) for 4 days (5 mg/kg i.p). **b** Hepatic glucose production at baseline and during euglycemic hyperinsulinemic clamp of T2D KKA$^y$ mice treated with vehicle or SN-401 ($n = 9$ each) for 4 days (5 mg/kg i.p). **c** Glucose uptake determined from 2-deoxyglucose (2-DG) uptake in inguinal white adipose tissue (iWAT), gonadal white adipose tissue (gWAT) and heart during traced clamp of T2D KKA$^y$ mice treated with vehicle or SN-401 ($n = 9$ each) for 4 days (5 mg/kg i.p). **d** Glucose uptake into glycogen determined from 2-DG uptake in liver (vehicle, $n = 9$ and SN-401, $n = 8$), iWAT (vehicle, $n = 7$ and SN-401, $n = 6$) and gastrocnemius muscle (vehicle, $n = 7$ and SN-401, $n = 6$) during clamp of T2D KKA$^y$ mice treated with vehicle or SN-401 for 4 days (5 mg/kg i.p). **e** Schematic representation of treatment of C57BL/6 mice injected with vehicle or SN-401 ($n = 6$ each) during HFD-feeding. **f** Liver mass (left) and body weight (right) of HFD-T2D mice following treatment with vehicle or SN-401 (5 mg/kg i.p.) in **e. g–i** Liver triglycerides (**g**; $n = 6$ each), corresponding hematoxylin- and eosin-stained liver sections (**h**), histological scoring for steatosis, lobular inflammation, hepatocyte damage (ballooning), and NAFLD-activity score (NAS) (**i**) of liver sections shown in **h. j, k**. Plasma adiponectin (**j**) and FGF21 (**k**) levels in T2D KKA$^y$ mice treated with vehicle or SN-401 (5 mg/kg i.p daily × 5 days). Mean presented ±SEM. Two-tailed unpaired $t$-test was used in **a–d**, **f**, **g**, and **i–k**. Statistical significance is denoted by *, **, and *** representing $p < 0.05$, $p < 0.01$, and $p < 0.001$, respectively. ns no significance. Scale bar - 100 μm in **h**.

fits identified to the experimental data, the density is not sufficiently resolved (as a result of overall reconstruction resolution, alignment error, and multiple binding modes adopted by SN-407) to unambiguously define the binding pose from the cryo-EM data alone.

However, collectively, these molecular docking, cryo-EM, and functional experiments indicate that SN-401 and SWELL1-active congeners (SN-403/406/407) bind to SWELL1-LRRC8 hexamers at both R103 (via carboxylate end) and at the interface between adjacent LRRC8 monomers (via hydrophobic end), to potentially stabilize the closed state of the channel, and thereby inhibit I$_{Cl,SWELL}$ activity. Indeed, SN-401/SN-407 cryo-EM data and molecular docking simulations likely reflect the closed state of the SWELL1 homomer. This raises the possibility that SN-401 and active SN-40X compounds bind with higher affinity to SWELL1-LRRC8 complexes in the closed state most commonly encountered under native, physiological conditions, as compared to the open state. Moreover, we hypothesize these SN-40X compounds may function as molecular tethers to stabilize assembly of the SWELL1-LRRC8 hexamer by binding between R103 and adjacent LRRC8 monomers, potentially reducing SWELL1-LRRC8 complex disassembly, subsequent proteasomal degradation, and thereby augment translocation from ER to plasma membrane signaling domains, akin to pharmacological chaperones[52–56].

**SN-401 and SWELL1-active congener SN-406 inhibit I$_{Cl,SWELL}$ but promote SWELL1-dependent signaling at sub-micromolar concentrations**. To test this hypothesis, we compared the potency of SN-401/SN-406 to inhibit I$_{Cl,SWELL}$ when applied to closed as compared to open SWELL1-LRRC8 channels. SN-401/SN-406 concentrations of 6–10 μM are required to effectively inhibit channels when first opened by hypotonic activation, while concentrations of 1 μM are relatively ineffective (Fig. 6f, g, i)[32]. In contrast, application of 1 μM of SN-401 or SN-406 to SWELL1-LRRC8 channels in the closed state (i.e., for 30 min prior to hypotonic activation) markedly suppressed and delayed subsequent hypotonic activation of I$_{Cl,SWELL}$, compared to either vehicle alone, or Inactive compounds (Fig. 8a, b). SN-401 and SN-406 at concentrations as low as 250 nM had a similar effect (Fig. 8c, d). These data support the notion that SN-40X compounds bind with higher affinity to SWELL1-LRRC8 channels in the closed/resting state than the open/activated state, and stabilize the closed conformation at less than one-tenth the concentration required to inhibit activated SWELL1-LRRC8 channels.

Next, we applied SWELL1-active (SN-401 and SN-406), and Inactive (Inactive 1 and Inactive 2) compounds to differentiated 3T3-F442A adipocytes under basal culture conditions for 4 days and then measured SWELL1 protein after 6 h of serum starving. At both 10 and 1 μM, SN-401 and SN-406 markedly augmented SWELL1 protein to levels 1.5–2.1-fold greater than vehicle-

treated controls, while SWELL1-inactive congeners Inactive 1 and Inactive 2 did not significantly increase SWELL1 protein levels (Fig. 8e–h). SN-401 and SN-406 also enhanced plasma membrane (PM) localization of endogenous SWELL1 in preadipocytes compared to vehicle- or Inactive 1 (Fig. 8i, j), consistent with the increased endoplasmic reticulum (ER) to plasma membrane trafficking of SWELL1. Notably, SN-401 and SN-406 are capable of augmenting both SWELL1 protein and trafficking at concentrations as low as 1 μM (Fig. 8g–j and Supplementary Fig. S11), or an order of magnitude below the ~10 μM concentration required for inhibiting activated SWELL1-LRRC8 (upon hypotonic stimulation). These findings are consistent with the results of SN-401/SN-406 I$_{Cl,SWELL}$ inhibition when pre-applied to closed SWELL1-LRRC8 channels (Fig. 8a–d) and also with our observations that 500 nM SN-401 is sufficient to augment SWELL1-dependent insulin-AKT2-AS160 signaling in 3T3-F442A adipocytes (Fig. 2i, j). As we have recently shown that SWELL1 regulates AKT-eNOS signaling in endothelium and endothelium-targeted SWELL1 depletion contributes to vascular dysfunction in the setting of T2D[27], we next examined SN-401 mediated signaling in human umbilical vein endothelial cells (HUVECs). SN-401 applied at 500 nM to HUVECs increases eNOS phosphorylation (Ser1177), a downstream AKT target, and this effect is abrogated by small interfering/short hairpin mediated *SWELL1* knockdown in HUVECs, supporting a SWELL1-dependent mechanism (Supplementary Fig. S12a–d). Furthermore, SN-406 applied to HUVECs at 100 nM was sufficient to induce SWELL1 1.5-fold, basal pAKT2 2.8-fold, and downstream p-eNOS 5.5-fold as compared to vehicle (Fig. 8k, l), while Inactive 1 had no effect at 500 nM (Fig. 8m, n). Indeed, SN-401 exhibits dose-dependent induction of p-eNOS in HUVECs that saturates beyond 100 nM (Fig. 8o, p).

**SWELL1-active compounds prevent reductions in SWELL1 protein and rescue SWELL1-dependent islet insulin secretion under glucolipotoxic conditions**. We next asked whether endoplasmic reticulum (ER) stress associated with glucolipotoxicity in metabolic syndrome may promote SWELL1 protein degradation, and thereby reduce I$_{Cl,SWELL}$ and SWELL1 protein in T2D (Fig. 1). In this context, we hypothesized that SN-401 and SN-406 might assist with SWELL1-LRRC8 assembly and rescue SWELL1-LRRC8 from degradation. To test this concept in vitro, we first treated 3T3-F442A adipocytes with either vehicle, SN-401, SN-406, or Inactive 2, and then subjected these cells to 1 mM palmitate + 25 mM glucose to induce glucolipotoxic stress (Fig. 9a, b). We found that SWELL1 protein was reduced by 50% upon palmitate/glucose treatment, consistent with ER stress-mediated SWELL1 degradation, and this reduction was entirely prevented by both SWELL1-active SN-401 and SN-406, but not by Inactive 2 (Fig. 9a, b). These data were consistent with the

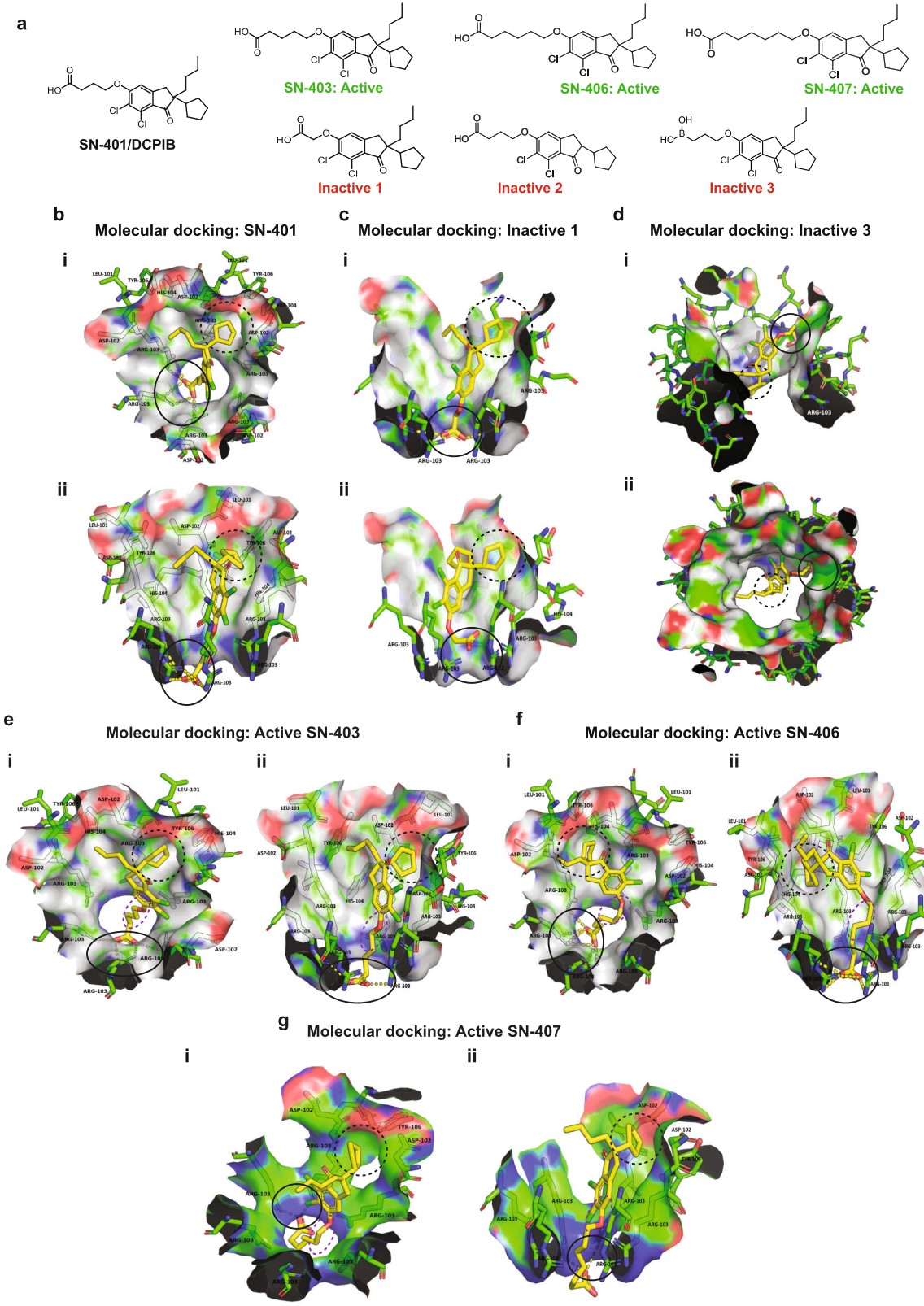

notion that SN-401 and SWELL1-active congeners are functioning to stabilize SWELL1-LRRC8 assembly and signaling under glucolipotoxic conditions associated with T2D and metabolic syndrome.

To examine whether these protective effects of SN-401 under glucolipotoxic conditions is also capable of rescuing islet insulin secretion and whether this effect occurs via target engagement, we measured dynamic glucose-stimulated insulin secretion (GSIS) by perifusion in human and murine islets ± palmitate ± SN-401 and ±SWELL1 as outlined in Fig. 9c. In both human and murine islets, we found that 16 h of 1 mM palmitate treatment reduces GSIS compared to baseline (Fig. 9d–g). However, when islets are treated with SN-401 (10 µM) for 4 days prior to palmitate treatment, then maintained during palmitate treatment, and

**Fig. 5 Molecular docking of SN-401 and synthesized congeners to SWELL1 reveal specific drug-target binding interactions. a** Chemical structures of SN-401/DCPIB, SN-403, SN-406, SN-407, Inactive 1, Inactive 2, and Inactive 3. **b** top (i) and side (ii) view of binding poses of SN-401; SN-401 carboxylate groups interact with R103 residue guanidine groups (solid black circle), the SN-401 cyclopentyl group occupies a shallow hydrophobic cleft at the interface of two monomers formed by SWELL1 D102 and L101 (black broken circle)#. **c** Binding poses for Inactive 1; (i) side view of the first binding pose of Inactive 1 showing potential electrostatic interaction with R103 (solid black circle) but unable to reach into and occupy the hydrophobic cleft (black broken circle); (ii) side view of the second pose for Inactive 1 with the cyclopentyl group occupying the hydrophobic cleft (black broken circle) but the carboxylate group unable to reach and interact with R103 (solid black circle). **d** Binding pose for Inactive 3; (i) side and (ii) top view of Inactive 3 on the top of the pore not able to occupy the hydrophobic cleft (black broken circle) or electrostatically interact with R103 (solid black circle). **e**–**g** (i) top and (ii) side view of binding poses of SN-403 (**e**), SN-406 (**f**), and SN-407 (**g**); the carboxylate groups interact with guanidine group of R103 residues (solid black circle), the cyclopentyl group occupies a shallow hydrophobic cleft at the interface of two monomers formed by D102 and L101 (black broken circle) and the alkyl side chain SN-403/SN-406/SN-407 interacts with the alkyl side chain of R103 (purple broken circle). #Poses generated for respective compounds by docking into PDB 6NZZ using Molecular Operating Environment 2016 (MOE) software package. SN-401 congeners are depicted as yellow sticks and R103, D102, and L101 are depicted as green sticks.

subsequently, SN-401/palmitate washed off during GSIS, insulin secretion is normalized (Fig. 9d–g and Supplementary Table. S2). Importantly, this SN-401 mediated GSIS normalization under glucolipotoxic conditions is SWELL1-dependent, since this rescue is completely abrogated in SWELL1 KD human islets (Fig. 9d, e) and β-cell-targeted SWELL1 KO murine islets (Fig. 9f, g).

**SWELL1-active SN-401 congeners improve glycemic control in murine T2D.** To determine if the effects of SN-401 observed in vivo in T2D mice are attributable to SWELL1-LRRC8 binding, as opposed to off-target effects, we next measured fasting blood glucose and glucose tolerance in HFD-T2D mice treated with either SWELL1-active SN-403 or SN-406 as compared to SWELL1-inactive Inactive 1 (all at 5 mg/kg/day × 4 days). In mice treated with HFD for 8 weeks, SN-403 significantly reduced fasting blood glucose and improved glucose tolerance compared to Inactive 1 (Fig. 10a). In cohorts of mice raised on HFD for 12–18 weeks, with more severe obesity-induced T2D, SN-406 also markedly reduced fasting blood glucose and improved glucose tolerance (Fig. 10b). Similarly, in a separate experiment, SN-406 significantly improved glucose tolerance in HFD-T2D mice, compared to Inactive 1 (Fig. 10c), and this is associated with a trend toward improved insulin sensitivity based on the Homeostatic Model Assessment of Insulin Resistance (HOMA-IR)[57] (Fig. 10d), and significantly augmented insulin secretion in perifusion GSIS (Fig. 10e). Finally, based on the GTT AUC, SN-407 also improved glucose tolerance in T2D KKA$^y$ mice, compared to Inactive 1 (Fig. 10f) and increased GSIS (Fig. 10g). These data reveal the in vivo anti-hyperglycemic action of SN-401 and its bioactive congeners require SWELL1-LRRC8 binding and thus support the notion of SWELL1-on-target activity in vivo.

An important feature of the hypothesized mechanism of action of SN-40X is that these active compounds bind to SWELL1-LRRC8 channel complexes in vivo in the ~100–500 nM range to augment SWELL1-dependent signaling (Figs. 2i, j, 8k–p and Supplementary Fig. S12a–d) without achieving the serum concentrations necessary for open channel SWELL1 current inhibition (~5–10 uM)[32], followed by unbinding. As such, SWELL1 function may be rescued without significant SWELL1-LRRC8 or VRAC inhibition. Consistent with this hypothesis, in vivo pharmacokinetics (PK) of SN-401 and SN-406 in mice following i.p. or p.o. administration of 5 mg/kg of SN-401 or SN-406 reveals plasma concentrations that either transiently approach (Supplementary Fig. S13a, i.p. dosing), or remain well below $I_{Cl,SWELL}$ inhibitory concentrations (Supplementary Fig. S13b, p.o. dosing) while exceeding concentrations sufficient for induction of SWELL1 signaling activity (>~100 nM) for 8–12 h. Importantly, SN-401 PK in a target tissue, adipose, also closely tracks serum concentrations via both i.p. and p.o. administration (Supplementary Fig. S13c). Finally, these in vivo

PK studies demonstrate that SN-401 has high oral bioavailability ($AUC_{p.o.}/AUC_{i.v.}$ = 79%, Supplementary Table S7), and when administered via oral gavage to HFD-fed T2D C57 mice at 5 mg/kg/day SN-401 fully retains in vivo therapeutic efficacy (Supplementary Fig. S13d). Collectively, these PK data reveal that appropriate SN-401 concentrations are attained to achieve the observed therapeutic effect while remaining insufficient to inhibit activated VRAC.

**Discussion**
Our current working model is that the transition from compensated obesity (pre-diabetes, normoglycemia) to decompensated obesity (T2D, hyperglycemia) reflects, among other things, a relative reduction in SWELL1-dependent signaling in peripheral insulin-sensitive tissues[30,33] and in pancreatic β-cells[58,59]—metabolically phenocopying SWELL1-loss-of-function models[24,26–30]. This contributes to the combined insulin resistance and impaired insulin secretion associated with poorly-controlled T2D and hyperglycemia. SWELL1 forms a macro-molecular signaling complex that includes hetero-hexamers of SWELL1 and LRRC8b-e[22,23], with stoichiometries that likely vary from tissue to tissue. We propose SWELL1-LRRC8 signaling complexes are inherently unstable, and thus a proportion of complexes succumb to disassembly and degradation. Glucolipo-toxicity and ensuing ER stress associated with T2D states[60–62] provide an unfavorable environment for SWELL1-LRRC8 complex assembly, contributing to SWELL1 degradation and reductions in SWELL1 protein and SWELL1-mediated $I_{Cl,SWELL}$ observed in T2D in some tissues. We speculate that small molecules SN-401 and active congeners (SN-40X) serve as pharmacological chaperones[52] to stabilize the formation of the SWELL1-LRRC8 complex. This reduces SWELL1 degradation and enhances the passage of SWELL1-LRRC8 heteromers through the ER and Golgi apparatus to the plasma membrane—thereby rectifying the SWELL1-deficient state in multiple meta-bolically important tissues in the setting of metabolic syndrome to improve glycemic control via both insulin sensitization[24,25,30] and secretion[28,29] mechanisms. Indeed, the concept of small molecule inhibitors acting as therapeutic molecular chaperones[52] to support the folding, assembly, and trafficking of proteins (including ion channels) has been demonstrated for Niemann-Pick C disease[53,54] and congenital hyperinsulinism (SUR1-K$_{ATP}$ channel mutants)[63–65]. Similarly, in the case of congenital hyperinsulinism, the SUR1-K$_{ATP}$ chemical chaperones are also themselves, paradoxically, K$_{ATP}$ channel inhibitors[63–65]. Also, this therapeutic mechanism is analogous to small molecule cor-rectors for another chloride channel, CFTR (VX-659/VX-445, Vertex Pharmaceuticals)[55,56], which is proving to be a break-through therapeutic approach[66,67] for cystic fibrosis. In vivo, we hypothesize SN-40X compounds bind to SWELL1-LRRC8

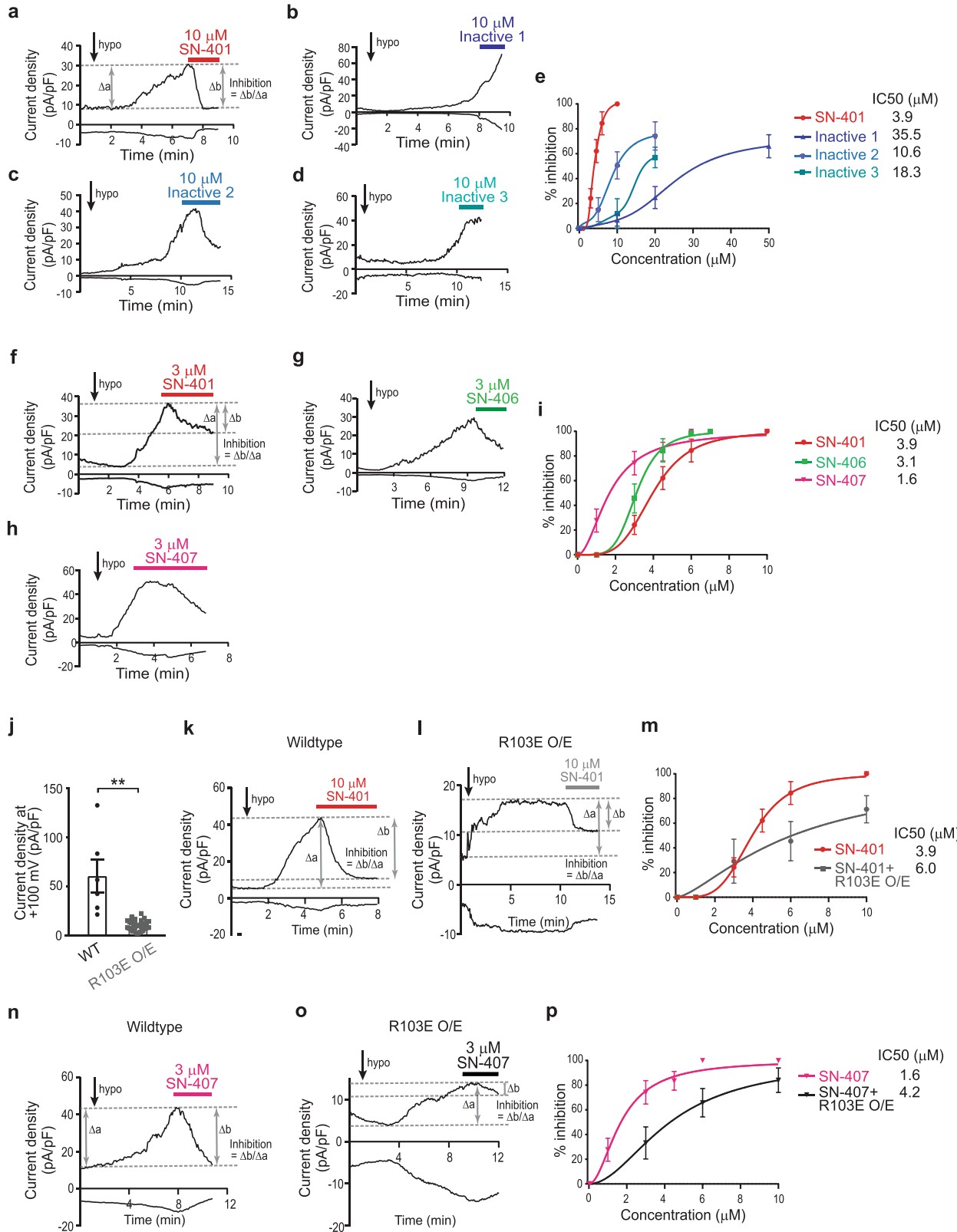

complexes in the closed state, within the concentration range between $C_{max}$ and ~100 nM. This shifts the balance toward maintaining stable SWELL1-LRRC8 complexes to preserve normal levels and localization (trafficking) within the T2D glucolipotoxic milieu. SN-40X may then unbind from the SWELL1-LRRC8 complex, thereby restoring insulin signaling in target tissues, and permitting SWELL1-mediated β-cell insulin

secretion. This seemingly paradoxical mechanism may rely on the *phasic* SN-40X concentrations observed in vivo (see in vivo PK data) to allow for SN-401 binding, resultant chaperone-mediated rescue, followed by unbinding; as opposed to tonic SN-40X-SWELL1 binding. Another prediction of this model is that lower-affinity SN-40X compounds may be preferable to very high-affinity congeners, to provide the appropriate pharmacodynamics

**Fig. 6 SN-401 congener I$_{CI,SWELL}$ inhibitory activity support predicted structural requirements for on-target activity. a–d** I$_{CI,SWELL}$ inward/outward current over time upon hypotonic (210 mOsm) stimulation and subsequent inhibition with 10 μM of SN-401 applied at 30 pA/pF outward I$_{CI,SWELL}$ current density (**a**), Inactive 1 (**b**), Inactive 2 (**c**), and Inactive 3 (**d**) in HEK-293 cells. **e** Percentage inhibition of outward current blocked by SN-401/Inactive 1/ Inactive 2/Inactive 3 at varying concentrations ($n = 4–10$ cells) in HEK-293 cells and the corresponding IC$_{50}$ values. **f–h** I$_{CI,SWELL}$ inward/outward current over time upon hypotonic (210 mOsm) stimulation and subsequent inhibition with 3 μM SN-401 applied at 30 pA/pF outward I$_{CI,SWELL}$ current density (**f**), SN-406 (**g**), and SN-407 (**h**) in HEK-293 cells. **i** Percentage inhibition of outward current blocked by SN-401/SN-406/SN-407 at varying concentrations ($n = 4–10$ cells) in HEK-293 cells and the corresponding IC$_{50}$ values. **j** Mean outward I$_{CI,SWELL}$ current densities at +100 mV from HEK-293 (wildtype, $n = 6$) and R103E overexpression in HEK-293 (R103E O/E, $n = 70$) cells. **k, l** I$_{CI,SWELL}$ inward/outward current over time upon hypotonic (210 mOsm) stimulation and subsequent inhibition with 10 μM of SN-401 applied at 30 pA/pF outward I$_{CI,SWELL}$ current density in HEK-293 (wildtype, **k**) and at maximum current density in R103E expressing HEK-293 (R103E O/E, **l**) cells. **m** Percentage inhibition of outward current blocked by SN-401 at varying concentrations in wild-type and R103E O/E HEK-293 cells and the corresponding IC$_{50}$ values ($n = 4–10$ cells). **n, o** I$_{CI,SWELL}$ inward/outward current over time upon hypotonic (210 mOsm) stimulation and subsequent inhibition with 3 μM of SN-407 at 30 pA/pF outward I$_{CI,SWELL}$ current density in HEK-293 (wildtype, **n**) and at maximum current density in R103E expressing HEK-293 (R103E O/E, **o**) cells. **p** Percentage inhibition of outward current blocked by SN-407 at different concentrations in wildtype and R103E O/E HEK-293 cells and the corresponding IC$_{50}$ values ($n = 4–10$ cells). Data were presented as mean ± SEM. A two-tailed unpaired $t$-test was used in **j**.

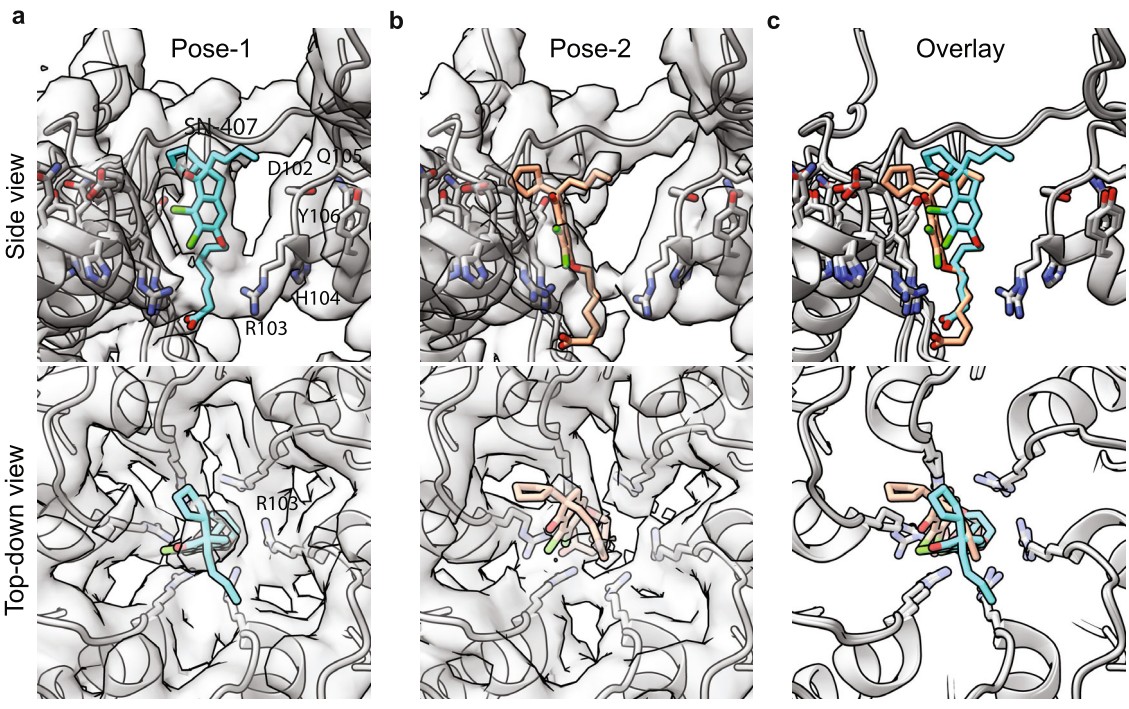

**Fig. 7 Cryo-EM of SN-407-SWELL1 complex. a–c** Cryo-EM images revealing: Pose-1 (**a**), Pose-2 (**b**), and overlay of Pose 1 and 2 (**c**) views of selectivity filter with SN-407 bound from the membrane plane (side view) and top-down view. The atomic model is represented as ribbons and sticks within the cryo-EM density with three subunits removed in the side view for clarity. Cryo-EM density is represented in transparent gray, nitrogens are colored blue, oxygens red, chlorines green, protein carbons gray, and SN-407 carbons teal (Pose1) or orange (Pose-2).

required for unbinding and optimal therapeutic efficacy. Indeed, this mechanism is reminiscent of the paradoxical use of insulin secretagogue sulfonylurea receptor inhibitors as pharmacological chaperones to rescue K$_{ATP}$ mutants in congenital hyperinsulinism by binding (and inhibiting) these mutant K$_{ATP}$ channels[63–65], and then unbinding, thereby favoring lower-affinity inhibitors: tolbutamide and carbamazepine, over glibenclamide.

Through structure-activity relationship (SAR), in silico molecular docking studies, and cryo-EM studies, we identified hotspots on opposing ends of the SN-401 molecule that interact with separate regions of the SWELL1-LRRC8 complex: the carboxylate group with R103 in multiple LRRC8 subunits at a constriction in the pore, and the cyclopentyl group within the hydrophobic cleft formed by adjacent LRRC8 monomers; functioning like a molecular staple or tether to bind loosely associated SWELL1-LRRC8 monomers (especially in the setting of T2D) into a more stable hexameric structure. Indeed, the cryo-EM structure obtained in lipid nanodiscs of SN-401[31] and derivative SN-407 supports

hypothesized binding models of SN-40X with SWELL1 homomer.

Another advantage provided by SAR studies was the identification and synthesis of SN-401 congeners that removed (Inactive 1,2 and 3) or enhanced (SN-403/406/407) SWELL1 binding, as these provided powerful tools to query SWELL1-on-target activity directly in vitro and in vivo, and also validated the proof-of-concept for developing SN-401 congeners with enhanced efficacy. Indeed, this approach was necessary to prove SWELL1-LRRC8 on-target activity of the SN-40X series in vivo, because SWELL1-LRRC8 is expressed broadly in numerous insulin-sensitive tissues and in islet cells. As the global SWELL1/ LRRC8a KO mouse is essentially embryonically lethal[20] testing SN-40X compounds in global SWELL1$^{-/-}$ mice is not possible, and generating multi-tissue (adipose, liver, skeletal muscle, and β-cell) SWELL1 KO mice is outside the scope of the current study. Therefore, using the SAR to generate SWELL1-LRRC8 inactive compounds (Inactive 1–3) as a negative control provided an

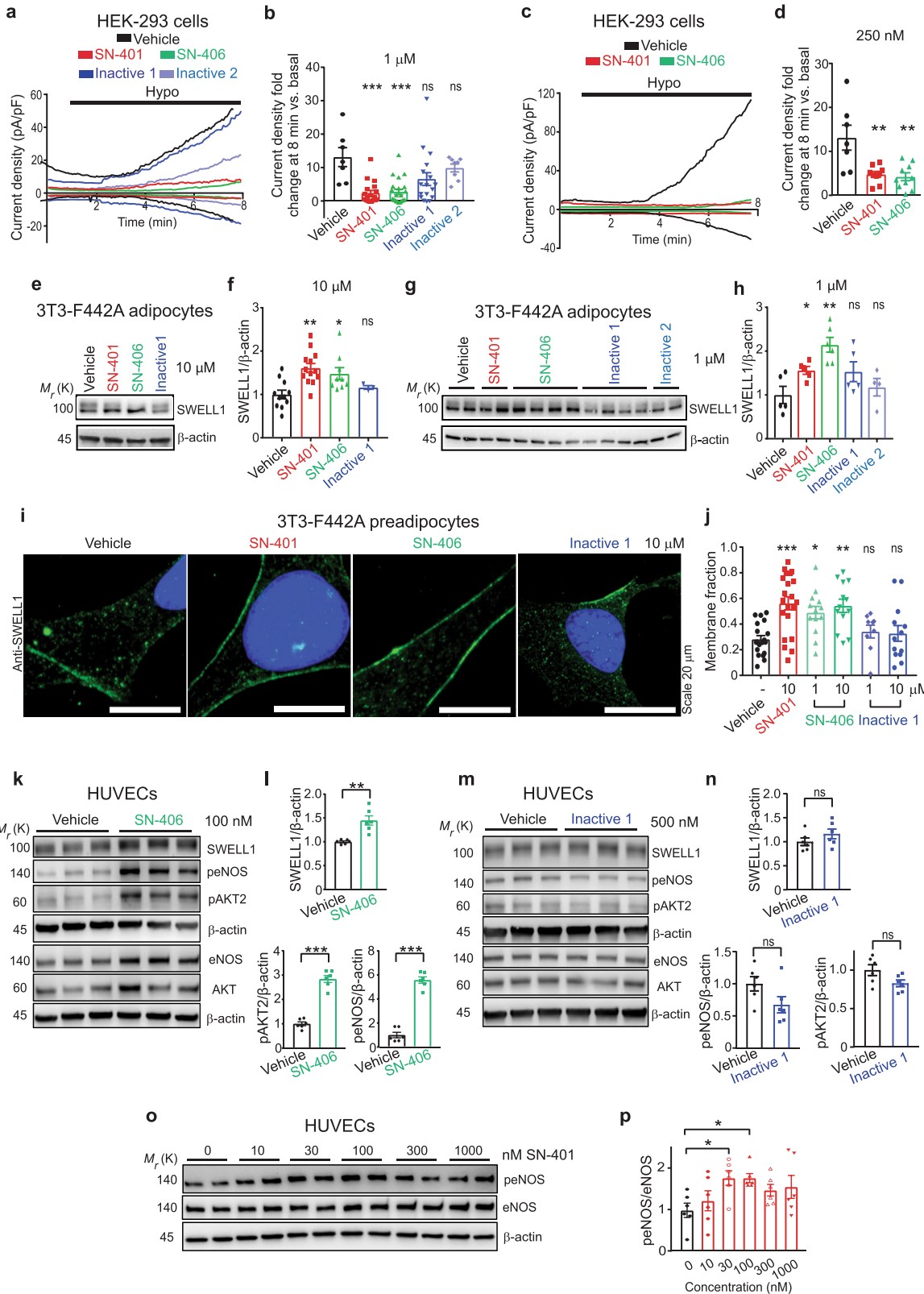

alternative approach to prove in vivo on-target activity on a broadly expressed signaling molecule. In addition to this medicinal chemistry approach to test on-target activity in vitro and in vivo, we found that SN-40X mediated induction of AKT-AS160 and AKT-eNOS signaling requires SWELL1 in cultured adipocytes, hepatocytes, and HUVECs. Moreover, SN-401 mediated rescue of islet insulin secretion under glucolipotoxic conditions in vitro also requires SWELL1. Finally, it is important to note that the studies demonstrating promiscuity of SN-401/DCPIB with other ion channel targets all applied DCPIB at ~10–200 μM[68–73]. This is 100–200-fold higher than the concentrations required to potentiate SWELL1-dependent signaling in vitro (Figs. 2i, j, 8k–p and Supplementary Fig. S12a–d), and similarly higher than SN-401 and SN-406 concentrations

**Fig. 8 SN-401 and SWELL1-active congener SN-406 inhibit $I_{Cl,SWELL}$ and promote SWELL1-dependent signaling at sub-micromolar concentrations. a–d** Representative $I_{Cl,SWELL}$ inward/outward current traces over time in HEK-293 cells preincubated for 30 min with vehicle, SN-401, SN-406, Inactive 1, or Inactive 2 at 1 µM (**a**) and vehicle, SN-401, and SN-406 at 250 nM (**c**), and subsequent hypotonic stimulation with the compound. Fold changes in mean outward $I_{Cl,SWELL}$ current density at +100 mV for 7 min timepoint after hypotonic stimulation are shown in **b**, **d**. **e**, **f** Representative western blots for SWELL1 and β-actin (**e**) and densitometry (**f**) from 3T3-F442A adipocytes treated with vehicle ($n = 8$), SN-401 ($n = 10$), SN-406 ($n = 6$), or Inactive 1 ($n = 3$) at 10 µM for 96 h. **g**, **h** Representative western blots for SWELL1 and β-actin (**g**) and densitometry (**h**) from 3T3-F442A adipocytes treated with vehicle ($n = 5$), SN-401 ($n = 5$), SN-406 ($n = 6$), Inactive 1 ($n = 5$), or Inactive 2 ($n = 4$) at 1 µM for 96 h. **i**, **j** Representative immunostaining images demonstrating endogenous SWELL1 localization in 3T3-F442A preadipocytes treated with vehicle or SN-401/SN-406/Inactive 1 at 10 µM for 48 h (Scale – 20 µm) (**i**) and mean SWELL1 membrane- versus cytoplasm-localized fraction from vehicle ($n = 19$), SN-401 ($n = 21$), SN-406 ($n = 13$ for 1 µM and 10 µM), or Inactive 1 ($n = 9$ for 1 µM and $n = 13$ for 10 µM) treated cells (**j**). **k**, **l** Representative western blots for SWELL1, p-eNOS$^{Ser1177}$, eNOS, pAKT2$^{Ser474}$, AKT2, and β-actin in HUVEC cells treated with vehicle or 100 nM SN-406 for 96 h (**k**) and densitometry (**l**, $n = 6$). **m**, **n** Representative western blots for SWELL1, p-eNOS$^{Ser1177}$, eNOS, pAKT2$^{Ser474}$, AKT2, and β-actin in HUVEC cells treated with vehicle or 500 nM Inactive 1 for 96 h (**m**) and densitometry (**n**, $n = 6$). **o**, **p** Representative western blots for p-eNOS$^{Ser1177}$, eNOS, and β-actin in HUVEC cells treated with vehicle or different concentrations of SN-401 for 96 h (**o**, $n = 6$) and densitometry (**p**). Data were represented as mean ± SEM. Two-tailed unpaired $t$-test was used in **f** and **h** (compared to vehicle), **l**, **n**. One-way ANOVA was used for **b**, **d**, **j**, and **p**. *, **, and *** represents $p < 0.05$, $p < 0.01$, and $p < 0.001$. ns not significant.

predominantly attained in vivo (Supplementary Fig. S13a–c) to achieve a therapeutic effect (Figs. 3, 4, 10). Accordingly, these studies are not applicable with respect to putative off-target mechanisms for the therapeutic effects observed from SN-40X compounds.

SWELL1-LRRC8 complexes are broadly expressed in multiple tissues, and consist of unknown combinations of SWELL1, LRRC8b, LRRC8c, LRRC8d, and LRRC8e, indicating SWELL1 complexes may be enormously heterogeneous. However, SWELL1-LRRC8 stabilizers like SN-401 may be designed to target many, if not all, possible channel complexes since all will contain the elements necessary for SN-401 binding: at least one R103 (from the requisite SWELL1 monomer: carboxyl group binding site), and the nature of the hydrophobic cleft (cyclopentyl binding site), which is conserved among all LRRC8 monomers. Indeed, traced glucose clamps did reveal insulin sensitization effects in vivo in multiple tissues, including adipose, skeletal muscle, liver, and heart, and this data was supported by additional signaling studies in vitro in cultured adipocytes, HUVECs, primary hepatocytes, and isolated islets, as well as in adipose tissue and skeletal muscle. The increased glucose uptake in the heart is particularly interesting, since this may provide salutary effects on cardiac energetics that could favorably impact both systolic (HFrEF) and diastolic (HFpEF) function in diabetic cardiomyopathy, and thereby potentially improve cardiac outcomes in T2D, as observed with SGLT2 inhibitors[74–79]. Moreover, the finding that SN-401 robustly increases serum FGF21 levels may provide an additional secondary metabolic molecular mechanism for the observed improvements in glucose metabolism, and hepatic steatosis observed in these SN-401 treated T2D models and warrants future investigation.

The current study provides an initial proof-of-concept for pharmacological induction of SWELL1 signaling using SWELL1 modulators (SN-40X) to treat metabolic diseases at multiple homeostatic nodes, including adipose, skeletal muscle, liver, and pancreatic β-cell, whereby SN-40X compounds function to restore both insulin sensitivity and insulin secretion. Hence, SN-401 may represent a tool compound from which a novel drug class may be derived to treat T2D, NASH, and other metabolic diseases.

## Methods
**Patients.** The University of Iowa Institutional Review Board (approval number 201103721) and the Human Research Protection Office at Washington University School of Medicine in St. Louis, MO (approval number 201808128) approved the studies involving human adipose samples. Written, informed consent was obtained from all participants before participating in this study. Information on participant's sex, age, body mass index (BMI), HbA1c, and random or fasting plasma glucose concentration was obtained either during an initial screening visit or medical

records. Visceral adipose tissue was obtained from esophageal or omental adipose tissue depots in participants with obesity undergoing bariatric surgeries and in normal-weight, control subjects scheduled for non-bariatric surgeries. Fat tissue was kept in ice-cold PBS and transferred to the laboratory within 20 min for patch-clamp experiments, or snap-frozen in liquid nitrogen to assess adipose tissue SWELL1 protein expression.

**Animals.** All experimental procedures involving mice were approved by the Institutional Animal Care and Use Committee of the University of Iowa and Washington University at St. Louis and performed in accordance with ethical regulations. All C57BL/6 mice involved in this study were purchased from Charles River Labs or Taconic Biosciences. Both KK.Cg-Ay/J (KKA$^y$) and KK.Cg-Aa/J (KKA$^a$) mice involved in the study were obtained from Jackson Labs (Stock No: 002468). Liver-specific-SWELL1 knockout mouse was generated by crossing Albumin-Cre (Jax stock No: 003574) with the SWELL1$^{fl/fl}$ mouse[24]. Mice were gender/age-matched and bred up for experiments. All mice were fed ad libitum with either regular chow (RC; NIH31 irradiated, #7913 or Lab diet, Picolab@ Rodent Diet 20, #5053) or a high-fat diet (Research Diets, Inc., 60 kcal% fat) with free access to water and housed in a 12 h light/dark cycle-, temperature-, and humidity- controlled room. For high-fat diet (HFD) studies, only male mice were used and were started on an HFD regimen at the age of 6–9 weeks. For all experiments involving KKA$^y$ and KKA$^a$ mice, both males and females were used at ~50/50 ratio. In all experiments, investigators were kept blinded for different treatment groups in mice both during the experiment and subsequent analysis.

**Small molecule treatment.** All compounds were dissolved in Kolliphor® EL (Sigma, #C5135). Either vehicle (Kolliphor® EL), SN-401 (DCPIB, 5 mg/kg of body weight/day, Tocris, D1540), SN-403, SN-406, SN-407, or Inactive 1 were administered i.p. as indicated using 1cc syringe/26 G × 1/2 inch needle daily for 4–10 days. In one experiment, SN-401 was administered daily for 8 weeks. SN-401, formulated as above, was also administered by oral gavage at 5 mg/kg/day for 5 days using a 20 G × 1.5 inch reusable metal gavage needle.

**Adenovirus.** Human adenoviruses type 5 with hLRRC8A/SWELL1-shRNA (Ad5-mCherry-U6-hLRRC8A/SWELL1-shRNA, $2.2 \times 10^{10}$ PFU/ml), a scrambled non-targeting control (Ad5-U6-scramble-mCherry, $1 \times 10^{10}$ PFU/ml), Ad5-CAG-LoxP-stop-LoxP-3XFlag-SWELL1 ($1 \times 10^{10}$ PFU/ml), β-cell-targeted adenovirus type 5 with Ad5-RIP2-GFP ($4.1 \times 10^{10}$ PFU/ml), GCaMP6s (Ad5-RIP1-GCaMP6s, $4.9 \times 10^{10}$ PFU/ml), and GCaMP6s-2A-iCre (Ad5-GCaMP6s-RFP-2A-Cre, $5.8 \times 10^{10}$ PFU/ml) were obtained from Vector Biolabs. Adenovirus type 5 with Ad5-CMV-Cre-eGFP ($8 \times 10^{10}$ PFU/ml) and Ad5-CMV-Cre-mCherry ($3 \times 10^{10}$ PFU/ml) were obtained from the University of Iowa Viral Vector Core.

**Cell culture.** 3T3-F442A preadipocytes (Sigma, #00070654) were maintained in 90% DMEM (25 mM D-Glucose and 4 mM L-Glutamine) containing 10% fetal bovine serum (FBS) and 100 IU penicillin and 100 µg/ml streptomycin on collagen-coated (rat tail type-I collagen, Corning) plates at 37 °C and 5% CO$_2$. Upon reaching confluency, the cells were differentiated in the above-mentioned media supplemented with 5 µg/ml insulin (Cell Applications) and replenished every other day with the differentiation media. For insulin signaling studies on WT and KO adipocytes with or without SWELL1 overexpression (O/E), the cells were differentiated for 10 days and transduced with Ad5-CAG-LoxP-stop-LoxP-SWELL1-3XFlag virus (MOI 12) on day 11 in 2% FBS containing differentiation medium. To induce the overexpression, Ad5-CMV-Cre-eGFP (or mcherry) (MOI 12) was added on day 13 in 2% FBS containing a differentiation medium. The cells were then switched to 10% FBS containing differentiation medium from day 15 to 17. On day 18, the cells were

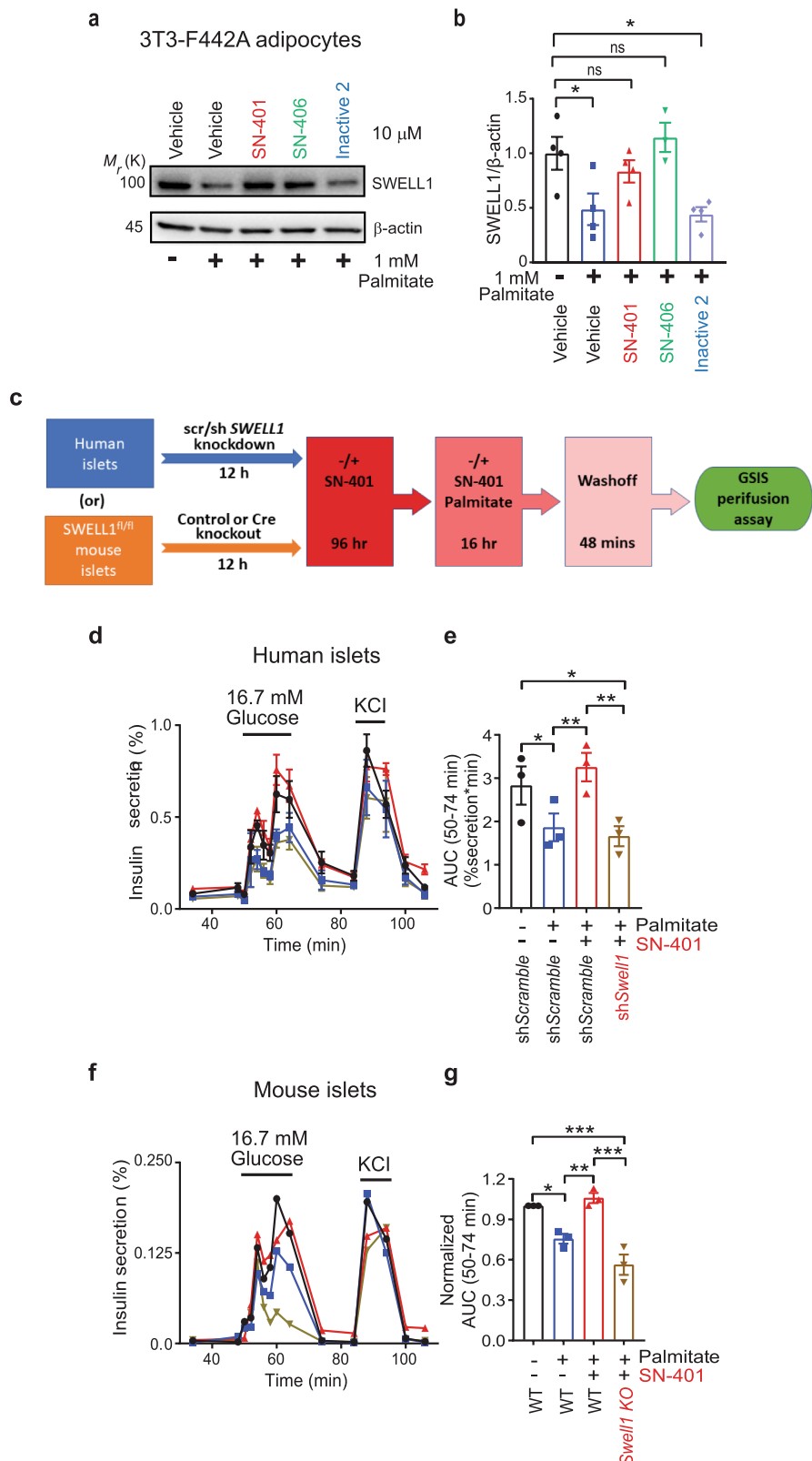

starved in serum-free media for 6 h and stimulated with 0 and 10 nM insulin for 15 min. Either Ad5-CAG-LoxP-stop-LoxP-SWELL1-3XFlag or Ad5-CMV-Cre-eGFP (or mcherry) virus transduced cells alone were used as controls. Based on GFP/mcherry fluorescence, viral transduction efficiency was ~90%.

For SN-401 treatment and insulin signaling studies in 3T3-F442A preadipocytes, the cells were incubated with either vehicle (DMSO) or 10 μM SN-401 for 96 h. The cells were serum-starved for 6 h with vehicle (DMSO) or SN-401 and washed with PBS three times and stimulated with 0, 3, and 10 nM insulin-containing media for 15 min

prior to collecting lysates. In the case of 3T3-F442A adipocytes, the WT and KO cells were treated with either vehicle (DMSO), 1 or 10 μM SN-40X (after 7–11 days of differentiation) for 96 h and then stimulated with 0 and 10 nM insulin/serum-containing media with vehicle (DMSO) or SN-40X for 15 min for SWELL1 detection. For AKT and AS160 signaling, the WT and KO cells were treated with either vehicle (DMSO) or 500 nM SN-401 for 96 h and serum-starved in the presence of vehicle or SN-401 (500 nM) for 6 h. The cells were washed twice in hypotonic buffer (240 mOsm; 90 mM NaCl, 5 mM NaHCO$_3$, 4.8 mM KCl, 1.2 mM KH$_2$PO$_4$, 2.5 mM CaCl$_2$, 2.4 mM

**Fig. 9 SWELL1-active compounds prevent reductions in SWELL1 protein and rescue SWELL1-dependent islet insulin secretion under glucolipotoxic conditions. a, b** Representative western blots detecting SWELL1 and β-actin in 3T3-F442A adipocytes pretreated with vehicle, SN-401, SN-406, or Inactive 2 (10 μM) for 96 h and subsequently treated with ±palmitate in the absence or presence of compounds for 16 h (**a**, n = 3–4 in each condition) and corresponding densitometric ratio for SWELL1/β-actin (**b**). **c** Schematic for glucose-stimulated insulin secretion (GSIS) perfusion assay in human and mouse islets. **d, e** GSIS perfusion assay of islets obtained from cadaveric human islets transduced with either adenoviral short hairpin control (shScramble) or SWELL1 (shSWELL1) for 12 h, then pretreated with either vehicle or SN-401 (10 μM) for 96 h and subsequently treated with ±palmitate or palmitate with ±SN-401 (n = 3 each) for 16 h (**d**) and the corresponding area under the curve (AUC) (**e**). **f, g** GSIS perfusion assay of WT and β-cell SWELL1 KO islets obtained by isolation of islets from SWELL1[fl/fl] mice and transduced either with adenoviral control (WT) or Cre-recombinase (SWELL1 KO) for 12 h, then pretreated with either vehicle or SN-401 (10 μM) for 96 h and subsequently treated with ±palmitate or palmitate with ±SN-401 (n = 3 each) for 16 h (**f**) and the corresponding area under the curve (AUC) (**g**). Data were represented as mean ± SEM. One-way ANOVA was used for **b**, **e**, and **g**. *, **, and *** represents p < 0.05, p < 0.01, and p < 0.001 respectively and "ns" indicates the difference was not significant.

MgSO₄, 10 mM HEPES, and 25 mM Glucose, pH 7.4) and then incubated at 37 °C in a hypotonic buffer for 10 min followed by a serum-free media wash and subsequent stimulation with insulin/serum-containing media for 30 min at 37 °C without SN-401 (or vehicle). To simulate glucolipotoxicity, 8 mM sodium palmitate was dissolved in 18.4% fatty-acid free BSA at 37 °C in DMEM medium with 25 mM glucose to obtain a conjugation ratio of 1:3 palmitate:BSA[80]. As described above, the 3T3-F442A adipocytes were incubated with vehicle or SN-401, SN-406, Inactive 2 at 10 μM for 96 h and treated with 1 mM palmitate for an additional 16 h in the presence of compounds and lysates were collected and further processed.

For primary hepatocytes isolation, the *SWELL1[fl/fl]* (Wildtype, WT) and Albumin-Cre/*SWELL1[fl/fl]* (Knockout, KO) mice were anesthetized using isoflurane. A 24 G i.v. catheter was inserted into the portal vein and a buffer containing HBSS (Ca²⁺ and Mg²⁺ free) + 0.5 mM EGTA was perfused through the liver at a rate of 4.5 mL/min. The portal vein was quickly cut after the perfusion began. The liver was perfused until it was free from blood and subsequently perfused with DMEM + 1x Pyruvate+ Type IV Collagenase (Sigma C5138) (1 mg/mL) for 4–5 min until the liver was soft. The liver was excised and put into 10 mL DMEM + collagenase solution. The tissue was broken down by agitation and passed through 25- and 10-mL pipette until the cells were free. The cells were suspended in 15 mL cold DMEM + 10% FBS + 1% P/S (penicillin and streptomycin). The cells were then passed through a 100 μm filter and spun at 50×g for 2 min. The supernatant was discarded and the cell pellet was washed with 25 mL of DMEM + 10% FBS + 1% P/S media three times. Cells were checked for viability by diluting 1:2 with Trypan blue and batches with >80% viability were used. The hepatocytes were then plated on Type I Collagen (75 μg/mL) coated dish with a density of 500,000 cells per well (6-well) in DMEM + 10% FBS + 1% P/S media. After 4 h, media containing vehicle (DMSO) or 10 μM SN-401 was added to the hepatocytes and maintained for 18- and 42 h. The cells were then starved for 6 h in serum-free media and stimulated with serum-containing media (+DMSO or +SN-401) for 15 min and lysates collected.

HUVECs were purchased from ATCC (CRL-2922™) and were grown in M199 growth media supplemented with 20% FBS, 0.05 g Heparin Sodium Salt (Alfa Aesar), and 15 mg ECGS (Millipore Sigma). Cells were cultured on 1% of gelatin-coated plates at 37 °C and 5% CO₂. For SN-40X stimulated eNOS and AKT signaling assays, HUVECs were treated for 96 h with the respective compounds and serum-starved overnight (+DMSO or +SN-40X) for 16 h in M199 media plus 1% FBS (Atlanta Biological). After the serum starve, HUVECs were returned to normal growth media for 30 min (+DMSO or +SN-40X) prior to lysate collection. Small interfering RNA (siRNA) mediated knockdown was adapted from Koh et. al., 2008[81]. Briefly, HUVECs were transfected with either a silencer select siRNA with si-SWELL1 (Cat#4392420, sense: GCAACUUCUGGUUCCAAAUUTT antisense: AAUUUGAACCAGAAGUUGCTG, Invitrogen) or a non-targeting control silencer select siRNA (Cat# 4390846, Invitrogen) upon reaching 90–95% confluency. siRNA was transfected twice, 24 and 72 h after initial seeding of HUVECs. Each siRNA was combined with Opti-MEM (285.25 μl, Cat#11058-021, Invitrogen) siPORT™ amine (8.75 μl, Cat#AM4503, Invitrogen), and the silencer select siRNA (6 μl) in a final volume of 300 μl. HUVECs were transfected for 4 h at 37 °C in 1% FBS containing DMEM media. For short hairpin RNA (shRNA) mediated knockdown approach, HUVECs were transduced with either human adenovirus type 5 targeting SWELL1 (Ad5-shSWELL1) or scrambled non-targeting control (Ad5-shSCR) at a multiplicity of infection (MOI) of 50 for 24 h at 37 °C upon reaching 70% confluency. The cells were then washed with DMEM media and transduced a second time, with a fresh virus, with an MOI of 25 for 12 h at 37 °C. After the final transduction step, the cells were serum-starved as described above for HUVECs, and lysates were collected. The SN-40X compounds were present in the culture media throughout the transfection/transduction for both the si/sh RNA mediated knockdown approaches. HEK-293 (ATCC, #CRL-1573™) cells were maintained in 90% DMEM (25 mM D-Glucose and 4 mM L-Glutamine) containing 10% fetal bovine serum (FBS) and 100 IU penicillin and 100 μg/ml streptomycin.

**Molecular docking**. SN-401 and its analogs were docked into the expanded state structure of an SWELL1-SN-401 homo-hexamer in MSP1E3D1 nanodisc (PDB ID: 6NZZ, https://www.rcsb.org/structure/6NZZ) using Molecular Operating Environment (MOE) 2016.08 software package [Chemical Computing Group (Montreal, Canada)]. The 3D structure obtained from PDB (PDB ID: 6NZZ) was prepared for docking by first generating the missing loops using the loop generation functionality in the Yasara software package followed by sequentially adding hydrogens, adjusting the 3D protonation state, and performing energy minimization using Amber10 force-field in MOE. The ligand structures to be docked were prepared by adjusting partial charges followed by energy minimization using Amber10 force-field. The site for docking was defined by selecting the protein residues within 5 Å from a co-crystallized ligand (SN-401). Docking parameters were set as Placement: Triangle matcher; Scoring function: London dG; Retain Poses: 30; Refinement: Rigid Receptor; Re-scoring function: GBVI/WSA dG; Retain poses: 5. Binding poses for the compounds were predicted using the above-validated docking algorithm.

## Chemical Synthesis

*General information.* All commercially available reagents and solvents were used directly without further purification unless otherwise noted. Reactions were monitored either by thin-layer chromatography (carried out on silica plates, silica gel 60 F₂₅₄, Merck) and visualized under UV light. Flash chromatography was performed using silica gel 60 as stationary phase performed under positive air pressure. ¹H NMR spectra were recorded in CDCl₃ on a Bruker Avance spectrometer operating at 300 MHz at ambient temperature unless otherwise noted. All peaks are reported in ppm on a scale downfield from TMS and using the residual solvent peak in CDCl₃ (H δ = 7.26) or TMS (δ = 0.0) as an internal standard. Data for ¹H NMR are reported as follows: chemical shift (ppm, scale), multiplicity (s = singlet, d = doublet, t = triplet, q = quartet, m = multiplet and/or multiplet resonances, dd = double of doublets, dt = double of triplets, br = broad), coupling constant (Hz), and integration. All high-resolution mass spectra (HRMS) were measured on Waters Q-Tof Premier mass spectrometer using electrospray ionization (ESI) time-of-flight (TOF).

*2-cyclopentyl-1-(2,3-dichloro-4-methoxyphenyl)ethan-1-one* (**3**): Cyclopentyl acetyl chloride (15 g, 102 mmol, 1.1 equiv.) was added to a stirring solution of aluminum chloride (13.64 g, 102 mmol, 1.1 equiv.) in dichloromethane (250 ml) at 0 °C and the resulting solution was allowed to stir at 0 °C under nitrogen atmosphere for 10 min. To this was added a solution of 2, 3-dichloro anisole (16.46 g, 92.9 mmol, 1 equiv.) in dichloromethane (50 ml) at 0 °C, and the resulting solution was allowed to warm to room temperature and stirred for 16 h. Once complete, the reaction was added to cold concentrated hydrochloric acid (100 ml) followed by extraction in dichloromethane (150 ml × 3). The organic fractions were pooled, concentrated, and purified by silica gel chromatography using 0–15% ethyl acetate in hexanes as eluent to furnish compound **3** as white solid (22.41 g, 84%). ¹H NMR (300 MHz, CDCl₃) δ 7.39 (d, J = 8.7 Hz, 1H), 6.89 (d, J = 8.7 Hz, 1H), 3.96 (s, 3H), 2.96 (d, J = 7.2 Hz, 2H), 2.38 – 2.21 (m, 1H), 1.92 – 1.75 (m, 2H), 1.69 – 1.46 (m, 4H), 1.28 – 1.05 (m, 2H). HRMS (ESI), m/z calcd for C₁₄H₁₇Cl₂O₂ [M + H]⁺ 287.0605, found 287.0603.

*6,7-dichloro-2-cyclopentyl-5-methoxy-2,3-dihydro-1H-inden-1-one* (**4**): To 2-cyclopentyl-1-(2,3-dichloro-4-methoxyphenyl)ethan-1-one (**3**) (21.5 g, 74.8 mmol, 1 equiv.) in a round bottom flask was added paraformaldehyde (6.74 g, 224.5 mmol, 3 equiv.), dimethylamine hydrochloride (30.52 g, 374 mmol, 5 equiv.) and acetic acid (2.15 ml) and the resulting mixture was allowed to stir at 85 °C for 16 h. To the reaction was then added dimethylformamide (92 ml) and the resulting solution was allowed to stir at 85 °C for 4 h. Once complete, the reaction was diluted with ethyl acetate and then washed with 1 N hydrochloric acid. The organic fractions were collected and concentrated under vacuum and used for the next step without purification. To the concentrated product in a round bottom flask was added cold concentrated sulfuric acid (120 ml) at 0 °C and the resulting solution was allowed to stir at room temperature for 18 h. Once complete, the reaction was diluted with cold water and extracted thrice with ethyl acetate (100 ml). The organic fractions were pooled, concentrated, and purified by silica gel chromatography using 0–15% ethyl acetate in hexanes as eluent to furnish compound **4** as beige solid (18.36 g, 82%). ¹H NMR (300 MHz, CDCl₃) δ 6.88 (s, 1H), 4.00 (s, 3H), 3.16 (dd, J = 18.1, 8.7 Hz, 1H), 2.80 (d, J = 14.4 Hz, 2H),

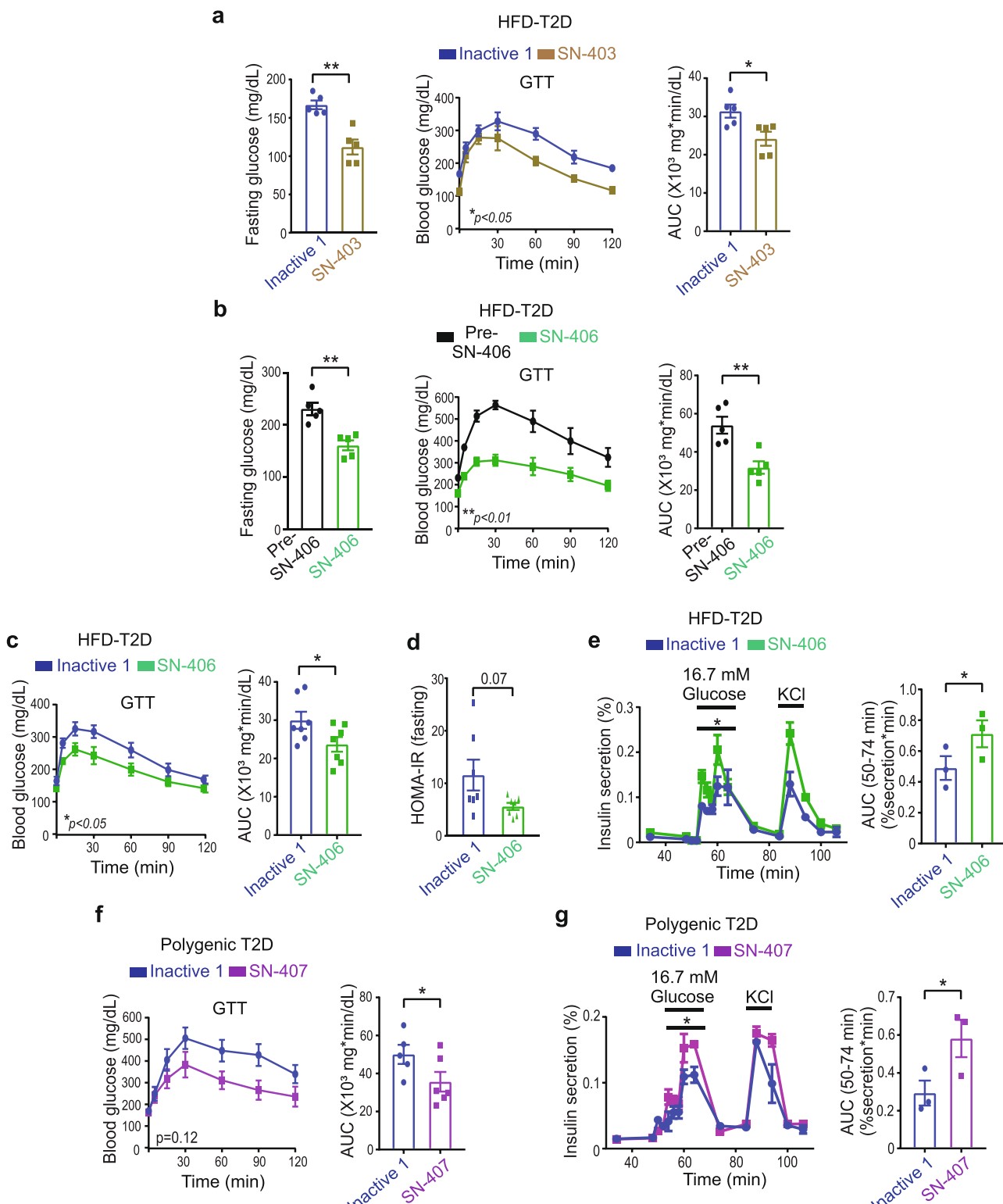

2.43 – 2.22 (m, 1H), 1.96 (s, 1H), 1.73 – 1.48 (m, 5H), 1.46 – 1.33 (m, 1H), 1.17 – 1.00 (m, 1H). LRMS (ESI), $m/z$ calcd for $C_{15}H_{17}Cl_2O_2$ [M + H]$^+$ 299.0605, found 299.0614.

*2-butyl-6,7-dichloro-2-cyclopentyl-5-methoxy-2,3-dihydro-1H-inden-1-one* (**5**): A stirring suspension of **4** (23 gm, 76.8 mmol, 1 equiv.) in anhydrous tert-butanol (220 ml) was allowed to reflux at 95 °C for 30 min. To the resulting solution was added potassium tert-butanol (1 M in tert-butanol) (84 ml, 84.5 mmol, 1.1 equiv.) and the resulting solution was refluxed for 30 min. The reaction was then cooled to room temperature followed by addition of iodobutane (44.2 ml, 384 mmol, 5 equiv.) and the reaction was then allowed to reflux for additional 60 min. The reaction was allowed to cool, concentrated and purified by silica gel

chromatography using 0–10% ethyl acetate in hexanes as eluent to furnish compound **5** as clear oil (17.75 g, 65%). $^1$H NMR (300 MHz, CDCl$_3$) δ 6.89 (s, 1H), 4.09 – 3.90 (m, 3H), 2.98 – 2.70 (m, 2H), 2.36 – 2.18 (m, 1H), 1.89 – 1.71 (m, 2H), 1.58 – 1.42 (m, 5H), 1.33 – 1.09 (m, 4H), 1.09 – 0.94 (m, 2H), 0.93 – 0.73 (m, 4H). HRMS (ESI), $m/z$ calcd for $C_{19}H_{25}Cl_2O_2$ [M + H]$^+$ 355.1231, found 355.1231.

2-butyl-6,7-dichloro-2-cyclopentyl-5-hydroxy-2,3-dihydro-1H-inden-1-one (**6**): To **5** (3.14 g, 8.87 mmol, 1 equiv.) was added aluminum chloride (2.36 g, 17 mmol, 2 equiv.) and sodium iodide (2.7 g, 17 mmol, 2 equiv.) and the resulting solid mixture was triturated and allowed to stir at 70 °C for 60 min. Once complete, the reaction was diluted with dichloromethane and washed with aqueous saturated sodium thiosulfate solution. The organic fractions were collected and concentrated

**Fig. 10 SWELL1-active SN-401 congeners improve glycemic control in murine T2D models. a** Fasting glucose levels, GTT, and the corresponding area under the curve (AUC) of 8-week HFD-fed mice treated with Inactive 1 or SN-403 (5 mg/kg i.p) for 4 days ($n = 5$ in each group). **b** Fasting glucose levels, GTT, and the corresponding AUC of 12 weeks HFD-fed mice pre- and post-treatment with SN-406 (5 mg/kg i.p) for 4 days ($n = 5$ in each group). **c, d** GTT and corresponding AUC (**c**) of 12 weeks HFD-fed mice treated with Inactive 1 or SN-406 (5 mg/kg i.p) for 4 days ($n = 7$ in each group) and the corresponding HOMA-IR index (**d**). **e** Glucose-stimulated insulin secretion (GSIS) perifusion assay of islets isolated from mice in **c** (left) and the corresponding AUC (right). **f** GTT and corresponding AUC of polygenic-T2D KKA$^y$ mice treated with Inactive 1 ($n = 5$) or SN-407 ($n = 6$) (5 mg/kg i.p) for 4 days. **g** Glucose-stimulated insulin secretion (GSIS) perifusion assay of islets isolated from mice in **f** (left) and the corresponding AUC (right). Data were represented as mean ± SEM. A two-tailed unpaired $t$-test was used in **a**, **c**–**g** for FG, GTT AUC, GSIS AUC, and HOMA-IR. Paired $t$-test was used in **b** for FG and GTT AUC. Two-way ANOVA was used in **a**–**c** and **f** for GTTs. Statistical significance is denoted by *, **, and *** representing $p < 0.05$, $p < 0.01$, and $p < 0.001$, respectively.

to give a beige solid which was then washed multiple times with hexanes to provide compound **6** as white solid (2.87 g, 95%). $^1$H NMR (300 MHz, CDCl$_3$) δ 7.03 (s, 1H), 6.32 (s, 1H), 2.97 – 2.73 (m, 2H), 2.36 – 2.17 (m, 1H), 1.88 – 1.68 (m, 2H), 1.62 – 1.39 (m, 6H), 1.31 – 1.11 (m, 3H), 1.08 – 0.97 (m, 2H), 0.97 – 0.87 (m, 1H), 0.83 (t, $J = 7.3$ Hz, 3H). HRMS (ESI), $m/z$ calcd for C$_{18}$H$_{23}$Cl$_2$O$_2$ [M + H]$^+$ 341.1075, found 341.1089.

*2-((2-butyl-6,7-dichloro-2-cyclopentyl-1-oxo-2,3-dihydro-1H-inden-5-yl)oxy) acetic acid* (**7**) (**Inactive 1**): To a stirring solution of **6** (170 mg, 0.50 mmol, 1 equiv.) in anhydrous dimethylformamide (1 ml) was added potassium carbonate (76 mg, 0.56 mmol, 1.1 equiv.) and ethyl 2-bromoacetate (61 ml, 0.56 mmol, 1.1 equiv.) and the reaction was allowed to stir at 60 °C for 2 h. Once complete, to the reaction was added 4 N NaOH (1 ml) and the reaction was allowed to stir at 100 °C for 60 min. Once complete, reaction was concentrated and purified by column chromatography using 0–10% methanol in dichloromethane as eluent to provide **Inactive 1** as a clear solid (173 mg, 87%). $^1$H NMR (300 MHz, CDCl$_3$) δ 6.80 (s, 1H), 5.88 (s, 1H), 4.88 (s, 2H), 2.87 (q, $J = 17.9$ Hz, 2H), 2.34 – 2.20 (m, 1H), 1.91 – 1.69 (m, 2H), 1.66 – 1.39 (m, 6H), 1.32 – 1.13 (m, 3H), 1.10 – 0.95 (m, 2H), 0.94 – 0.86 (m, 1H), 0.83 (t, $J = 7.3$ Hz, 3H). HRMS (ESI), $m/z$ calcd for C$_{20}$H$_{25}$Cl$_2$O$_4$ [M + H]$^+$ 399.1130, found 399.1132. See Supplementary Fig. S14a.

*4-((2-butyl-6,7-dichloro-2-cyclopentyl-1-oxo-2,3-dihydro-1H-inden-5-yl)oxy) butanoic acid* (**8**) (**SN-401**): To a stirring solution of **6** (100 mg, 0.29 mmol, 1 equiv.) in anhydrous dimethylformamide (1 ml) was added potassium carbonate (45 mg, 0.32 mmol, 1.1 equiv.) and ethyl 4-bromobutyrate (46 ml, 0.32 mmol, 1.1 equiv.) and the reaction was allowed to stir at 60 °C for 2 h. Once complete, to the reaction was added 4 N NaOH (1 ml) and the reaction was allowed to stir at 100 °C for 60 min. Once complete, reaction was concentrated and purified by column chromatography using 0–10% methanol in dichloromethane as eluent to provide **SN-401** as a clear solid (111 mg, 89%). $^1$H NMR (300 MHz, CDCl$_3$) δ 10.77 (s, 1H), 6.86 (s, 1H), 4.21 (t, $J = 5.9$ Hz, 2H), 2.88 (t, $J = 14.4$ Hz, 2H), 2.69 (t, $J = 7.0$ Hz, 2H), 2.26 (dd, $J = 12.6, 6.1$ Hz, 3H), 1.87 – 1.73 (m, 2H), 1.64 – 1.44 (m, 6H), 1.35 – 1.10 (m, 4H), 1.08 – 0.95 (m, $J = 15.0, 7.7$ Hz, 2H), 0.82 (t, $J = 7.3$ Hz, 3H). HRMS (ESI), $m/z$ calcd for C$_{22}$H$_{29}$Cl$_2$O$_4$ [M + H]$^+$ 427.1443, found 427.1446. See Supplementary Fig. S14a.

*5-((2-butyl-6,7-dichloro-2-cyclopentyl-1-oxo-2,3-dihydro-1H-inden-5-yl)oxy) pentanoic acid* (**9**) (**SN-403**): To a stirring solution of **6** (100 mg, 0.29 mmol, 1 equiv.) in anhydrous dimethylformamide (1 ml) was added potassium carbonate (45 mg, 0.32 mmol, 1.1 equiv.) and ethyl 6-bromovalerate (51 ml, 0.32 mmol, 1.1 equiv.) and the reaction was allowed to stir at 60 °C for 2 h. Once complete, to the reaction was added 4 N NaOH (1 ml) and the reaction was allowed to stir at 100 °C for 60 min. Once complete, reaction was concentrated and purified by column chromatography using 0–10% methanol in dichloromethane as eluent to provide **SN-403** as a clear solid (114 mg, 88%). $^1$H NMR (300 MHz, CDCl$_3$) δ 10.95 (s, 1H), 6.85 (brs, 1H), 4.16 (t, $J = 5.7$ Hz, 2H), 2.96 – 2.75 (m, 2H), 2.61 – 2.44 (m, 2H), 2.35 – 2.17 (m, 1H), 2.10 – 1.87 (m, 4H), 1.86 – 1.70 (m, 2H), 1.66 – 1.38 (m, 6H), 1.32 – 1.13 (m, 3H), 1.08 – 0.96 (m, 2H), 0.94 – 0.86 (m, 1H), 0.86 – 0.73 (m, 3H). HRMS (ESI), $m/z$ calcd for C$_{23}$H$_{31}$Cl$_2$O$_4$ [M + H]$^+$ 441.1599, found 441.1601. See Supplementary Fig. S14a.

*6-((2-butyl-6,7-dichloro-2-cyclopentyl-1-oxo-2,3-dihydro-1H-inden-5-yl)oxy) hexanoic acid* (**10**) (**SN-406**): To a stirring solution of **6** (100 mg, 0.29 mmol, 1 equiv.) in anhydrous dimethylformamide (1 ml) was added potassium carbonate (45 mg, 0.32 mmol, 1.1 equiv.) and ethyl 6-bromohexanoate (58 ml, 0.32 mmol, 1.1 equiv.) and the reaction was allowed to stir at 60 °C for 2 h. Once complete, to the reaction was added 4 N NaOH (1 ml) and the reaction was allowed to stir at 100 °C for 60 min. Once complete, reaction was concentrated and purified by column chromatography using 0–10% methanol in dichloromethane as eluent to provide **SN-406** as a clear solid (115 mg, 86%). $^1$H NMR (300 MHz, CDCl$_3$) δ 11.70 (s, 1H), 6.85 (s, 1H), 4.13 (t, $J = 6.2$ Hz, 2H), 2.93 – 2.74 (m, 2H), 2.43 (t, $J = 7.3$ Hz, 2H), 2.32 – 2.17 (m, 1H), 1.98 – 1.87 (m, 2H), 1.85 – 1.68 (m, 4H), 1.66 – 1.40 (m, 8H), 1.28 – 1.12 (m, 3H), 1.07 – 0.93 (m, 2H), 0.91 – 0.70 (m, 4H). HRMS (ESI), $m/z$ calcd for C$_{24}$H$_{33}$Cl$_2$O$_4$ [M + H]$^+$ 455.1756, found 455.1756. See Supplementary Fig. S14a.

*7-((2-butyl-6,7-dichloro-2-cyclopentyl-1-oxo-2,3-dihydro-1H-inden-5-yl)oxy) heptanoic acid* (**11**) (**SN-407**): To a stirring solution of **6** (100 mg, 0.29 mmol, 1 equiv.) in anhydrous dimethylformamide (1 ml) was added potassium carbonate (45 mg, 0.32 mmol, 1.1 equiv.) and ethyl 7-bromoheptanoate (63 ml, 0.32 mmol,

1.1 equiv.) and the reaction was allowed to stir at 60 °C for 2 h. Once complete, to the reaction was added 4 N NaOH (1 ml) and the reaction was allowed to stir at 100 °C for 60 min. Once complete, reaction was concentrated and purified by column chromatography using 0–10% methanol in dichloromethane as eluent to provide **SN-407** as a clear solid (122 mg, 89%). $^1$H NMR (300 MHz, CDCl$_3$) δ 11.52 (s, 1H), 6.85 (s, 1H), 4.12 (t, $J = 6.3$ Hz, 2H), 2.84 (q, $J = 18.2$ Hz, 2H), 2.47 – 2.32 (m, 2H), 2.32 – 2.18 (m, 1H), 1.96 – 1.84 (m, 2H), 1.83 – 1.64 (m, 4H), 1.62 – 1.39 (m, 10H), 1.28 – 1.14 (m, 3H), 1.08 – 0.94 (m, 2H), 0.91 (d, $J = 8.5$ Hz, 1H), 0.81 (t, $J = 7.3$ Hz, 3H). HRMS (ESI), $m/z$ calcd for C$_{25}$H$_{35}$Cl$_2$O$_4$ [M + H]$^+$ 469.1912, found 469.1896. See Supplementary Fig. S14a.

*4-((6,7-dichloro-2-cyclopentyl-1-oxo-2,3-dihydro-1H-inden-5-yl)oxy)butanoic acid* (**12**) (**Inactive 2**): To **4** (100 mg, 0.36 mmol, 1 equiv.) was added aluminum chloride (89 mg, 0.67 mmol, 2 equiv.) and sodium iodide (101 mg, 0.67 mmol, 2 equiv.) and the resulting solid mixture was triturated and allowed to stir at 70 °C for 60 min. Once complete, the reaction was diluted with dichloromethane and washed with an aqueous saturated sodium thiosulfate solution. The organic fractions were collected and concentrated to give a beige solid which was then washed multiple times with hexanes to provide phenol intermediate as a white solid which was used for the next step. To a stirring solution of the product form the first step in anhydrous dimethylformamide (1 ml) was added potassium carbonate (53 mg, 0.39 mmol, 1.1 equiv.) and ethyl 4-bromobutyrate (55 ml, 0.39 mmol, 1.1 equiv.), and the reaction was allowed to stir at 60 °C for 2 h. Once complete, the reaction was added 4 N NaOH (1 ml) and the reaction was allowed to stir at 100 °C for 60 min. Once complete, reaction was concentrated and purified by column chromatography using 0–10% methanol in dichloromethane as eluent to provide Inactive 2 as a clear solid (107 mg, 86%). $^1$H NMR (300 MHz, CDCl$_3$) δ 6.87 (s, 1H), 4.21 (t, $J = 5.9$ Hz, 2H), 3.26 – 3.02 (m, 1H), 2.94 – 2.56 (m, 4H), 2.40 – 2.19 (m, 3H), 2.03 – 1.90 (m, 1H), 1.74 – 1.50 (m, 5H), 1.47 – 1.32 (m, 1H), 1.19 – 1.00 (m, 1H). HRMS (ESI), $m/z$ calcd for C$_{18}$H$_{21}$Cl$_2$O$_4$ [M + H]$^+$ 371.0817, found 371.0808. See Supplementary Fig. S14b.

*(3-((2-butyl-6,7-dichloro-2-cyclopentyl-1-oxo-2,3-dihydro-1H-inden-5-yl)oxy) propyl)boronic acid* (**13**) (**Inactive 3**): To a stirring solution of 6 (100 mg, 0.29 mmol, 1 equiv.) in anhydrous dimethylformamide (1 ml) was added potassium carbonate (45 mg, 0.32 mmol, 1.1 equiv.) and 2-(3-bromopropyl)-4,4,5,5-tetramethyl-1,3,2-dioxaborolane (80 mg, 0.32 mmol, 1.1 equiv.) and the reaction was allowed to stir at 60 °C for 4 h. Once complete, the solvent was removed under vacuum and crude reaction was then diluted with 3 ml of 0.1 N HCl/acetone (1:1 v/v). To this was added methane boronic acid (53 mg, 0.88 mmol, 3 equiv.) and the reaction was allowed to stir at room temperature for 16 h. Once complete, reaction was concentrated and purified by column chromatography using 0–10% methanol in dichloromethane as eluent to provide Inactive 3 as a clear solid. $^1$H NMR (300 MHz, CDCl$_3$) δ 6.88 (d, $J = 3.4$ Hz, 1H), 4.15 (dd, $J = 11.2, 5.4$ Hz, 2H), 2.85 (q, $J = 17.8$ Hz, 2H), 2.27 (m, 1H), 2.13 – 1.98 (m, 2H), 1.76 (m, 2H), 1.53 (m, 6H), 1.22 (m, 3H), 1.12 – 0.94 (m, 4H), 0.90 (s, 1H), 0.86 – 0.78 (m, 3H). HRMS (ESI), $m/z$ calcd for C$_{21}$H$_{30}$BCl$_2$O$_4$ [M + H] 427.1614, found 427.1618. See Supplementary Fig. S14c. The $^1$H NMR spectra of all compounds are provided as Supplementary Data. 1a–k.

**Electrophysiology**. Patch-clamp recordings of β-cells, 3T3-F442A adipocytes, and mature human adipocytes were performed as described previously[24,28]. Briefly, all patch-clamp experiments were conducted at room temperature using either an Axopatch 200B amplifier or a MultiClamp 700B amplifier paired to a Digidata 1550 digitizer. Data were acquired using pClamp 10.4 software. For hypotonic swelling, extracellular solution consisted of the following (in mM): 90 NaCl, 2 CsCl, 1 MgCl$_2$, 1 CaCl$_2$, 10 HEPES, 10 mannitol, pH 7.4 with NaOH (210 mOsm/kg). The isotonic extracellular solution consisted of the same composition above but with 110 mM instead of 10 mM mannitol (300 mOsm/kg). The swell-activated current was elicited by perfusing cells with a hypotonic solution (210 mOsm/kg). The intracellular solution contained (in mM): 120 L-aspartic acid, 20 CsCl, 1 MgCl$_2$, 5 EGTA, 10 HEPES, 5 MgATP, 120 CsOH, 0.1 GTP, pH 7.2 with CsOH and had an osmolarity of 280–290 mOsm/kg. The patch electrodes were prepared from borosilicate glass capillaries (WPI) and had a resistance of 2.5–4.8 MΩ when filled with pipette solution. For perforated-patch recordings in isolated human adipocytes, the intracellular solution was as above but without ATP and GTP, and contained 360 µg/ml Amphotericin B (Sigma). The holding potential was 0 mV.

Voltage ramps from −100 to +100 mV (at 0.4 mV/ms) were applied every 4 s. The sampling interval was 100 μs and filtered at 10 kHz. For perforated-patch recordings, cells with a membrane resistance below 1 GΩ or access resistance above 20 MΩ were discarded.

3T3-F442A WT and KO preadipocytes were prepared as described in the Cell culture section above. For SWELL1 overexpression recordings, preadipocytes were first transduced with Ad5-CAG-LoxP-stop-LoxP-3XFlag-SWELL1 (MOI 12) in 2% FBS culture medium for 2 days and then overexpression induced by adding Ad5-CMV-Cre- eGFP (MOI 10-12) in 2% FBS culture medium for 2 more days and changed to 10% FBS containing culture media and were selected based on GFP expression (~2–3 days).

For β-cell recordings, islets were transduced with Ad-RIP2-GFP and then dispersed after 48–72 h for patch-clamp experiments. GFP + cells marked β-cells selected for patch-clamp recordings. Non-T2D islets were isolated from mice on a regular chow diet between 8–13 weeks of age. Of these mice, four had an average body weight of 28.6 ± 0.51 g and blood glucose level of 148 ± 6.49 mg/dl respectively. T2D islets were obtained from mice fed with HFD for 4–5 months and their average body weight and glucose levels were 52.7 ± 3.0 g and 229 ± 21.4 mg/dl, respectively.

For measuring $I_{Cl,SWELL}$ inhibition by SN-401 and SN-401 congeners (SN-40X, and Inactive 1–3) after activation of $I_{Cl,SWELL}$, HEK-293 cells were perfused with hypotonic solution (Hypo, 210 mOsm/L) and then SN-401 and SN-40X + Hypo were applied at varying concentrations (1, 3, 4.5, 6, and 10 uM) to assess for % $I_{Cl,SWELL}$ inhibition, which was calculated based on the following equation: (Peak current − current after compound application for 3 min)/(peak current − baseline current)*100. See also Fig. 6 for a schematic representation. For consistency among recordings, SN-401, SN-40X compounds, and Inactive compounds were applied when the endogenous HEK-293 cell $I_{Cl,SWELL}$ current density reached 30 pA/pF. For SWELL1-R103E overexpression experiments, HEK-293 cells were transfected with either control Cre plasmid (CMV-Cre, McDonnell Genome Institute in Washington University in St. Louis) or Cre plus R103E-GFP (CAG-LoxP-stop-LoxP-SWELL1-R103E/3XFlag:P2A:EGFP, Vector Biolabs) plasmid using Lipofectamine 2000 as per the manufacturer's protocol. SWELL1-R103E expressing HEK-293 cells were identified by EGFP + fluorescence.

To assess for $I_{Cl,SWELL}$ inhibition upon application of SN-401 congeners prior to $I_{Cl,SWELL}$ activation (ie., to the closed SWELL1-LRRC8 channel), HEK-293 cells were preincubated with vehicle (or SN-401, SN-406, Inactive 1, and Inactive 2) for 30 min prior to hypotonic stimulation and then stimulated with hypotonic solution + SN-401 congeners.

**Western blot**. Adipocytes were washed twice in ice-cold phosphate buffer saline (PBS) and lysed in RIPA buffer (150 mM NaCl, 20 mM HEPES, 1% NP-40, 5 mM EDTA, pH 7.4) with proteinase/phosphatase inhibitors (Roche). The cell lysate was further sonicated in 10 s cycle intervals 2–3 times and centrifuged at 18,407×g for 30 min at 4 °C and repeated one more time to remove the excess fat. The supernatant was collected and further estimated for protein concentration using a DC protein assay kit (Bio-Rad). Fat and liver tissues were homogenized and suspended in RIPA buffer with inhibitors and further processed in a similar fashion as described above. HUVECs were also prepared in a similar fashion except the lysates were spun only once to remove the cells debris to obtain the clear supernatant. For islets, either human or mouse islets were washed twice with phosphate buffer saline (PBS) and lysed using RIPA buffer containing protease and phosphatase inhibitors. The lysate was further clarified by freezing in liquid nitrogen for about 10 s for three cycles. The supernatant was collected from the whole lysate centrifuged at 16,260×g for 20 min at 4 °C. For in vivo signaling studies, mice treated with either vehicle or SN-401 were fasted for 6 h and tissues (fat, liver, and muscle) were excised and snap-frozen upon euthanization. For soleus muscle, the tissue was homogenized in a Bullet Blender homogenizer (Speed-9, Time-3 mins) in 8 volumes of ice-cold muscle homogenization buffer (50 mM Tris, 25 mM NaCl, 0.2% Nonidet P-40, 5 mM EGTA, 2.5 mM EDTA, 20 mM NaF, 20 mM $Na_4P_2O_7 \cdot 10H_2O$, 2 mM $Na_3VO_4$, pH 7.4) supplemented with protease/phosphatase inhibitor (Roche). The homogenized lysate was centrifuged two times at 18,500×g for 20 min at 4 °C and supernatant was collected for protein estimation. Protein samples were further prepared by boiling in 2X or 4X Laemmli dye. Approximately 10–60 μg of total protein was loaded in 4–15% gradient gel (Bio-Rad) for separation and protein transfer was carried out onto the PVDF membranes (Bio-Rad). Membranes were blocked in 5% BSA (or 5% milk for SWELL1) in TBST buffer (0.2 M Tris, 1.37 M NaCl, 0.2% Tween-20, pH 7.4) for 1 h and incubated with appropriate primary antibodies (5% BSA or milk) overnight at 4 °C. The membranes were further washed in TBST buffer before adding secondary antibody (Bio-Rad, Goat-anti-rabbit, #170-6515) in 1% BSA (or 1% milk for SWELL1) in TBST buffer for 1 h at RT. The signals were developed by chemiluminescence (Pierce) and visualized using a Chemidoc imaging system (Bio-Rad). The images were further analyzed for band intensities using ImageJ software. Following primary antibodies were used: anti-phospho-AKT2 (Ser474, #8599 s,1:1000), anti-AKT2 (#3063 s,1:1000), anti-AKT (#4685,1:1000), anti-phospho-AS160 (Thr642, #4288 s,1:1000), anti-AS160 (#2670 s,1:1000), anti-GAPDH (#D16H11, 1:2000), p-eNOS (Ser1177, #9571,1:1000), Total eNOS (#32027,1:1000), pGSK3β (Ser9, #9336,1:1000), Total GSK3β (#9832,1:1000) and anti-β-actin (#8457 s,1:2500) from Cell Signaling; anti-SWELL1 (1:1000), rabbit polyclonal antibody was custom generated against the epitope QRTKSRIEQGIVDRSE (Pacific Immunology, CA).

**Immunofluorescence**. 3T3-F442A preadipocytes (WT, KO) and differentiated adipocytes without or with SWELL1 overexpression (WT + SWELL1 O/E, KO + SWELL1 O/E) were prepared as described in the Cell culture section on collagen-coated coverslips. In the case of SWELL1 membrane trafficking, the 3T3-F442A preadipocytes were incubated in the presence of vehicle (or SN-401, SN-406, and Inactive 1) at either 1 or 10 μM for 48 h. The cells were fixed in ice-cold acetone for 15 min at −20 °C and then washed four times with 1XPBS. Further, cells were permeabilized using 0.1% Triton X-100 in 1XPBS for 5 min at RT and subsequently blocked with 5% normal goat serum for 1 h at RT. Either anti-SWELL1 (1:400) or anti-Flag (1:1500, Sigma #F3165) antibody was added to the cells and incubated overnight at 4 °C. The cells were then washed three times (1XPBS) prior to and after the addition of 1:1000 Alexa Fluor 488/568 secondary antibody (anti-rabbit, #A11034 or anti-mouse, #A11004) for 1 h at RT. Cells were counterstained with nuclear TO-PRO-3 (Life Technologies, #T3605) or DAPI (Invitrogen, #D1306) staining (1 μM) for 20 min followed by three washes with 1XPBS. Coverslips were further mounted on slides with ProLong Diamond anti-fading media. All images were captured using Zeiss LSM700/LSM510 confocal microscope with 63X objective (NA 1.4). SWELL1 membrane localization was quantified by stacking all the z-images and converting it into a binary image where the cytoplasmic intensity per unit area was subtracted from the total cell intensity per unit area using ImageJ software.

**Metabolic phenotyping**. Mice were fasted for 6 h prior to glucose tolerance tests (GTT). Baseline glucose levels at 0 min timepoint (fasting glucose, FG) were measured from a blood sample collected from tail snipping using glucometer (Bayer Healthcare LLC). Either 1 or 0.75 g D-Glucose/kg body weight were injected (i.p.) for lean or HFD mice, respectively and glucose levels were measured at 5 or 7, 15, 30, 60, 90, and 120 min timepoints after injection. For insulin tolerance tests (ITTs), the mice were fasted for 4 h. Similar to GTTs, the baseline blood glucose levels were measured at 0 min timepoint and 15, 30, 60, 90, and 120 min timepoints post-injection (i.p.) of insulin (HumulinR, 1 U/kg bodyweight for lean mice or 1.25 U/kg body weight for HFD mice). GTTs or ITTs with vehicle (or SN-401, SN-403, SN-406, SN-407, and Inactive 1) treated groups were performed approximately 24 h after the last injection. For measuring serum insulin levels, the vehicle (or SN-401, SN-406, and Inactive 1) treated HFD mice were fasted for 6 h and injected (i.p.) with 0.75 g D-Glucose/kg body weight and blood samples were collected at 0, 7, 15, and 30 min time points in microvette capillary tubes (SARSTEDT, #16.444) and centrifuged at 2000×g for 20 min at 4 °C. The collected plasma was then measured for insulin content by using Ultra-Sensitive Mouse Insulin ELISA Kit (Crystal Chem, #90080). All mouse studies were performed in a blinded fashion. Body weights for all the mice are listed in the supplementary table.

**Adipose tissue morphology and size quantification**. Mice epididymal white adipose tissue (eWAT) was isolated and fixed in 10% neutral buffer formalin for 24 h at 4 °C. The formalin-fixed eWAT tissue was washed twice with 1XPBS buffer and dehydrated in a series of ethanol gradient. The dehydrated tissue was embedded in paraffin wax and 10 μm sections were cut by automated microtome and mounted on positively charged slides. Hematoxylin and eosin staining of the slides was performed at the Musculoskeletal Research Center, Washington University in St. Louis. Images were captured by Zeiss Axioscop with 10X objective and analyzed by ImageJ software. Measurement of adipocyte area was performed using the polygon selection tool of ImageJ. All quantifications were performed by blinded independent observers.

**Islet isolation and perifusion assay**. For patch-clamp studies involving primary mouse β-cells, mice were anesthetized by injecting Avertin (0.0125 g/ml in $H_2O$) followed by cervical dislocation. HFD or polygenic KKAy mice treated with either vehicle or SN-401, SN-406, SN-407, and Inactive 1 were anesthetized with 1–4% isoflurane followed by cervical dislocation. The pancreas was perfused via the common bile duct with 2–3 ml HBSS containing type V collagenase (0.8 mg/ml), removed, and digested at 37 °C for 10 min. Islets were then dissociated by gentle agitation, washed in RPMI containing 1% FBS, and purified on Histopaque 1077 and 1119 gradients. Islets were subsequently transferred to a 60 mm Petri dish with a culture medium for short-term culture (<24 h). For GSIS experiments, islets were sorted for equal size and cultured in 24-well plates. For isolation of primary β-cells, isolated islets were further incubated in trypsin for 5 min, dispersed into single cells, and then transferred to Matrigel-coated coverslips for patch-clamp.

Cadaveric human islets were obtained from the Integrated Islet Distribution Program (IIDP), Prodo Laboratories, and the Alberta Diabetes Institute Islet Core. The human islets obtained from the Integrated Islet Distribution program and Prodo laboratories were exempt from Institutional Review Board approval under 45 CRR 46.101 (b) category (4) and 45 CFR 46.102 h, respectively. Human islets were obtained from the Alberta Diabetes Institute IsletCore at the University of Alberta in Edmonton (http://www.bcell.org/adi-isletcore.html) with the assistance of the Human Organ Procurement and Exchange (HOPE) program, Trillium Gift of Life Network (TGLN), and other Canadian organ procurement organizations. Islet isolation was approved by the Human Research Ethics Board at the University of Alberta (Pro00013094). All donors' families gave informed consent for the use of pancreatic tissue in research. Human islets were cultured in RPMI media with 2% FBS overnight. The next day either scramble or shSWELL1 adenoviral transduction

was carried out (final concentration of $5 \times 10^7$ PFU/ml) and the islets were incubated for 12 h. The islets were then washed with 1XPBS three times and cultured in RPMI medium with 10% FBS for 4–5 days. For SN-401/Palmitate experiments, human islets were either transduced with an adenoviral short hairpin for control (shScramble) or SWELL1 knockdown (shSWELL1), and murine islets isolated from floxed-SWELL1 mouse (SWELL1$^{fl/fl}$) were either transduced with adenoviral control (Ad-RIP1-GCaMP6s) or Cre-recombinase (Ad-RIP1-GCaMP6s-2A-Cre) virus for 12 h, respectively in 2% FBS containing RPMI media. The islets were then washed in 1XPBS three times and treated with either vehicle or SN-401 (10 μM) for 96 h followed by treatment with 1:3 palmitate:BSA with or without SN-401 in 10% FBS containing RPMI media for 16 h (Fig. 9c). The GSIS perifusion assay for islets were performed using a PERI4-02 from Biorep Technologies. For each experiment, around 50 freshly isolated islets (all from the same isolation batch) were handpicked to match the size of islets across the samples and loaded into the polycarbonate perifusion chamber between two layers of polyacrylamide-microbead slurry (Bio-Gel P-4, Bio-Rad) by the same experienced operator. Perifusion buffer contained (in mM): 120 NaCl, 24 NaHCO₃, 4.8 KCl, 2.5 CaCl₂, 1.2 MgSO₄, 10 HEPES, 2.8 glucose, 27.2 mannitol, 0.25% w/v bovine serum albumin, pH 7.4 with NaOH (300 mOsm/kg). Perifusion buffer kept at 37 °C was circulated at 120 μl/min. After 48 min of washing with 2.8 mM glucose solution for stabilization, islets were stimulated with the following sequence: 16 min of 16.7 mM glucose, 40 min of 2.8 mM glucose, 10 min of 30 mM KCl, and 12 min of 2.8 mM glucose. Osmolarity was matched by adjusting mannitol concentration when preparing a solution containing 16.7 mM glucose. Serial samples were collected either every 1 or 2 min into 96 wells kept at 4 °C. Insulin concentrations were further determined using a commercially available ELISA kit (Mercodia). The area under the curve (AUC) for the high-glucose-induced insulin release was calculated for time points between 50 to 74/84 min. At the completion of the experiment, islets were further lysed by the addition of RIPA buffer and the amount of insulin was detected by ELISA.

**Adiponectin and FGF21 measurements.** Polygenic-T2D KKA$^y$ mice (males) between 11–12 weeks old were treated with either vehicle or SN-401 (5 mg/kg/day, i.p.,) for 5 days. On day 6, mice were fasted for 6 h and blood samples were collected in microvette capillary tubes (SARSTEDT, #16.444) and centrifuged at 2000×g for 20 min at 4 °C. Plasma adiponectin and FGF21 levels were measured using Mouse Adiponectin/Acrp30 Quantikine ELISA (Bio-Techne R&D Systems, MRP300) and mouse/rat FGF21 ELISA (BioVendor R&D, RD291108200R) kits respectively, as per the manufacturer's instructions.

**Drug pharmacokinetics.** The pharmacokinetic studies for SN-401 and SN-406 were performed at Charles River Laboratory as outlined below. Male C57/BL6 mice were used in the study and assessed for a single dose (5 mg/kg) administration. The compounds were prepared in Cremaphor for i.p. and p.o dose routes and in 5% ethanol, 10% Tween-20, and water mix for i.v. route at a final concentration of 1 mg/mL. Terminal blood samples were collected via cardiac venipuncture under anesthesia at time points 0.08, 0.5, 2, 8 h post-dose for i.v and at timepoints 0.25, 2, 8, 24 h post-dose for i.p. and p.o. groups, respectively with a sample size of 3 mice per timepoint. The blood samples were collected in tubes with K2 EDTA anticoagulant and further processed to collect plasma by centrifugation at 1383×g at 5 °C for 10 min. Samples were further processed in LC/MS to determine the concentration of the compounds. Non-compartmental analysis was performed to obtain the PK parameters using the PKPlus software package (Simulation Plus). The area under the plasma concentration-time curve (AUC$_{inf}$) was calculated from time 0 to infinity where the C$_{max}$ is the maximal concentration achieved in plasma and t$_{1/2}$ is the terminal elimination half-life. Oral bioavailability was calculated as AUC$_{Oral}$/AUC$_{IV}$*100.

**Hyperinsulinemic euglycemic glucose clamps.** Sterile silicone catheters (Dow-Corning) were placed into the jugular vein of mice under isoflurane anesthesia. The placed catheter was flushed with 200 U/mL heparin in saline and the free end of the catheter was directed subcutaneously via a blunted 14-gauge sterile needle and connected to a small tubing device that exited through the back of the animal. Mice were allowed to recover from surgery for 3 days, then received IP injections of vehicle or SN-401 (5 mg/kg) for 4 days. Hyperinsulinemic euglycemic clamps were performed on day 8 post-surgery on unrestrained, conscious mice as described elsewhere[82,83], with some modifications. Mice were fasted for 6 h at which time insulin and glucose infusion were initiated (time 0). At 80 min prior to time 0 basal sampling was conducted, where whole-body glucose flux was traced by infusion of 0.05 μCi/min D-[3-³H]-glucose (Perkin Elmer), after a priming 5 μCi bolus for 1 min. After the basal period, starting at time 0 D-[3-³H]-glucose was continuously infused at the rate of 0.2 μCi/min and the infusion of insulin (Humulin, Eli Lilly) was initiated with a bolus of 80 mU/kg/min (10 μl volume for 1 min) then followed by continuous infusion of insulin at the dose of 8 mU/kg/min throughout the assay. Fifty percent dextrose (Hospira) was infused at variable rates (GIR) starting at the same time as the initiation of insulin infusion to maintain euglycemia at the targeted level of 150 mg/dL (8.3 mM). Blood glucose (BG) measurements were taken every ten minutes via tail vein sampling using a Contour glucometer (Bayer). After the mouse reached stable BG and GIR (typically, after 75 min since starting the insulin infusion; for some mice,

a long time was required to achieve steady-state) a single bolus of 12 μCi of [1-¹⁴C]-2-deoxy-D-glucose (Perkin Elmer) in 96 μl of saline was administered. Plasma samples (collected from centrifuged blood) for determination of tracers enrichment, glucose level, and insulin concentration were obtained at times −80, −20, −10, 0, and every 10 min starting at 80 min post-insulin (5 min after [1-¹⁴C]-2-deoxy-D-glucose bolus was administered) until the conclusion of the assay at 140 min. Tissue samples were then collected from mice under isofluorane anesthesia from organs of interest (e.g., liver, heart, kidney, white adipose tissue, brown adipose tissue, gastrocnemius, soleus, etc.) for determination of 1-¹⁴C]-2-deoxy-D-glucose tracer uptake. Plasma and tissue samples were processed as described previously[82]. Briefly, plasma samples were deproteinized with Ba(OH)₂ and ZnSO₄ and dried to eliminate tritiated water. The glucose turnover rate (mg/kg-min) was calculated as the rate of tracer infusion (dpm/min) divided by the corrected plasma glucose specific activity (dpm/mg) per kg body weight of the mouse. Fluctuations from steady-state were accounted for by the use of Steele's model. Plasma glucose was measured using the Analox GMD9 system (Analox Technologies).

Tissue samples (~30 mg each) were homogenized in 750 μl of 0.5% perchloric acid, neutralized with 10 M KOH, and centrifuged. The supernatant was then used for first measuring the abundance of total [1-¹⁴C] signal (derived from both 1-¹⁴C -2-deoxy-D-glucose, 1-¹⁴C -2-deoxy-D-glucose 6 phosphate) and, following a precipitation step with 0.3 N Ba(OH)₃ and 0.3 N ZnSO₄, for the measuring of non-phosphorylated 1-¹⁴C -2-deoxy-D-glucose. Glycogen was isolated by ethanol precipitation from 30% KOH tissue lysates, as described[84]. Insulin levels in plasma at T0 and T140 were measured using a Stellux ELISA rodent insulin kit (Alpco).

**Protein purification.** A baculoviral P3 stock for expression of Mus musculus SWELL1 with an added C-terminal PreScission protease cleavage site, linker sequence, superfolder GFP (sfGFP), and 7xHis tag (mmSWELL1-SNS-LEVLFQGP-SRGGSGAAAGSGSGS-sfGFP-GSS-7xHis)[31] was used to infect 1 L of Sf9 cells in ESF 921 media (Expression Systems) at 4 million cells/mL at an MOI ~2–5. After 72 h, infected cells containing expressed protein were harvested by centrifugation at 2500×g and frozen at −80 °C.

Cells from 1 L of culture (~15–20 mL of cell pellet) were thawed in 100 mL of Lysis Buffer containing (in mM) 50 HEPES, 150 KCl, 1 EDTA pH 7.4. Protease inhibitors (Final Concentrations: E64 (1 μg/mL), Pepstatin A (1 μg/mL), Soy Trypsin Inhibitor (10 μg/mL), Benzamidine (1 mM), Aprotinin (1 μg/mL), Leupeptin (1 μg/mL), and PMSF (1 mM)) were added to the lysis buffer immediately before use. Benzonase (4 μl) was added after cell thaw. Cells were then lysed by sonication and centrifuged at 150,000×g for 45 min. The supernatant was discarded, and residual nucleic acid was removed from the top of the membrane pellet using DPBS. Membrane pellets were scooped into a Dounce homogenizer containing Extraction Buffer (50 mM HEPES, 150 mM KCl, 1 mM EDTA, 1% n-Dodecyl-β-D-Maltopyranoside (DDM, Anatrace, Maumee, OH), 0.2% Cholesterol Hemisuccinate Tris Salt (CHS, Anatrace) final pH 7.4. A 10/2% solution of DDM/CHS was first dissolved and clarified by bath sonication in 200 mM HEPES pH 8 prior to addition to buffer to the indicated final concentration. Membrane pellets were homogenized (using a loose-fitting followed by ten strokes with a tight-fitting) and the mixture was then gently stirred at 4 °C for 3 h. The extraction mixture was centrifuged at 33,000×g for 45 min and the supernatant, containing solubilized membrane protein, was bound to 5 mL of sepharose resin coupled to anti-GFP nanobody for 1 h at 4 °C. The resin was collected in a column and washed with 10 mL of Buffer A (20 mM HEPES, 150 mM KCl, 1 mM EDTA, 0.025% DDM, 0.005% CHS, pH 7.4), 40 mL of Buffer B (As A, but with 500 mM KCl), and then 10 mL of Buffer A. The resin was then resuspended in 8 mL of Buffer A with 0.5 mg PreScission protease and rocked gently in the capped column overnight (~12 h) for tag cleavage. SWELL1 protein was eluted with an additional 6 mL of Buffer A, spin concentrated to ~500 μl with Amicon Ultra spin concentrator 100 kDa cutoff (Millipore), and then loaded onto a Superose 6 Increase column (GE Healthcare, Chicago, IL) on an NGC system (Bio-Rad, Hercules, CA) equilibrated in Buffer C (20 mM HEPES, 150 mM KCl, 1 mM EDTA, 0.025% DDM, pH 7.4). Peak fractions containing the SWELL1 channel were collected and spin concentrated for nanodisc preparation.

**Nanodisc formation.** Freshly purified SWELL1 from gel filtration in Buffer C (20 mM HEPES, 150 mM KCl, 1 mM EDTA, 0.025% DDM, pH 7.4) was reconstituted into MSP1E3D1 with a lipid mix (2:1:1 weight ratio of DOPE:POPC:POPS lipids (Avanti, Alabaster, Alabama)) at a final molar ratio of 1:2.5:200 (Monomer Ratio: SWELL1, MSP1E3D1, Lipid Mix). First, solubilized lipid in Column Buffer (20 mM HEPES, 150 mM KCl, 1 mM EDTA pH 7.4) was mixed with additional DDM detergent, Column Buffer, and SWELL1. This solution was mixed at 4 °C for 30 min before the addition of purified MSP1E3D1. This addition brought the final concentrations to ~10 μM SWELL1, 25 μM MSP1E3D1, 2 mM lipid mix, and 3.3 mM DDM in Column Buffer (1 mL reaction). The solution with MSP1E3D1 was mixed at 4 °C for 10 min before the addition of 130 mg of Biobeads SM2 (Bio-Rad, Hercules, CA). Biobeads (washed into methanol, water, and then Column Buffer) were weighed with liquid removed by P1000 tip (Damp weight). This mix was incubated at 4 °C for 30 min before the addition of another 130 mg of Biobeads (final 260 mg of Biobeads per mL). This final mixture was then mixed at 4 °C overnight (~14 h). The supernatant was cleared of beads by letting large beads settle and carefully removing the supernatant with a pipet. The sample was spun for 5 min at 21,000×g before loading onto a Superose 6 column in Column Buffer

without EDTA. Peak fractions corresponding to SWELL1 in MSP1E3D1 were collected, 100 kDa cutoff spin concentrated, and then re-run on the Superose 6. The fractions corresponding to the center of the peak were then pooled and concentrated prior to grid preparation.

**Grid preparation**. SN-407 in DMSO (Stock 10 mM) was added to the SWELL1-MSP1E3D1 sample to give a final concentration of 1 mg/mL SWELL1-MSP1E3D1 and 100 μM SN-407. The drug was allowed to equilibrate and bind complex on ice for ~1 h prior to freezing grids. The sample with the drug was cleared by a 5 min 21,000×g spin prior to grid making. For freezing grids, a 2 μl drop of protein was applied to freshly glow discharged Holey Carbon, 300 mesh R 1.2/1.3 gold grids (Quantifoil, Großlöbichau, Germany). A Vitrobot Mark IV (Thermo Fisher Scientific, Waltham, MA) was utilized with 22 °C, 100% humidity, one blot force, and a 3 s blot time, before plunge freezing in liquid ethane. Grids were then clipped in autoloader cartridges for collection.

**Data collection**. SWELL1-MSP1E3D1 with SN-407 grids were transferred to a Talos Arctica cryo-electron microscope (Thermo Fisher Scientific) operated at an acceleration voltage of 200 kV. Images were recorded in an automated fashion with SerialEM[85] using image shift with a target defocus range of −0.7 to −2.2 μm over 5.493 s as 50 subframes with a K3 direct electron detector (Gatan, Pleasanton, CA) in super-resolution mode with a super-resolution pixel size of 0.5685 Å[86]. The electron dose was 9.392 e−/Å/s (1.0318 e−/Å2/frame) at the detector level and the total accumulated dose was 51.59 e−/Å2.

**Cryo-EM data processing**. A total of 3576 movie stacks were collected, motion-corrected, and binned to 1.137 Å/pixel using MotionCor2 in RELION3.1[87], and CTF-corrected using Ctffind 4.1.13[88] (See Supplemental Fig. S9). Micrographs with a Ctffind reported resolution estimate worse than 5 Å were discarded. A particle set generated from manual picking and template-based auto picking in RELION3.1 was cleaned and processed to 60,803 particles representing diverse views of the SWELL1 particle. These particles were then used to train Topaz[89] to pick a set of 936,282 particles. This set was cleaned with 2D classification and heterogeneous refinement in cryoSPARCv2. We then generated a refinement in cryoSPARCv2 and then RELION3.1 using C6 symmetry and utilized this map to perform Bayesian Polishing. Polished particles were then refined in RELION3.1 with C6 symmetry with a mask for the extracellular domain (ECD), transmembrane, and linker domains of SWELL1 to 3.05 Å. This map did not show clear evidence of drug density in the ECD, which we hypothesized could be due to a combination of partial drug occupancy and asymmetric drug density (of the symmetry axis of SWELL1). To test this hypothesis we performed symmetry expansion (in C6) followed by sequential 3D classification with C1 symmetry in RELION3.1 using an increasingly tightened mask on the ECD. We noted and selected classes with putative drug density in the ECD. We then used C1 refinement with C6 symmetry relaxation[51] in RELION3.1 to further refine the density. These refinement angles were used for one additional 3D classification job to generate two classes: one with vertical density (85,831 particles) and one with tilted density (78,324 particles). The particles in each of these classes were then refined an additional time with symmetry relaxation and local angular sampling. Finally, these angles were used in a refinement in C1 with masking for the ECD, transmembrane, and linker domain to generate the final maps.

**Modeling and refinement**. Maps from the refinement for the vertical and tilted drug density were used for modeling in Coot[90]. First, the model[31] (PDB: 6NZW, https://www.rcsb.org/structure/6NZW) was docked in the map density. As the lipid mix used in this study contained a majority of DOPE lipid, the lipid acyl chains between SWELL1 subunits were modeled as DOPE. The model and restraints for SN-407 were generated using Phenix.elbow[91] and then SN-407 was placed and refined in the putative drug density for each map. Realspace refinement of each model was carried out using Phenix.real_space_refine. Molprobity[92] was used to evaluate the stereochemistry and geometry of the structure for subsequent rounds of manual adjustment in Coot and refinement in Phenix. Phenix.mtriage was then used map and model validation. Figures were prepared using Chimera 1.14[93], ChimeraX 1.0[94], Prism, and Adobe Photoshop and Illustrator software.

**Quantitative RT-PCR (qRT-PCR)**. 3T3-F442A preadipocytes cells were treated with either vehicle (DMSO) or 10 μM SN-401 for 96 h and solubilized in TRIzol reagent (Invitrogen). For differentiated adipocytes, vehicle or 10 μM SN-401 was added during 7–11 days of differentiation for 96 h and then serum-starved (+ DMSO/SN-401) for 6 h and stimulated with 0 and 10 nM insulin/serum-containing media (+ DMSO or SN-401) for 15 min and solubilized in TRIzol reagent. RNA samples were isolated using the PureLink RNA kit (Life Technologies) as per the manufacturer's instructions. Complementary DNA (cDNA) synthesis was carried out using an iScript Reverse Transcriptase kit (Bio-Rad). Power SYBR green PCR master mix (Applied Biosystems) containing 0.5 μl of cDNA (per each well) from the above step was used for the qRT-PCR reaction set up and the amplification curves were observed in StepOnePlus Real-Time PCR instrument (Applied Biosystems). Each sample measurement was carried out in triplicate. GAPDH was used as the internal standard and the fold change was calculated relative to GAPDH was normalized to vehicle group. The following qRT-

PCR primers (5′ to 3′) were used: GAPDH forward—TGCACCACCAACTGCTTAG and reverse— GATGCAGGGATGATGTTC; SWELL1 forward—AGCCACAACAA CCTGACCTT and reverse— TTGTTGCCCAGGTGTAGAGC; LRRC8b forward— CGAAGGCATTCCTCGGATCA and reverse— GGTATGCTTCAATCGGGAGGT; LRRC8c forward—TCCTTTTCTGCGGATACCCT and reverse—AACTCGGTCAC CGGAATCAT; LRRC8d forward—ATGGAGGAGTGAAGTCTCGC and reverse— GCAACTTCCGCAAGGGTAAA; LRRC8e forward—CGGAGTTCCATCCCTGA GCA and reverse— CACCAGGGTTTGAGCACCTT, respectively.

**Liver isolation, triglycerides, and histology**. HFD mice treated with either vehicle or SN-401 were anesthetized with 1–4% isoflurane followed by cervical dislocation. Gross liver weights were measured and identical sections from the right medial lobe of the liver were dissected for further examinations. Total triglyceride content was determined by homogenization of 10–50 mg of tissue in 1.5 ml of chloroform:methanol (2:1 v/v) followed by centrifugation at 16,260×g for 10 min at 4 °C. A 20 μl of the supernatant was evaporated in a 1.5 ml microcentrifuge tube for 30 min. Triglyceride content was determined by adding 100 μl of Infinity Triglyceride Reagent (Fisher Scientific) to the dried sample followed by 30 min incubation at RT. The samples were then transferred to a 96-well plate along with standards (0–2000 mg/dl) and absorbance was measured at 540 nm and the final concentration was determined by normalizing to tissue weight. For histological examination, liver sections were fixed in 10% zinc formalin and paraffin-embedded for sectioning. Hematoxylin and eosin (H&E) stained sections were then assessed for steatosis grade, lobular inflammation, and hepatocyte ballooning for nonalcoholic fatty liver disease (NAFLD) scoring[46,95,96].

**Quantification and statistical analysis**. Standard unpaired or paired two-tailed Student's t-test were performed while comparing two groups. One-way ANOVA was used for multiple group comparison. For GTTs and ITTs, a two-way analysis of variance (ANOVA) test was used. Data were presented as mean ± SEM. The threshold for significance was 0.05 for all statistical comparisons. *, **, and *** represent p values of <0.05, <0.01, and <0.001, respectively. Details of statistical analyses are presented in the figure legends.

All data analyses and statistical tests were carried out in GraphPad Prism 7.0 and Microsoft Excel (v2111).

**Reporting summary**. Further information on research design is available in the Nature Research Reporting Summary linked to this article.

## Data availability
The cryo-EM data generated in this study have been deposited in public databases. For SWELL1:SN-407 in MSP1E3D1 nanodiscs pose 1, the final model is in the PDB under 7M17 and the final map is in the Electron Microscopy Data Bank (EMDB) under EMD-23614. For SWELL1:SN-407 in MSP1E3D1 nanodiscs pose 2, the final model is in the PDB under 7M19 and the final map is in the Electron Microscopy Data Bank (EMDB) under EMD-23616. The original micrograph movies and final particle stacks are in the EMPIAR database under EMPIAR-10662. [1]H NMR spectra for all compounds are provided in the supplementary information. All requests for resources, reagents, and additional information that support the findings should be addressed to rajan.sah@wustl.edu and will be fulfilled upon reasonable request. Source data are provided with this paper.

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

## Acknowledgements

We thank K. Coleman, M. Knudson, and M. Rupe of the University of Iowa Tissue Procurement Core facility (TPC) for services provided related to the acquisition of human adipose specimens (NCI award number P30CA086862). We thank the Diabetes Research Core at the Washington University in St. Louis for triglyceride estimation DRC (NIH P30 DK 020579). We also thank the Fraternal Order of Eagles Diabetes Research Center at the University of Iowa for performing euglycemic hyperinsulinemic clamps and Michael Wright (University of Iowa) for gifting HEK-293 cells. P.R.C. and C.A.K. acknowledge the support of predoctoral fellowships from the University of Iowa Center for Biocatalysis and Bioprocessing affiliated with the NIH-sponsored Predoctoral Training Program in Biotechnology (T32 GM008365). This work was supported by the John L. & Carol E. Lach Chair in Drug Delivery Technology (R.K.), grants from the New York Stem Cell Foundation, NIGMS grant no. GM123496, a McKnight Foundation Scholar Award, a Rose Hill Innovator Award, and a Sloan Research Fellowship (S.G.B.), an NIGMS grant no. GM128263 (D.M.K.), NIH grants P30 DK056341 (Washington University Nutrition and Obesity Research Center), UL1 TR000448 (Washington University Institute of Clinical and Translational Sciences), and T32 HL130357 (Pre- and Postdoctoral Training Program in Obesity and Cardiovascular Disease), NIH NIDDK R01DK115791 (A.W.N.). NIH NIDDK R01DK106009, R01DK126068, R01DK127080, Leadership Entrepreneurship Acceleration Program (LEAP) from the Skandalaris Center for Interdisciplinary Innovation and Entrepreneurship at Washington University in St. Louis, the Roy J. Carver Trust, University of Iowa (R.S.), and grants from NIH NIDDK R43 DK121598 and R44 DK126600 (D.J.L.).

## Author contributions

Conceptualization, R.S.; methodology, S.K.G., L.X., J.H., C.K., A.K., P.R.C., J.M., M.E.-H., D.M.K., J.H., P.K., E.E.G., C.A.K., W.J.G., R.D.S., E.B.T., C.M.-T., G.I.S., J.W.B., S.K., S.G.B., Y.Z. and R.S.; formal analysis, R.S., S.K.G., L.X., C.K., D.M.K., J.H., P.R.C., J.H., J.M., M.E.-H., E.E.G., R.D.S., S.G.B., W.J.G., C.M.-T. and D.J.L.; investigation, R.S., S.K.G., L.X., C.K., J.M., P.R.C., J.H., P.K., J.H., R.K., W.J.G., C.M.-T.; resources, J.K.S., I.S., P.N., Y.I., S.K., R.K. and R.S.; writing (original draft), R.S.; writing (review and editing), R.S., S.K.G., A.W.N., Y.I., D.M.K., S.G.B., W.J.G., L.X., J.M., P.R.C., R.K. and D.J.L.; visualization, R.S., S.K.G., L.X., D.M.K., A.K., J.M., P.R.C., J.H., A.K. and C.K.; supervision, R.S., A.W.N., S.G.B., Y.I. and R.K.; funding acquisition, R.K., S.G.B. and R.S.

## Competing interests

R.S. is co-founder of Senseion Therapeutics, Inc., a start-up company developing SWELL1 modulators for human disease. D.J.L. is co-Founder and CEO of Senseion Therapeutics, Inc. The remaining authors declare no competing interests.

## Additional information

**Peer review information** *Nature Communications* thanks ShengPeng Wang, Wei Lu and the other anonymous reviewer(s) for their contribution to the peer review this work. Peer reviewer reports are available.

