## [Peer Review File · Nature Communications]

Reviewers' Comments:

Reviewer #1:

Remarks to the Author:

The SWELL1/LRRC8 forms a volume regulated anion channel (VRAC). Dysfunction of SWELL1 down-regulates insulin signaling in adipose and skeletal muscle, as well as insulin secretion from pancreatic beta cells, both of which are characteristic for type 2 diabetes (T2D).

In this manuscript, Gunasekar et al. investigated the effects of small molecule SWELL1 inhibitor SN-401 (previously known as DCPIB) and its congeners on the expression level of SWELL1 in adipose and pancreatic beta cells in murine and human T2D, insulin signaling, and insulin secretion. They showed that the SWELL1 currents and protein are reduced in murine and human adipocytes and pancreatic beta cells in the setting of T2D and hyperglycemia. In vivo, SN-401 and its congeners restored SWELL1 expression and plasma membrane trafficking, improved insulin sensitivity and insulin secretion in murine T2D at concentrations one order of magnitude lower than the IC50. The cryo-EM structure of SWELL1 in complex with SN-401 has been previously reported, which showed that SN-401 inhibits SWELL1 by blocking the restriction site in the extracellular domain. The authors performed molecular docking based on the published structure and proposed two key interactions that are important for the binding of SN-401: one electrostatic interaction between the SN-401 carboxylate group and R103, one hydrophobic interaction between the SN-401 cyclopentyl group and a cleft in the intersubunit interface. The congeners, which have subtle molecular differences relative to SN-401, are predicted to either enhance or delete on-target activity. This is validated by functional studies and cryo-EM structures of SWELL1 in complex with one of the derivatives (SN-407).

Overall it's an interesting and important study which demonstrated the potential of small molecular drugs targeting SWELL to treat metabolic disorders. Below are my comments, mainly focusing on the structure part based on my expertise.

- Line 157 and Fig. 1g, should it be lean T2D instead of lean?
- Lines 223-225. "Similarly, SN-401 increases SWELL1 expression in adipose tissue of T2D KKAy mice to levels comparable to both non-T2D C57/B6 mice and to the parental KKAy parental strain (Fig. 3b)." Please provide quantification.
- When comparing the inhibitory effect of SN-401 and its derivatives, the authors need to provide the IC50 values rather than making conclusion based on a single concentration. It's possible that the two "inactive" derivatives are actually active and just require a higher concentration to be active.
- The ligand densities are poorly resolved even after extensive classification and refinement (Fig. 5f), it is therefore impossible to unambiguously defined the exact pose of SN-407. The placements of both poses of SN-407 in Fig. 5f look arbitrary. Although this is expected given the location of the binding site, the authors should clearly state the limitation of the cryo-EM structure in the manuscript. For the same reason, it is important to validate the interaction using functional studies. At a minimum, the authors should test if SN-401 and its derivatives still inhibit mutants of R103. I also recommend the authors improve Fig. 5f, and label the residues involved in the hydrophobic interaction.
- Lines 360-363: The authors proposed that the available cryo-EM structures of SWELL1 represent a closed state, in other words, it does not conduct chloride ions. They further speculated that active SN-40x bind to this closed state with higher affinity, in contrast to an "open state" that is still unknown. However, the sizes of SN-40x are substantially larger than the one of a chloride ion. The authors need to clarify why this closed state can accommodate a large organic compound in its restriction site, while not allowing much smaller chloride ions to permeate.
- Lines 405-406 (Fig. 6o, p): A sample size of n=2 is not acceptable, and the error bar is missing. I also recommend adding one or two concentrations so the fitting is more reliable.

Reviewer #2:

Remarks to the Author:

Minor Revision

Recently, I have reviewed the manuscript entitled "Small molecule SWELL1-LRRC8 complex induction improves glycemic control and nonalcoholic fatty liver disease in murine Type 2 diabetes", (NCOMMS-21-01777-T), in this paper, the author summarized SN-40X compounds and SWELL1-LRRC8 complexes, their mechanism, and use for type 2 diabetes and nonalcoholic fatty liver disease. The paper also concerned about different strategies for treating diabetes chosen by the big pharmaceutical company. In principle, I think this paper is well written and documented, design idea is novel, and merit the interest of Bioorganic & Medicinal Chemistry. So I recommend publication of this paper in the journal. However, before acceptance, some suggestions and advice should be reconsidered by the author.

1. Targeted activity is the key to proving the effectiveness of your compounds, how the SN-40X compounds bind to the SWELL1-LRRC8 complexes. The authors need to add the link to convince readers.

2. Since your idea is to reduce the risk of NAFLD by decreasing insulin resistance, why did not refer any words related to the relationship in the article? Only experimental results is not enough of the problem.

Reviewer #3:

Remarks to the Author:

Comments to the Author

The authors clearly demonstrated a role of SWELL1-LRRC8 in pathophysiological conditions in vivo. In T2D mice and humans, SWELL1-LRRC8 protein expression was reduced in adipocytes, associated with decreased insulin signaling. In T2D mouse and human islets, reduced ICI, SWELL as well as decreased SWELL1-LRRC8 protein was also observed. The authors identified that SN-401 unexpectedly increased the protein expression of SWELL1 through post-transcriptional mechanisms. By systemic administration of SN-401, they demonstrated that restoration of SWELL1-LRRC8 protein improved systemic insulin sensitivity and secretion, tissue glucose uptake, and nonalcoholic fatty liver disease in murine T2D models. Moreover, using SN-40X, rationally-designed active derivatives of SN-401, the authors confirmed the therapeutic effect of stabilizing SWELL1-LRRC8 in murine T2D models.

Since the results of the experiments are considered to be very solid and interesting, the current study provides a potentially new insight into drug discovery for diabetes. However, several questions remain to be elucidated as to the results they observed.

[Major Criticisms]

(1) The authors presented that overexpression of SWELL1 and restoration of it promoted glucose uptake via enhanced insulin signaling in adipocytes. In relation to Figure 3 or 4, does transient restoration of SWELL1-LRRC8 protein in T2D models cause specific changes in adipose tissue? (for example, fat mass, adipocyte size, adipokines such as adiponectin and leptin, and inflammatory cytokines)

(2) In Figure 4e-i, significant reductions were observed in hepatic steatosis by administration of SN-401 for 5 weeks. One of the causes of decreased hepatic fat accumulation seems to be decreased substrate-driven hepatic lipogenesis due to enhanced glucose uptake in adipose tissue and the skeletal muscle. The authors should refer to the state of hepatic lipogenesis (e.g.: lipogenic gene expression levels such as ChREBP, FAS, ACC, and SREBP1c) and/or fat oxidation.

(3) In Figure 4a-d, the data from euglycemic hyper-insulinemic clamp test suggest that hepatic insulin signaling was ameliorated in T2D mice by administering SN-401. Is this a direct or secondary change due to the SN-401 administration? My concerns about the role of SWELL1-

LRRC8 protein in hepatic insulin signaling are as follows:

3-1) Does knockdown or overexpression of SWELL1-LRRC8 affect insulin signaling in primary hepatocyte or hepatocyte cell lines?

3-2) In the first place, is the expression of SWELL1-LRRC8 decreased in T2D mouse models as was seen in islets and adipocytes?

(4) In Fig.1, the authors showed that ICI, SWELL and SWELL1-LRRC8 proteins were reduced in T2D mouse and human islets. They previously reported that SWELL1-LRRC8 channel activity (conductive signaling) in the pancreatic β -cell was required for normal insulin secretion. In relation to Figure 3k-m, how does SN-401 improve insulin secretion in beta cells? Is that due to a change in channel activity or an increase in SWELL1-LRRC8 protein, or both?

[Minor Criticisms]

(1) SWELL1-LRRC8 protein seems to be expressed in the brain. Does administration of SN-401 affect food intake?

(2) The reductions in total Akt2 protein expression as well as phosphorylation levels were observed in SWELL1 KO adipocytes. How do the authors think SWELL1 regulates Akt expression?

(3) Regarding Figure 3a, there seem to be a lot of variations. I would like to see the uncropped images.

(4) There was the misspelling in line 203.

Reviewer #4:

Remarks to the Author:

The manuscript by Gunasekar et al contains interesting and novel information that, if adequately supported, would be a substantial advance. There are, however, a number of problems, some serious. A major conceptual issue is that the conclusion "SWELL1 modulators restore SWELL1-dependent insulin-sensitivity and insulin secretion in Type 2 diabetes" is not adequately supported by in vivo validation. The authors claim that testing SN-40X compounds in global SWELL1^{-/-} mice is not possible due to embryonic lethality, but the tissue specific SWELL1 KO mice was already used in their previous publications. In addition, the authors should further analyze how is SWELL1 activated in the context of Type 2 diabetes, what is the mechanism by which rationally-designed active derivatives SN-40X regulate the expression of SWELL1, how is this physiologically regulated and what are the implications? Furthermore, the authors state that SN-40X induced SWELL1 expression is not related to mRNA but rather post-transcriptional modifications, does SN-40X affect SWELL1 protein ubiquitination? Last, since SN-40X not only acts as an inhibitor of SWELL1 but also regulates SWELL1 expression, how these events are mechanistically linked/distinguished is not explained well. Some of the more specific questions and issues are pointed out below.

1. SWELL1 has recently been proposed to be involved in regulating adipocyte size, does SN-40X compounds have an effect on adipocyte size?
2. Inhibitor of the volume-regulated anion current, such as 4-[(2-Butyl-6,7-dichloro-2-cyclopentyl-2,3-dihydro-1-oxo-1H-inden-5-yl)oxy]butanoic acid (DCPIB) should be included in the characterization of ICI, SWELL.
3. Housekeeping genes such as GAPDH or β -actin were used for loading controls in all western blot analysis; some of the key analyses should be further re-evaluated with Ponceau S staining.
4. In Fig.1a, the expression of SWELL1 is missing.
5. In Fig.1g, is there a difference between obese non-T2D vs obese T2D?
6. In Fig.1h-i, since fat tissues were used, the specific cell type(s) that responsible for reduced expression of SWELL1 is not clear.
7. What is the rationale of using 3T3F442A pre-adipocyte for expression and functional analysis?
8. In several of the western blot expression analysis, re-probing of total protein in the same blot is missing. AKT2 expression in Fig.2f is missing. SWELL1 expression in Fig.2i is missing.

9. The effect of SN-40X on body weights seems different in Fig.4f vs Table S5, it is critical to resolve this discrepancy.
10. In Fig.8b and 8c, the control groups were all treated with 12 weeks HFD, why the glucose level is not comparable?
11. The phosphorylation sites for AKT2, AS160, eNOS are not mentioned.
12. Even though eNOS is a downstream target of AKT in HUVECs, what is the physiological rationale of studying eNOS phosphorylation here?
13. In general, sample size in many experiments is too small, for example, n=2 in Fig.2g, Fig.6j, Fig.6o; n=3 in Fig.2i, Fig.5b, Fig.6f, Fig.7a; n=4 in Fig.5c, Fig.6h. Likewise, the use of n=5-7 mice per group is on the low end as there is inherent variability in these in vivo models.

Responses to Reviewer's Comments

We thank the Reviewers for their careful reading and evaluation of the manuscript, and their suggested revisions. The comments were overall positive and we focused our revisions on the Reviewers' major concerns. Accordingly, we expanded the molecular docking to occupy an entire Figure 5. We included further functional assessments by patch-clamp to quantify SN-401, SN-406, SN-407 and Inactive compound binding to the wild-type channel in HEK-293 cells by measuring IC50s in new **Fig. 6**. We also introduced a new Inactive molecule, Inactive 3 that directly addressed the proposed binding based on docking and cryo-EM, and also mutagenesis experiments to alter the electropositivity of the pore, or selectivity filter, and then evaluated SN-401 and SN-407 binding. Finally, we performed a deeper analysis of adipose tissue by evaluating adipocyte size, secreted adipokines and hepatokines, and addressed a number of technical comments. We appreciate the Reviewers' comments as we feel that addressing them has improved the manuscript. Further details and itemized responses are described below:

Reviewer#1 (Remarks to Author):

"The SWELL1/LRRC8 forms a volume regulated anion channel (VRAC). Dysfunction of SWELL1 down-regulates insulin signaling in adipose and skeletal muscle, as well as insulin secretion from pancreatic beta cells, both of which are characteristic for type 2 diabetes (T2D).

In this manuscript, Gunasekar et al. investigated the effects of small molecule SWELL1 inhibitor SN-401 (previously known as DCPIB) and its congeners on the expression level of SWELL1 in adipose and pancreatic beta cells in murine and human T2D, insulin signaling, and insulin secretion. They showed that the SWELL1 currents and protein are reduced in murine and human adipocytes and pancreatic beta cells in the setting of T2D and hyperglycemia. In vivo, SN-401 and its congeners restored SWELL1 expression and plasma membrane trafficking, improved insulin sensitivity and insulin secretion in murine T2D at concentrations one order of magnitude lower than the IC50. The cryo-EM structure of SWELL1 in complex with SN-401 has been previously reported, which showed that SN-401 inhibits SWELL1 by blocking the restriction site in the extracellular domain. The authors performed molecular docking based on the published structure and proposed two key interactions that are important for the binding of SN-401: one electrostatic interaction between the SN-401 carboxylate group and R103, one hydrophobic interaction between the SN-401 cyclopentyl group and a cleft in the intersubunit interface. The congeners, which have subtle molecular differences relative to SN-401, are predicted to either enhance or delete on-target activity. This is validated by functional studies and cryo-EM structures of SWELL1 in complex with one of the derivatives (SN-407).

Overall it's an interesting and important study which demonstrated the penitential of small molecular drugs targeting SWELL to treat metabolic disorders. Below are my comments, mainly focusing on the structure part based on my expertise."

1. Line 157 and Fig. 1g, should it be lean T2D instead of lean?

This is now **Fig. 1h**. No, this is Lean and non-T2D. We are comparing among Lean non-T2D, Obese non-T2D and Obese-T2D. The pattern of SWELL1 current densities are similar between human adipocytes and murine adipocytes measured previously by Inoue et. al and shown in **Fig. 1g** (old **Fig. 1f**). Higher in obese, relatively insulin sensitive mice and humans and lower in obese, insulin resistant mice and humans (**Fig. 1g-k**).

2. **Lines 223-225. “Similarly, SN-401 increases SWELL1 expression in adipose tissue of T2D KKAY mice to levels comparable to both non-T2D C57/B6 mice and to the parental KKAa parental strain (Fig. 3b).” Please provide quantification.**

Thank you for pointing this out. This was a weakness that we have significantly improved. We have not only provided quantification, but we repeated the experiment with a more appropriate sample size comparing vehicle (n = 4) to SN-401 (n = 5) treatment in male KKAY mice (**Fig. 3b**) We also provide quantification, not only to beta actin, but also to total protein by normalizing to Ponceau S staining. This also addressed another Reviewer’s comment regarding normalization to Ponceau stain and increasing sample sizes in some experiments (see below). We have provided a similar quantification for vehicle versus SN-401 treated HFD mice in **Fig. 3a**.

3. **When comparing the inhibitory effect of SN-401 and its derivatives, the authors need to provide the IC₅₀ values rather than making conclusion based on a single concentration. It’s possible that the two “inactive” derivatives are actually active and just require a higher concentration to be active.**

This was a very astute comment and thank you pointing this out. We agreed wholeheartedly and spent additional time and effort to very carefully measure dose-responses and generate IC₅₀ for SN-401, the SAR derived compounds with higher efficacy (SN-406 and SN-407). We also did the same for “Inactive” compounds Inactive 1, Inactive 2, and designed and tested a new compound: Inactive 3. We also examined a SWELL1 mutant for IC₅₀ as described below. With these expanded measurements we split the older figure to create space for these more detailed and mechanistically informative series of experiments. As you can see in new **Fig. 6a-e** the Inactives are not *entirely* Inactive but *relatively* inactive. With higher concentrations, inhibition was observed with Inactive 2, and less so with Inactive 1 and 3. We were able to fit these curves and derive IC₅₀ to rank them from most to least inactive. In some cases, compound solubility limited our ability to explore higher concentrations, with Inactive 1 and 3. By doing the same for the actives, SN-406 and SN-407 we were able to demonstrate a shift in the IC₅₀ to the left (new **Fig. 6f-i**), consistent with improved binding, as suggested by the docking models (**Fig. 5**).

4. **The ligand densities are poorly resolved even after extensive classification and refinement (Fig. 5f), it is therefore impossible to unambiguously defined the exact pose of SN-407. The placements of both poses of SN-407 in Fig. 5f look arbitrary. Although this is expected given the location of the binding site, the authors should clearly state the limitation of the cryo-EM structure in the manuscript.**

We fully agree with the reviewer that while our models represent the best fits of the model to experimental maps and are consistent with other experimental and modeling data in the manuscript, there remains some ambiguity in the exact pose of SN-407. We have added a sentence to the text to convey this more clearly (lines 391-395).

5. **For the same reason, it is important to validate the interaction using functional studies. At a minimum, the authors should test if SN-401 and its derivatives still inhibit mutants of R103.**

Again, we agreed and set-out to replace the positively charged R103 in SWELL1 with a negatively charged glutamate R103E. We did this to determine if reducing the

electropositivity of this putative binding site would also destabilize SN-40X binding via the negative carboxylate. We overexpressed SWELL1-R103E on a WT HEK-293 cell background to competitively replace the WT SWELL1 R103 with the mutant R103E. We found that the SWELL1-R103E currents were 84% lower than the WT currents, consistent with a dominant negative effect on overall anion channel function, as the electropositivity of the constriction point in the pore of the channel is reduced with the introduction of electronegative glutamates at position R103. Consistent with both docking and cryo-EM data, the R103E currents were more resistant to inhibition by either SN-401 or SN-407, as demonstrated by the rightwardly shifted IC_{50} in the R103E mutant cells (new Fig. 6j-p).

Beyond this, we also approached this question using a medicinal chemistry approach. We eliminated the electronegative carboxylate from SN-401, while leaving everything else identical by replacing with carboxylate group with a boronic acid group, yielding Inactive 3 (Fig. 5a). At physiological pH the boronic acid group is expected to be uncharged. Remarkably this charge specific modification of SN-401 was sufficient to shift the IC_{50} from 3.9 to 18.3 μ M (new Fig. 6d&e). These convergent data strongly support R103 as an important binding site for SN-401 and SN-40X binding.

6. I also recommend the authors improve Fig. 5f, and label the residues involved in the hydrophobic interaction.

Thank you for this suggestion. We have now improved the figure (now new Fig. 7) with an additional view looking down the channel and clearer labeling. We do not state that there are any specific hydrophobic interactions, rather, hydrophobic portions of DCPIB/SN-401 (cyclopentyl and butyl chain) occupy a hydrophobic cleft or pocket formed by main and side chains of His-104, Tyr-106 and Asp-102. Also, for analogs with longer chains, the C6 or C7 chains of the analogs can interact with the side chain of arg103. The hydrophobic interactions function by improving binding energy for small molecules and receptors by kicking out water molecules out of the hydrophobic areas. In this way, there are no specific residues that form hydrophobic interactions, for example, a finite interaction with a specific atom on an amino acid. Instead, these regions create a hydrophobic nano-environment.

7. Lines 360-363: The authors proposed that the available cryo-EM structures of SWELL1 represent a closed state, in other words, it does not conduct chloride ions. They further speculated that active SN-40x bind to this closed state with higher affinity, in contrast to an “open state” that is still unknown. However, the sizes of SN-40x are substantially larger than the one of a chloride ion. The authors need to clarify why this closed state can accommodate a large organic compound in its restriction site, while not allowing much smaller chloride ions to permeate.

The electropositive constriction site at R103 is thought to attract negatively charged anions through the pore of the SWELL1-LRRC8 channel acting as a selectivity filter. These anions can be chloride ions, larger iodide ions, or even glutamic acid. So relatively large anions can permeate the SWELL1-LRRC8 ion channel. This is consistent with a carboxylate group from SN-40X reaching up to the R103 constriction site and then stabilized by R103-carboxylate electrostatic interactions on one end, and hydrophobic interactions on the other end between LRRC8 subunits (preventing further permeation of the drug into the channel). This is experimentally resolvable in the cryo-EM structure of SN-401 (Kern et al., 2019) and SN-407 bound to SWELL1 homomer (new Fig. 7), is predicted in all of our docking models (new Fig. 5), and also consistent with our functional experiments (new Fig. 6 and 8). Importantly, we do not think that the physical constriction at the selectivity filter/drug binding site provides the mechanism of channel closure, but rather that it is the N-terminus projecting into the lumen of

the pore (cytoplasmic to R103) that has been shown (Zhou, Polovitskaya and Jentsch 2018) or hypothesized in other work (Deneka et al., 2018; Kasuya et al., 2018; Kefauver et al., 2018; Kern et al., 2019) to be involved in channel gating. This region is not modeled in the structure, presumably because it adopts different conformations between particles. For this reason, although we cannot say definitively from the structure alone whether it represents a closed or open state, since we prepared the samples in 150 mM KCl (and not low ionic strength which independently opens the channel (Syeda et al., 2016)), we can only make a reasonable assumption from functional work that the structure represents a closed conformation.

- 8. Lines 405-406 (Fig. 6o, p): A sample size of n=2 is not acceptable, and the error bar is missing. I also recommend adding one or two concentrations so the fitting is more reliable.**

We agree with the comment and have revised this figure (**new Fig. 8o,p**) with a larger samples size to demonstrate this effect.

Reviewer#2:

“Minor Revision

Recently , I have reviewed the manuscript entitled “Small molecule SWELL1-LRRC8 complex induction improves glycemic control and nonalcoholic fatty liver disease in murine Type 2 diabetes”,(NCOMMS-21-01777-T) ,in this paper, the author summerrised SN-40X compounds and SWELL1-LRRC8 complexes, their mechanism , and use for type 2 diabetes and nonalcoholic fatty liver disease. The paper also concerned about different strategies for treating diabetes choosed by the big pharmaceutical company. In principle, I think this paper is well written and documented, design idea is novel, and merit the interest of Bioorganic & Medicinal Chemistry. So I recommend publication this paper in the journal. However, before acceptance, some suggestions and advice should be reconsidered by the author.”

We thank the Reviewer for recommending acceptance and publication of this paper.

- 1. Targeted activity is the key to proving the effectiveness of your compounds, how the SN-40X compounds bind to the SWELL1-LRRC8 complexes. The authors need to add the link to convince readers.**

We agree that targeted activity is important to demonstrate for a novel drug class. Accordingly, much of the revised manuscript addresses specifically this concern.

First, we demonstrate that SN-401 mediated induction of AKT2-AS160 signaling in 3T3-F442A adipocytes requires SWELL1, as this effect is markedly abrogated in the SWELL1 KO 3T3-F442A cells, and thus consistent with a SWELL1 targeted and dependent mechanism. Similarly, in new **Supplementary Fig. 10**, we demonstrate the same SWELL1-dependent SN-401 mediated effects with human umbilical vein endothelial cells (HUVEC) using SWELL1 knock-down approaches. These data supplement **Fig. 8k&l** and **Fig. 8o&p** with SN-406 and SN-401. Finally, in **Fig. 9c-g**, we demonstrate that SN-401 is able to prevent glucolipototoxicity-mediated reductions in glucose-stimulated insulin secretion in both human (**Fig. 9d-e**) and murine (**Fig. 9f&g**) islets, also in a SWELL1-dependent manner.

Second, in the process of developing the structure-activity relationship delineated in **Fig. 5-7**, we were able to develop SN-401 small molecule congeners that either augment or reduce SWELL1 on-target activity. We then used these compounds to further test the question of on-target activity by comparing the efficacy of active compounds (SN-401, SN-406, SN-407) as compared to

Inactive compounds (Inactive 1, Inactive 2) on channel binding to open (Fig. 6), and closed SWELL1-LRRC8 channel states (Fig. 8a-d) by patch-clamp recordings. Next, we compared active compounds (SN-401, SN-406) and Inactive compounds for their ability to augment SWELL1 protein (Fig. 8e-h), SWELL1 plasma membrane localization (Fig 8i&j, Supplementary Fig. 9), AKT2-eNOS signaling in HUVECs (Fig. 8k-n), and to prevent glucolipotoxic reductions in SWELL1 protein (Fig. 9a&b). In every case, the SWELL1 active SN-40X compounds demonstrated activity in these *in vitro* assays while the “negative control” Inactive compounds did not – consistent with SWELL1 on-target activity. We then extended these experiments *in vivo* in Fig. 10a&c-g to examine how negative control compound Inactive 1 compared to active SN-40X compounds for improving glycemic control (Fig. 10a,c,f), and glucose-stimulated insulin secretion (Fig. 10e&g), thereby controlling for off-target effects still present in Inactive.1 Again, we found that SN-40X performed better than Inactive 1 in all parameters measured suggesting that these SN-40X effects *in vivo* are also mediated by on-target binding.

2. Since your idea is to reduce the risk of NAFLD by decreasing insulin resistance, why did not refer any words related to the relationship in the article? Only experimental results is not enough of the problem.

We agree that further elaboration around this point is warranted. In the revised manuscript, in lines 295-299, we indicate that the “SN-401 mediated reductions in hepatic steatosis and hepatocyte damage are consistent with the observed increases in hepatic insulin sensitivity and consequent reductions in hepatic glucose production via gluconeogenesis available for hepatic *de novo* lipogenesis, as observed with other insulin sensitizers, such as metformin and TZDs.”

Reviewer#3:

“The authors clearly demonstrated a role of SWELL1-LRRC8 in pathophysiological conditions *in vivo*. In T2D mice and humans, SWELL1-LRRC8 protein expression was reduced in adipocytes, associated with decreased insulin signaling. In T2D mouse and human islets, reduced ICI, SWELL as well as decreased SWELL1-LRRC8 protein was also observed. The authors identified that SN-401 unexpectedly increased the protein expression of SWELL1 through post-transcriptional mechanisms. By systemic administration of SN-401, they demonstrated that restoration of SWELL1-LRRC8 protein improved systemic insulin sensitivity and secretion, tissue glucose uptake, and nonalcoholic fatty liver disease in murine T2D models. Moreover, using SN-40X, rationally-designed active derivatives of SN-401, the authors confirmed the therapeutic effect of stabilizing SWELL1-LRRC8 in murine T2D models. Since the results of the experiments are considered to be very solid and interesting, the current study provides a potentially new insight into drug discovery for diabetes. However, several questions remain to be elucidated as to the results they observed.”

1. The authors presented that overexpression of SWELL1 and restoration of it promoted glucose uptake via enhanced insulin signaling in adipocytes. In relation to Figure 3 or 4, does transient restoration of SWELL1-LRRC8 protein in T2D models cause specific changes in adipose tissue? (for example, fat mass, adipocyte size, adipokines such as adiponectin and leptin, and inflammatory cytokines)

In the majority of *in vivo* experiments in which we observed improved glucose tolerance, glucose tissue uptake and insulin sensitivity, we had only treated the mice for 4-10 days. During this relatively short period we did not observe any significant weight changes so we did not pursue a deeper analysis of adipose tissue characterization as described above. However, we now include a formal analysis of adipocyte size from eWAT of vehicle and SN-401 treated KKAy and C57 HFD fed mice in new Fig. 3g-h. In KKAy mice, we observed a small (9%) but statistically significant reduction in adipocyte size in SN-401 treated mice,

while in C57 HFD mice we observed no significant differences in adipocyte size. We also measured serum adiponectin and FGF-21 (new Fig. 4j-k) in SN-401 and vehicle treated KKAY mice and found no differences in adiponectin (new Fig. 4j), but remarkably FGF-21 was increased 3-fold in SN-401 treated mice (new Fig. 4k). This latter finding proved to be very interesting and may provide an additional metabolic molecular mechanism for the observed improvements in glucose metabolism, and hepatic steatosis observed in these SN-40X treated T2D models, as commented on in the Discussion (lines 610-613). We thank the Reviewer for suggesting this direction of inquiry as it has opened up a new direction for our laboratory to explore in future studies.

- In Figure 4e-i, significant reductions were observed in hepatic steatosis by administration of SN-401 for 5 weeks. One of the causes of decreased hepatic fat accumulation seems to be decreased substrate-driven hepatic lipogenesis due to enhanced glucose uptake in adipose tissue and the skeletal muscle. The authors should refer to the state of hepatic lipogenesis (e.g.: lipogenic gene expression levels such as ChREBP, FAS, ACC, and SREBP1c) and/or fat oxidation.**

We agree that a more thorough examination and characterization of hepatic lipogenesis can be performed and we plan to do this as part of another manuscript. The current manuscript is already very comprehensive and spans a number of techniques and concepts in order to rigorously examine the mechanism of action of this novel class of compounds for the treatment of T2D and metabolic syndrome. Metabolic syndrome is by nature a highly complex and pathophysiologically integrated disease process and full evaluation of the mechanism of action of these drugs will involve further studies in liver, adipose, skeletal muscle and islet cells – as well as possible cross-talk between these tissues via adipokines, and hepatokines, as the FGF-21 result suggests (new Fig. 4k). As much as we would like to answer all outstanding questions in a single manuscript, at some point this become unfeasible, and examining additional mechanism(s) is best reserved for future papers.

- Figure 4a-d, the data from euglycemic hyper-insulinemic clamp test suggest that hepatic insulin signaling was ameliorated in T2D mice by administering SN-401. Is this a direct or secondary change due to the SN-401 administration? My concerns about the role of SWELL1-LRRC8 protein in hepatic insulin signaling are as follows:
3-1) Does knockdown or overexpression of SWELL1-LRRC8 affect insulin signaling in primary hepatocyte or hepatocyte cell lines?
3-2) In the first place, is the expression of SWELL1-LRRC8 decreased in T2D mouse models as was seen in islets and adipocytes?**

The Reviewer raises an excellent question since hepatic insulin sensitivity, best described by its ability to suppress hepatic glucose production (Fig. 4b), can occur via endogenous, direct (i.e. liver-autonomous) or exogenous, indirect effects on liver insulin signaling and glucose metabolism. In fact, if adipose is behaving differently (which it is), there will certainly be indirect effects *in vivo*. This is nicely reviewed in a recent paper by Lewis, GF et al. (Lewis, GF, Cell Metab 2021). Since SWELL1 is expressed in multiple tissues important for regulating systemic glucose homeostasis, and SN-40X series may be acting on many, if not all, of these tissues, it will be very difficult to answer this question within the scope of the current manuscript. We suspect that testing SN-40X effects on hepatic *gluconeogenesis in vivo* using euglycemic hyper-insulinemic clamp in liver SWELL1 KO mice as compared to tissue-specific SWELL1 KO of peripheral tissues (adipose/skeletal muscle) may best address this question in a separate manuscript.

4. **In Fig.1, the authors showed that ICI,SWELL and SWELL1-LRRC8 proteins were reduced in T2D mouse and human islets. They previously reported that SWELL1-LRRC8 channel activity (conductive signaling) in the pancreatic b-cell was required for normal insulin secretion. In relation to Figure 3k-m, how does SN-401 improve insulin secretion in beta cells? Is that due to a change in channel activity or an increase in SWELL1-LRRC8 protein, or both?**

Our working model for the mechanism of action of SN-40X is that these compounds bind SWELL1-LRRC8 to increase active complexes at the plasma membrane of multiple cell types in different tissues, and then unbind as serum levels drop (see pharmacokinetic data in Supplementary Fig. 11a-b), and this is associated with an increase in SWELL1 complex channel signaling activity. In old Fig. 3k-m and Fig. 8e&g, now new Fig. 3m-o and Fig. 10e&g, the islet perfusion experiments were performed 24 hours after the last SN-40X dose. Also, in Fig. 9d-g, in which islets were incubated under glucolipotoxic conditions (high glucose + palmitate) +/- SN-401, the SN-401 was washed out prior to the islet perfusion experiments to simulate the conditions in new Fig. 3m-o and Fig. 10e&g. Based on these data we think that increased insulin secretion in beta cells is due to increased SWELL1-LRRC8 protein and associated increases in signaling.

Minor

1. **SWELL1-LRRC8 protein seems to be expressed in the brain. Does administration of SN-401 affect food intake?**

SWELL1 is expressed in the brain (Formaggio et al., The FASEB Journal, 2018; Zhou et al., Experimental Neurology, 2020) but SN-401 does not cross the blood brain barrier (Zhang et al., Experimental Neurology, 2008). In past studies examining SN-401 in stroke prevention it was injected directly into the brain to study these effects. Accordingly, we are not concerned about off-target effects on the brain. Since there were no differences in body weight in the short-course experiments between vehicle and SN-40X compounds (Supplementary Table. S5), and no meaningful change in adipocyte size (new Fig. 3g-h), despite significant metabolic differences, we did not formally examine food intake in these mice in the current study – though this can be done in future studies.

2. **The reductions in total Akt2 protein expression as well as phosphorylation levels were observed in SWELL1 KO adipocytes. How do the authors think SWELL1 regulates Akt expression?**

The Reviewer raises a great question. This reduction in AKT2 protein in SWELL1 KO or SWELL1 KD cells appears to be at the level of mRNA expression, and has been published in multiple RNA sequencing data sets from our laboratory, including 3T3-F442A adipocytes (Zhang, Y et al., Nature Cell Bio 2017), C2C12 myotubes (Kumar, A., et al., Elife 2020) and HUVECs (Alghanem, A., et al., Elife 2021). This does remain an open question, and as described in Alghanem, A et al. (2021), we can only speculate that SWELL1/LRRC8A is somehow regulating *Akt2* and *eNOS* gene expression. The PI3K signaling pathway have been described to regulate total AKT2 expression (Fayard,E et al., Current Topics in Microbiology and Immunology, 2010; Tsuchiya, A., et al., Journal of Endocrinology, 2014), and we observe reductions in both PI3K signaling upon SWELL1 KO/KD in 3T3-F442A adipocytes (Zhang, Y et al., Nature Cell Bio 2017), C2C12 myotubes (Kumar, A., et al., Elife

2020) and HUVECs (Alghanem, A., et al., Elife 2021). Collectively, these findings across multiple cell types support a PI3K mediated mechanism of regulation AKT2 gene expression that we intend to explore in future studies.

3. Regarding Figure 3a, there seem to be a lot of variations. I would like to see the uncropped images.

We agree with the Reviewer on this point that the western blot showed marked variability and the quality could be improved. Accordingly, we reran the samples in a new gel and now show the entire blot in **Fig. 3a**.

4. There was the misspelling in line 203.

Thank you. We fixed this.

Reviewer #4:

“The manuscript by Gunasekar et al contains interesting and novel information that, if adequately supported, would be a substantial advance. There are, however, a number of problems, some serious. A major conceptual issue is that the conclusion “SWELL1 modulators restore SWELL1-dependent insulin-sensitivity and insulin secretion in Type 2 diabetes” is not adequately supported by in vivo validation. The authors claim that testing SN-40X compounds in global SWELL1-/- mice is not possible due to embryonic lethality, but the tissue specific SWELL1 KO mice was already used in their previous publications.”

We thank the Reviewer for this comment, and agree that a critical aspect of this work that makes it a substantial advance is demonstrating that these SWELL1 modulators improve insulin-sensitivity and insulin secretion in a SWELL1-dependent manner. Indeed, we have devoted a substantial amount of this revised manuscript toward addressing this point, as outlined below.

It is true, the global SWELL1 KO mice are largely embryonically lethal and those that survive are so sick and runted (Kumar et al, The Journal of Experimental Medicine, 2014) that they are not appropriate for metabolic studies. Several tissue-specific SWELL1 KO are viable, and we (Zhang, Y. et al, Nat Cell Bio 2017; Kang, C, et al. Nat Comm 2018; Kumar, A, et al, Elife 2020; Alghanem, A., et al, Elife 2021), and others (Stuhlmann, T. et. al. Nat Comm 2018) have published their metabolic phenotypes, wherein tissue-specific SWELL1 ablation impairs insulin secretion, or AKT-AS160, or AKT-eNOS signaling, depending on the tissue/cell-type targeted. However, since these small molecule SWELL1 modulators (SN-40X) likely target many different metabolically active tissues, genetic dissection of these effects through the use of tissue-specific SWELL1 KO mice is just not feasible. Accordingly, in the revised manuscript, we have used structure-activity relationship directed medicinal chemistry to synthesize relatively inactive congeners to control for off-target effects and determine the SWELL1 dependent contributions (**Fig. 5-10**). These studies included comparing SWELL1 active SN-40X compounds to Inactive 1 *in vivo*, for *in vivo* validation. We demonstrate that SWELL1 binding is required to improve glucose tolerance (**Fig. 10a-c, e, f**), insulin sensitivity (**Fig. 10d**) and insulin secretion (**Fig. 10e&g**) *in vivo*. We also demonstrate that SWELL1 is required for SN-40X mediated improvements in islet glucose-stimulated insulin secretion under glucolipotoxic conditions (**Fig. 9c-g**).

“In addition, the authors should further analyze how is SWELL1 activated in the context of Type 2 diabetes, what is the mechanism by which rationally-designed active derivatives SN-40X regulate the expression of SWELL1, how is this physiologically regulated and what are the implications?”

We thank the reviewer for raising this point. As described in the revised manuscript, we hypothesize that SWELL1 channel complexes are reduced in the setting of decompensated, hyperglycemic T2D due to glucolipotoxic stress (Fig. 9a-b), and that SN-40X augment SWELL1 channel complexes by functioning as molecular chaperones to maintain or restore SWELL1 complexes or signaling (Fig. 8) in the setting of this glucolipotoxic stress (Fig. 9c-g) and T2D (Fig. 4&10) in a SWELL1-dependent manner.

“Furthermore, the authors state that SN-40X induced SWELL1 expression is not related to mRNA but rather post-transcriptional modifications, does SN-40X affect SWELL1 protein ubiquitination?”

We thank the reviewer for raising this point. As stated above, and described in the revised manuscript, we hypothesize that SN-40X function as molecular chaperones to stabilize SWELL1 complexes under unfavorable glucolipotoxic conditions of hyperglycemic T2D. It is also possible that SWELL1 protein ubiquitination may provide another molecular mechanism, and we will evaluate this alternative mechanism in future studies.

“Last, since SN-40X not only acts as an inhibitor of SWELL1 but also regulates SWELL1 expression, how these events are mechanistically linked/distinguished is not explained well.”

We thank the reviewer for raising this point. In the revised manuscript, we hypothesize that SN-40X inhibitors are functioning as molecular chaperones, in a manner mechanistically similar to glibenclamide for dysfunctional KATP channels (Yan, FF., et al., Journal of Biological Chemistry 2004 and 2006) and to pharmacological correctors for CFTR (Pedemonte, N., et al., Frontiers in Pharmacology 2012). This concept is explained throughout the revised manuscript and the detailed molecular mechanisms explored at an atomic level through molecular docking, functional experiments and cryo-EM (Fig. 5,6,7).

1. SWELL1 has recently been proposed to be involved in regulating adipocyte size, does SN-40X compounds have an effect on adipocyte size?

To address this question, we now include a formal analysis of adipocyte size from eWAT of vehicle and SN-401 treated KKAY and C57 HFD fed mice in new Fig. 3g-h. In KKAY mice, we observed a small (9%) but statistically significant reduction in adipocyte size in SN-401 treated mice, while in C57 HFD mice we observed no significant differences in adipocyte size.

2. Inhibitor of the volume-regulated anion current, such as 4-[(2-Butyl-6,7-dichloro-2-cyclopentyl-2,3-dihydro-1-oxo-1H-inden-5-yl)oxy]butanoic acid (DCPIB) should be included in the characterization of ICI,SWELL.

We thank the reviewer for raising this point. Just to be clear, DCPIB is SN-401. So, the entire revised manuscript is a characterization of DCPIB effects on SWELL1. Also, DCPIB (SN-401) is the foundation upon which all subsequent SN-40X compounds have been designed.

3. **Housekeeping genes such as GAPDH or β -actin were used for loading controls in all western blot analysis; some of the key analyses should be further re-evaluated with Ponceau S staining.**

To address this we have now included Ponceau S staining in **Fig. 3a-b**.

4. **In Fig.1a, the expression of SWELL1 is missing.**

We agree that the associated SWELL1 protein levels in islets isolated from hyperglycemic T2D mice compared to non-T2D mice was not included in the evaluation of SWELL1 to complement the ICI,SWELL current traces shown in **Fig. 1a**. We have now provided this additional data in new **Fig. 1e** and this shows a clear 66% decrease in SWELL1 protein levels in T2D islets. We thank the Reviewer for pointing this out. Addition of this data has strengthened the association of reduced SWELL1 activity and protein levels in the setting of T2D.

5. **In Fig.1g, is there a difference between obese non-T2D vs obese T2D?**

In old **Fig. 1g**, now new **Fig. 1h**, there was not a statistically significant difference between obese non-T2D and obese T2D (p value = 0.1). However, the pattern of SWELL1 current densities in these human adipocytes was similar to previously published SWELL1 currents in adipocytes isolated from C57, KKA^a and KKA^y mice (new **Fig. 1g**), and also followed a similar pattern to the statistically significant decrease in SWELL1 protein in eWAT of humans with T2D as compared to non-T2D patients (**Fig. 1k**).

6. **In Fig.1h-i, since fat tissues were used, the specific cell type(s) that responsible for reduced expression of SWELL1 is not clear.**

This is true, and the same holds true for the SWELL1 protein quantification in islets. For this reason, direct measurement of SWELL1 current densities by whole-cell patch clamp in adipocytes from KKA^a and KKA^y mice (new **Fig. 1g** plotted from Inoue, H., et al., American journal of physiology, Cell physiology, 2010) and human adipocytes (new **Fig. 1h**) provides additional data to support a reduction of SWELL1 activity specifically in adipocytes, just as direct patch-clamp measurements of SWELL1 currents in pancreatic beta cells (**Fig. 1a-d**) supports the reductions in mouse and human islet SWELL1 protein (new **Fig. 1e&f**).

7. **What is the rational of using 3T3F442A pre-adipocyte for expression and functional analysis?**

The rationale for using 3T3-F442A pre-adipocytes in **Fig. 2f&g**, **Supplementary Fig. 2a** was simply to demonstrate that SN-401 mediated effects are present in pre-adipocytes, as well as differentiated adipocytes. This demonstrated SN-401 mediated SWELL1 induction and AKT signaling to be independent from adipocyte differentiation. The later use in **Fig. 8i&j**, **Supplementary Fig. 9** was to quantify SWELL1 immunolocalization. Preadipocytes were found to be easier to image and stain endogenous SWELL1 than lipid laden differentiated 3T3-F442A adipocytes, under the fixation conditions that we optimized for *endogenous SWELL1 immunostaining* with our custom antibody (see Methods). Similarly, these pre-adipocytes are easier to patch-clamp than differentiated adipocytes, especially after viral transduction (**Supplementary Fig. 1a-c**). This is because these cells need to be trypsinized

and replated on coverslips, and preadipocytes are far more amenable to this. This was required to validate re-expression of SWELL1 in SWELL1 KO 3T3-F442A adipocytes reconstituting both SWELL1 protein expression (Fig. 2a), plasma membrane localization (Supplementary Fig. 1d), SWELL1-mediated currents (Fig. 2c, Supplementary Fig. 1a-c) and SWELL1 plasma membrane signaling (Fig. 2a-b). These were the only instances in which we used 3T3-F442A pre-adipocytes. All other experiments: Fig. 2a, Fig. 2h-j, Fig. 8e-f, Fig. 9a-b, Supplementary Fig. 1d and Supplementary Fig. 2b-c used differentiated 3T3-F442A adipocytes, and all other data in the manuscript was primary islets, primary beta cells (mouse/human), mature primary adipocytes (mouse/human), adipose tissue (mouse/human), and primary human umbilical vein endothelial cells (HUVECs) – in addition to in vivo data.

8. In several of the western blot expression analysis, re-probing of total protein in the same blot is missing. AKT2 expression in Fig.2f is missing. SWELL1 expression in Fig.2i is missing.

Re-probing of total protein (such as pAKT2 and then AKT2) within the same blots involves stripping of the membrane and can sometimes introduce artifacts especially if the antibodies used for probing are very sensitive. Accordingly, we normalized the signal to total protein using housekeeping proteins such as beta-actin or GAPDH from the same gel as the protein of interest to confirm equal protein loading across conditions. Also, as recommended by the Reviewer above, we have included normalization to Ponceau S for total protein normalization, in a few blots.

It is true that we do not have AKT2 expression for the data in Fig. 2f, and unfortunately, the lysates from this experiment are no longer available to run this blot. For this reason, in this experiment, we normalize pAKT2 to beta-actin. However, we show pAKT2 and either total AKT2 or total AKT in numerous other experiments, including those in Fig. 2a, Fig. 2i, Fig. 8k, and Fig. 8m, and data are all consistent with increased pAKT2 (and downstream signaling) upon induction of SWELL1 expression, or by SN-40X compounds.

Yes, SWELL1 expression is missing in Fig. 2i, in which the focus of the experiment was to evaluate SN-401 mediated pAKT2 and pAS160 induction, as well as the SWELL1 dependence of SN-401 signaling by comparing with SWELL1 KO 3T3-F442A adipocytes. As in Fig. 2f, the lysates from the experiment in Fig. 2i are also no longer available. However, we demonstrate SN-40X mediated SWELL1 protein induction in Fig. 2h, as well as in numerous other experiments including Fig. 2f, Fig. 8e-f, Fig. 8g-h, Fig 8k-l, Fig. 9a-b *in vitro* and Fig. 3a-b *in vivo*. So, although SWELL1 protein blots are absent in this specific experiment, there are multiple examples throughout the manuscript of this SN-40X effect on SWELL1 protein levels.

9. The effect of SN-40X on body weights seems different in Fig.4f vs Table S5, it is critical to resolve this discrepancy.

We have checked this and it appears to be accurate.

10. In Fig.8b and 8c, the control groups were all treated with 12 weeks HFD, why the glucose level is not comparable?

In old Fig. 8b (now new Fig. 10b), the control group is pre-SN-406, while in old Fig. 8c (now new Fig. 10c) the control group is Inactive 1. So, these control groups are different and should not be compared. It is possible that Inactive 1, though relatively inactive as compared to SN-401 and SN-406, might have some mild activity as suggested by the weak $I_{CI,SWELL}$ inhibitory activity at higher concentrations shown in new Fig. 6e. On the other hand, SN-406 GTT can be compared between these experiments, and these data are very similar.

11. The phosphorylation sites for AKT2, AS160, eNOS are not mentioned.

We thank the Reviewer for raising this point. We have now included the specific phosphorylation sites in the main text, methods and figure legends.

12. Even though eNOS is a downstream target of AKT in HUVECs, what is the physiological rationale of studying eNOS phosphorylation here?

We thank the Reviewer for raising this point. In addition to being a downstream target of AKT there are a number of reasons we included these data. First, it demonstrates SN-40X mediated pAKT2 signaling and downstream p-eNOS signaling in a human primary cell, as opposed to a differentiating murine cell line. Second, the p-eNOS induction in HUVECs by SN-40X compounds is very robust and thus provides a good model system for evaluating SN-401 dose-responses, and SN-40X versus inactive signaling. Third, based on our recently published findings linking endothelial SWELL1 depletion to impaired AKT-eNOS signaling and vascular dysfunction (Alghanem, A. et al. Elife 2021), SN-40X induction of peNOS in HUVECs suggests that SN-40X compounds may improve eNOS signaling and vascular function in metabolic syndrome. Therefore, this result may predict SN-40X compounds improving cardiovascular end-points and outcomes in the setting of T2D and metabolic syndrome, and this is important for a novel T2D therapeutic. We added a few lines in the manuscripts (lines 442-447) to provide some of this rationale.

13. In general, sample size in many experiments is too small, for example, n=2 in Fig.2g, Fig.6j, Fig.6o; n=3 in Fig.2i, Fig.5b, Fig.6f, Fig.7a; n=4 in Fig.5c, Fig.6h. Likewise, the use of n=5-7 mice per group is on the low end as there is inherent variability in these in vivo models.

We agreed with the reviewer on this and increased the samples sizes for many of these experiments listed above.

Reviewers' Comments:

Reviewer #1:

Remarks to the Author:

The authors have addressed my concerns.

Reviewer #2:

Remarks to the Author:

Authors have revised the manuscript carefully, the modified content is good enough to solve the problems raised by reviewers. From my point of view, the revised version is of good quality and meets the requirements of the journal, so I recommend it can be accepted.

Reviewer #3:

Remarks to the Author:

The authors made a considerable effort to meet the previous reviewer's concerns. They have performed several additional experiments and have shown new experimental data, which made this research more readable and valuable. The results of changes in adipocyte size and high blood FGF21 levels seem very interesting. Nevertheless, in the current manuscript, the authors should describe in more detail on how SN-401 prevented NAFLD progression.

In Fig 3 and 4, SN-401 appears to act primarily on adipose tissue and pancreatic β -cells. In other words, increased glucose uptake in skeletal muscle, suppression of fat accumulation in the liver, and improved hepatic insulin resistance may be caused by improved insulin sensitivity in adipose tissue. On the other hand, the phenomenon of SN-401 increasing the expression of SWELL1 protein has been observed not only in adipocytes and preadipocytes but also in HUVEC cells, and it is possible that it causes widespread changes in many tissues.

Even though the authors mention that the duration of treatment with SN-401 is relatively short, the suppression of fatty liver seems to be remarkable. Again, what do the authors think about the direct effect of SN-401 on the liver?

Since a series of results showed that the metabolic phenotype is improved when SN-401 inhibits the downregulation of SWELL1 in pancreatic beta cells and adipose tissue in pathological models, it would be good to evaluate whether the protein expression of SWELL1 in the liver and skeletal muscle is decreased in HFD-fed mice. If there is no change in the expression of SWELL1 in the liver or skeletal muscle in the DIO model, improved insulin resistance in the liver and skeletal muscle can be considered to be secondary changes, even without using tissue-specific SWELL1 KO of peripheral tissues (adipose/skeletal muscle).

It seems important to investigate in which tissues SN-401 acts, which could lead to a novel treatment for diabetes and NAFLD.

Reviewer #4:

Remarks to the Author:

The authors have done a superb job of answering all of the criticisms in a constructive and fully satisfying way. I strongly recommend acceptance.

Responses to Reviewer's Comments – second revision

We thank the Reviewers for their careful reading and evaluation of the manuscript, and are appreciative and thankful for Reviewers 1, 2 and 4 enthusiastically accepting the paper for publication. Reviewer 3 requested some additional experiments to determine if SWELL1 and associated signaling was directly increased in liver and skeletal muscle by SN-401, or if the effects on liver insulin sensitivity and glucose production were instead indirect, as might be concluded if SWELL1 protein levels in liver were unchanged. Our new experiments and responses are outlined below, and all changes in the manuscript are highlighted.

Reviewer #1 (Remarks to the Author):

The authors have addressed my concerns.

Reviewer #2 (Remarks to the Author):

Authors have revised the manuscript carefully, the modified content is good enough to solve the problems raised by reviewers. From my point of view, the revised version is of good quality and meets the requirements of the journal, so I recommend it can be accepted.

Reviewer #4 (Remarks to the Author):

The authors have done a superb job of answering all of the criticisms in a constructive and fully satisfying way. I strongly recommend acceptance.

Reviewer #3: (Remarks to the Author):

“The authors made a considerable effort to meet the previous reviewer’s concerns. They have performed several additional experiments and have shown new experimental data, which made this research more readable and valuable. The results of changes in adipocyte size and high blood FGF21 levels seem very interesting. Nevertheless, in the current manuscript, the authors should describe in more detail on how SN-401 prevented NAFLD progression.

In Fig 3 and 4, SN-401 appears to act primarily on adipose tissue and pancreatic β -cells. In other words, increased glucose uptake in skeletal muscle, suppression of fat accumulation in the liver, and improved hepatic insulin resistance may be caused by improved insulin sensitivity in adipose tissue. On the other hand, the phenomenon of SN-401 increasing the expression of SWELL1 protein has been observed not only in adipocytes and preadipocytes but also in HUVEC cells, and it is possible that it causes widespread changes in many tissues. Even though the authors mention that the duration of treatment with SN-401 is relatively short, the suppression of fatty liver seems to be remarkable. Again, what do the authors think about the direct effect of SN-401 on the liver?

Since a series of results showed that the metabolic phenotype is improved when SN-401 inhibits the downregulation of SWELL1 in pancreatic beta cells and adipose tissue in pathological models, it would be good to evaluate whether the protein expression of SWELL1 in the liver and skeletal muscle is decreased in HFD-fed mice. If there is no change in the expression of SWELL1 in the liver or skeletal muscle in the DIO model, improved insulin resistance in the liver and skeletal muscle can be considered to be secondary changes, even without using tissue-specific SWELL1 KO of peripheral tissues (adipose/skeletal muscle). It seems important to investigate in which tissues SN-401 acts, which could lead to a novel treatment for diabetes and NAFLD.”

To address this Reviewers concerns we evaluated SWELL1 protein expression in liver by Western blot in both HFD- and KKAY- T2D mouse models (new **Supplementary Fig. 1a&b**). As described in the Results (highlighted, lines 169-173) hepatic SWELL1 is induced 9-fold and 3.2-fold in HFD and

KKA^y models respectively. This finding is consistent with our previously published work that examined hepatic SWELL1 expression in early obesity (Xie, L, et al, Channels 2017). We speculate that this may represent a compensatory increase in hepatic SWELL1 protein in the face of impaired SWELL1 trafficking/signaling under glucolipotoxic conditions in the liver and/or other tissues, or an increase in SWELL1 from infiltrating cells such as macrophages, and other inflammatory cells associated with steatohepatitis. In response to SN-401 treatment we observed no measurable change in liver SWELL1 (new Fig. 4g) in KKA^y mice, in contrast to the increases observed in adipose tissue (Fig. 3a&b). Similarly, treating primary hepatocytes *in vitro* with SN-401 for 24-48 hours yielded no measurable change in SWELL1 protein (new Fig. 4h&i, Supplementary Fig. 5), but did demonstrate a clear induction in pAS160 signaling in WT hepatocytes (new Fig. 4h&j, Supplementary Fig. 5), but not in hepatocytes from liver-specific SWELL1 KO mice (Alb-Cre;SWELL1^{fl/fl} mice; new Fig. 4h&j, Supplementary Fig. 5) – indicating a SWELL1-dependent mechanism of SN-401 action on hepatic AS160 signaling, hepatic glucose uptake, and glucose metabolism. These findings are described in the Results section (highlighted, lines 280-290; 309-310), and Discussion (highlighted, lines 616-618; 623). **Collectively, these data suggest that SN-401 is acting at least in part directly on liver signaling in a SWELL1-dependent manner, however, the relative contributions of SN-401 action via direct versus indirect mechanisms (i.e. FGF-21) remains to be determined in future work.**

In the second round of review the Reviewer raised a new question not mentioned in the first round of review. This was whether SN-401 could also mediate SWELL1 induction in skeletal muscle as another putative site of SN-401 action to impact glucose homeostasis, in addition to adipose, liver, and pancreatic β -cells. This question could in part be motivated by our findings in skeletal muscle SWELL1 KO mice (Kumar, A., et. al. Elife 2020). To address this question, we measured SWELL1 protein, AKT2, AS160, GSK3 β signaling in soleus muscle from HFD-T2D mice treated with SN-401 as compared to vehicle (new Fig. 4e&f). We found a trend toward SN-401 mediated SWELL1 induction in soleus muscle, associated with significant increases in AKT2, AS160 and GSK3 β signaling (new Fig. 4e&f). **These data support the results of the traced ³H-glucose and ¹⁴C-deoxyglucose euglycemic hyperinsulinemic clamps (Fig. 4c&d) and support the notion that SN-401 may act directly on multiple tissues to improve glycemic control in T2D models.** These findings are described in the Results section (highlighted, lines 280-283; 288-290; 309-310) and Discussion (highlighted, lines 616-618; 623).

We thank the Reviewer for these additional comments and hope that these additional experiments and manuscript updates are satisfactory.

Reviewers' Comments:

Reviewer #3:

Remarks to the Author:

The authors responded to my concerns in a very appropriate manner. By performing additional experiments, the authors showed that the protein expression of SEWLL1 in the liver is elevated in the pathological model and that SN401 is involved in glucose metabolism in the liver via the AS160-mediated pathway. They also presented that SN-401 can enhance insulin sensitivity in skeletal muscle through increasing SWELL1. These results were very interesting and seemed to be an important finding for further research. I could understand that SN401 acts directly and indirectly on multiple organs to improve systemic metabolism in T2D and NAFLD. As a reviewer, I have no further concern.